# Black-Box Combinatorial Optimization with Order-Invariant Reinforcement Learning

Olivier Goudet [1]   Quentin Suire [1]   Adrien Goëffon [1]   Frédéric Saubion [1]   Sylvain Lamprier [1]

## Abstract

We introduce an order-invariant reinforcement learning framework for black-box combinatorial optimization. Classical estimation-of-distribution algorithms (EDAs) often rely on learning explicit variable dependency graphs, which can be costly and may fail to capture complex interactions efficiently. In contrast, we parameterize a multivariate autoregressive generative model trained without a fixed variable ordering. By sampling random generation orders during training, a form of information-preserving dropout, the model is encouraged to be invariant to variable order, promoting search-space diversity, and shaping the model to focus on the most relevant variable dependencies, improving sample efficiency. We adapt Group Relative Policy Optimization (GRPO) to this setting, providing stable policy-gradient updates from scale-invariant advantages. Across a wide range of benchmark problem instances of varying sizes, our method frequently achieves the best performance and consistently avoids catastrophic failures.

## 1. Introduction

Black-box optimization (Audet & Kokkolaras, 2016; Brochu et al., 2010) consists of maximizing a function $f : \mathcal{X} \to \mathbb{R}$ over the discrete space $\mathcal{X}$ without any structural or analytical knowledge of $f$. The function $f$ is typically costly to evaluate (e.g., computationally expensive simulation, querying a physical experiment, or executing a complex algorithm). The interactions among the variables of $f$ are not available, making black-box optimization particularly challenging, especially in high-dimensional and structured discrete domains (Doerr et al., 2019).

A wide range of methods and concepts have been explored to solve black-box optimization problems. Among them, Bayesian optimization (BO) is a model-based optimization framework that constructs a probabilistic surrogate model over the objective function and uses an acquisition function to determine where to sample next in the search space. It is particularly effective for global optimization under tight evaluation budgets, making it well-suited for expensive black-box problems (Forrester & Keane, 2009; Shahriari et al., 2015; Frazier, 2018). Evolutionary Algorithms (EAs) are also recognized as powerful methods for solving discrete black-box optimization problems. These metaheuristics operate by iteratively evolving a population of candidate solutions through variation operators (mutation, crossover) and selection mechanisms (Corne & Lones, 2025). Unlike Bayesian optimization, EAs do not build explicit models of the objective function, making them more flexible and easier to implement (Back, 1996; Eiben & Smith, 2015).

As a specific subclass of EAs, Estimation-of-Distribution Algorithms (EDAs) are stochastic black-box optimization methods that guide the search for optima by explicitly learning and sampling from a probabilistic model $P$ of promising candidate solutions by means of a distribution that captures patterns in high-performing solutions (Mühlenbein & Paaß, 1996; Larranaga, 2002). EDAs can be conceptually positioned between the two main paradigms of black-box optimization, EAs and BO. Some widely used and effective EDAs such as the Covariance Matrix Adaptation Evolution Strategy (CMA-ES) (Hansen & Ostermeier, 2001; Hansen, 2016)—designed for continuous landscapes, and Population-Based Incremental Learning (PBIL) (Baluja, 1994) for discrete landscapes, can also be interpreted within the Information-Geometric Optimization (IGO) framework (Ollivier et al., 2017). This connection provides a formal interpretation of EDAs as performing natural gradient descent in the space of probability distributions, thus explaining their ability to fine-tune solutions and converge reliably in continuous or discrete spaces. While continuous EDAs, particularly CMA-ES, have attracted significant attention, a less explored body of research focuses on EDAs for discrete and combinatorial spaces. Early work in this area has demonstrated the effectiveness of multivariate discrete EDAs in applications such as scheduling and routing (Lozano, 2006).

---

[1]LERIA, Université d'Angers, 2 Boulevard Lavoisier Angers 49045 France. Correspondence to: Olivier Goudet <olivier.goudet@univ-angers.fr>.

*Proceedings of the $43^{rd}$ International Conference on Machine Learning*, Seoul, South Korea. PMLR 306, 2026.

In this paper, we highlight and exploit the connection between EDAs and reinforcement learning (RL), as both frameworks rely on sampling from a generative model (a policy in RL) and updating this model to increase the likelihood of high-fitness or high-reward samples. From this perspective, we revisit discrete multivariate EDAs by modeling the joint distribution with neural-network–parameterized conditional distributions, trained with modern policy-gradient RL methods. In the proposed approach, each variable is modeled conditionally on the others, yielding a highly flexible generative model capable of capturing complex inter-variable dependencies while keeping the total number of parameters polynomial in the instance size. However, this introduces the challenging choice of a generation order, as solution generation is thus formulated as a sequential assignment of variable values.

Inspired by recent work on permutation-invariant autoregressive language generation (Pannatier et al., 2024), we depart from classical EDAs such as MIMIC (De Bonet et al., 1996) and BOA (Pelikan, 2005), which rely on an explicit generation order. Instead of assuming or learning a sparse directed acyclic graph to define this order—which typically requires repeatedly solving an NP-hard structure-learning problem during exploration—we propose an order-invariant multivariate generative model. This undirected model does not commit to any fixed variable ordering during generation.

In contrast with RL-based construction methods (Kim et al., 2022; Kwon et al., 2020), which exploit symmetries of the solution space itself (e.g., rotations or permutations of solutions), we focus on invariance with respect to the generation order. This property is orthogonal to solution-space symmetries: it leaves the solution space unchanged, while encouraging flexible internal structure in the generative model and facilitating the discovery of dependencies between variables. Experiments on a diverse set of benchmark problems and instance sizes illustrate the advantages of this order-invariant RL approach for black-box optimization.

To summarize, beyond our contributions to black-box optimization and EDAs through the use of RL techniques, our work advances machine learning methodology along three main directions:

- We show that varying generation orders can be incorporated into PPO-like algorithms in a theoretically principled way.

- We demonstrate that the structural input dropout induced by random generation orders acts as an effective regularization mechanism, while preserving information from the underlying joint distribution.

- We provide a theoretical analysis of a scale-invariant reward function based on sample ranks within a pop-

ulation, which can be interpreted as a form of group-relative advantage in GRPO-like approaches (Shao et al., 2024).

The remainder of this paper is organized as follows. Section 2 introduces the discrete black-box optimization problem, reviews related work and discusses the motivations for this work. Section 3 presents the derivation of our proposed RL-EDA approach, which builds on a GRPO RL backbone and is designed to tackle this class of problems. Section 4 reports empirical results comparing our algorithm with state-of-the-art methods. Various versions of the approach are also compared to analyze the benefits of each of its components.

## 2. Preliminaries: Problem Setting, Related Work and Motivations

In this section, we first formally introduce the discrete black-box optimization problem. We then review existing work on multivariate EDAs proposed to tackle such problems. Finally, we discuss the opportunities offered by neural generators in this context, particularly regarding their flexibility in capturing implicit inter-variable dependencies. We also highlight the potential benefits of leveraging random variable orderings for both generation and training under stringent sample-efficiency constraints within the EDA training regime.

### 2.1. Discrete Black-Box Optimization

Let $\mathcal{X} = \mathcal{X}_1 \times \cdots \times \mathcal{X}_n$ be the discrete search space of size $n$, where each $\mathcal{X}_j$ is a finite set (binary or categorical), and let $f : \mathcal{X} \to \mathbb{R}$ be an objective function accessible only as a black box, i.e., without any structural information (such as convexity or smoothness). A combinatorial optimization (CO) problem is then defined by the pair $(\mathcal{X}, f)$. Without loss of generality, the task is to maximize $f$: $\arg\max_{x \in \mathcal{X}} f(x)$. In the following, $x = (x_1, \ldots, x_n) \in \mathcal{X}$ denotes a candidate solution (not necessarily the best) of the CO problem. $X_i$ denotes the variable associated with $\mathcal{X}_i$, whose value in $\mathcal{X}_i$ is $x_i$. Various existing solving techniques for black-box CO include Bayesian optimization methods and evolutionary algorithms, which have been enhanced by machine learning techniques (Talbi, 2021). More related work on combinatorial optimization is given in Appendix A.

### 2.2. Multivariate Estimation of Distribution Algorithms

Multivariate EDAs are evolutionary algorithms that solve a CO problem by iteratively building and updating a probabilistic model over the search space $\mathcal{X}$. An EDA with parameters $(\mu, \lambda) \in \mathbb{N}^2$ with $0 < \mu < \lambda$ performs the following steps at each iteration $t$:

1. Draw a population of $\lambda$ candidate solutions $x^1, \ldots, x^\lambda$ from the model $P_t$ and compute fitness values $f^i = f(x^i)$, for $i = 1, \ldots, \lambda$.

2. Select the $\mu$ best individuals $\mathcal{S}_t = \{x^{r_i} : i \in [\![1, \mu]\!]\}$, where $(r_1, \ldots, r_\lambda)$ is a permutation of $[\![1, \lambda]\!]$ such that $f^{r_1} \geq \cdots \geq f^{r_\lambda}$, and use $\mathcal{S}_t$ to estimate the updated probabilistic model $P_{t+1}$.

Following this framework, EDAs mainly differ in how they model the generative distribution $P_t$ used to sample new candidate solutions at each generation $t$. Some approaches, such as PBIL (Baluja, 1994) or UMDA (Mühlenbein & Paaß, 1996), approximate $P_t$ as a product of independent univariate distributions: $P_t(x) = \prod_{i=1}^n P_t^i(X_i = x_i)$, where $P_t^i$ denotes the $i$-th marginal distribution. While such approaches have proved effective on problems with little or no interaction among variables, they suffer from important limitations: they fail to capture complex inter-variable relationships (including combinatorial or logical dependencies) and are prone to premature convergence or loss of diversity in multimodal landscapes. To overcome these limitations, classical multivariate EDAs need to employ more expressive probabilistic models that explicitly capture dependencies between variables from best candidates in $S_t$ at each generation $t$. In the case of Bayesian networks, dependencies are represented by a directed acyclic graph (DAG) $\mathcal{G} = (\mathcal{V}, \mathcal{E})$, whose set of vertices $\mathcal{V}$ contains all the variables $X_j$ for $j = 1, \ldots, n$ and whose directed edges $\mathcal{E}$ represent causality relationships. Hence, at any iteration $t$ of the EDA process, the joint density $P_t(x)$ can be factorized as the product of the densities of each variable conditionally on its parents as $P_t(x) = \prod_{j=1}^n P_t(X_j = x_j | X_{\text{Pa}(j;\mathcal{G}_t)} = x_{\text{Pa}(j;\mathcal{G}_t)})$ (Markov factorization) with $\mathcal{G}_t = (\mathcal{V}, \mathcal{E}_t)$ the considered DAG at iteration, $X_{\text{Pa}(j;\mathcal{G}_t)} = \{X_i \in \mathcal{V} : (X_i, X_j) \in \mathcal{E}_t\}$ the set of the parents of the variable $X_j$ in $\mathcal{G}_t$ and $x_{\text{Pa}(j;\mathcal{G}_t)}$ their corresponding values. Given a DAG $\mathcal{G}_t$, such a factorization allows to significantly reduce the number of required parameters to approximate $P_t$. It also permits sampling the variables sequentially according to a topological ordering consistent with the causal dependencies encoded by the graph. However, optimal DAGs are usually unknown at the beginning of the process, and need to be learned efficiently from selected candidates $S_t$ at each generation, together with the parameters of each factor of the Markov factorization (more details on EDAs with DAGs can be found in Appendix A).

## 2.3. The Case of Neural Estimators

Traditionally, EDAs based on Bayesian networks estimate each component of the Markov factorization by contingency tables reporting counts of all joint realizations of the dependent variables together with the combinations of its parents' values. In this setting, restricting the dependencies of each

outcome to a small subset of causal variables is crucial to avoid the exponential growth of complexity with the problem dimension. This limitation has motivated a long line of research on structural learning heuristics, pruning strategies, and regularization techniques designed to control the combinatorial explosion (Echegoyen et al., 2008).

Neural estimators fundamentally alter this picture. In classical EDAs, learning an explicit dependency graph was unavoidable: the sampling model could only be specified once the graph structure had been identified. Neural approaches dispense with this requirement. By parameterizing the joint distribution directly—often through autoregressive factorizations with arbitrary variable orderings (Germain et al., 2015; Uria et al., 2016) or via invertible transformations in flow-based models (Papamakarios et al., 2021)—they sidestep the need to commit to a learned structure. However, despite their success in density estimation and generative modeling, such neural approaches have scarcely been explored in the context of multivariate EDAs. To the best of our knowledge, no prior work has applied autoregressive models to EDAs or investigated their interaction with iterative optimization dynamics.

In practice, fitting a flexible neural density estimator is frequently simpler and more robust than inferring the "correct" graph, especially under the limited and evolving sample regimes typical of EDAs. Following an autoregressive model, we can consider any given factorization using any order of variables. That is, given an arbitrary order $\sigma$ of the dimensions of the problem, we can write $P(X = x) = \prod_{i=1}^n P(x_{\sigma_i} | x_{\sigma<i})$, where $x_{\sigma_i}$ stands as the value of the $i$-th dimension of $x$ in the permutation $\sigma$ and $x_{\sigma<i}$ corresponds to the sequence of values of $x$ with rank less than $i$ in permutation $\sigma$ (with $x_{\sigma<1}$ standing as an empty sequence). Given $N$ samples of $P$, this can be estimated by a neural network $P_\theta$, with parameters $\theta$ obtained via maximum likelihood estimation (MLE): $\arg\max_{\theta \in \Theta} \frac{1}{N} \sum_{j=1}^N \prod_{i=1}^n P_\theta(x_{\sigma_i}^j | x_{\sigma<i}^j)$, where $x^1, \ldots, x^N$ are sampled from the target distribution $P$. We note that this holds for any given permutation $\sigma$. In particular, assuming infinite amounts of data and infinite capacity of the used neural networks, at convergence of the MLE, we get that: $\forall \sigma, \sigma' : P_\theta(X|\sigma) = P_{\theta'}(X|\sigma')$, where $\theta$ and $\theta'$ are optimal parameters (according to MLE) for permutation $\sigma$ and $\sigma'$ respectively. NADE (Uria et al., 2016) exploits this idea by defining ensembles of models, each associated with a different variable ordering, which enables sampling from a more diverse set of outcomes. Yet, to the best of our knowledge, such permutation-based ensembles have never been explored in multivariate EDAs, despite population diversity being a key ingredient for black-box optimization and effective exploration. Beyond sampling, we argue that training a single model across multiple orderings provides an additional benefit: it acts as a form of noise reduction

when learning from limited data, as it is typically the case in online EDAs. In Appendix E, we show that this mechanism can be interpreted as an information-preserving analogue of dropout, allowing the model to efficiently identify the dominant dependencies between variables while mitigating overfitting to transient fluctuations.

# 3. Multivariate EDA with Order-Invariant Reinforcement Learning

Our proposed algorithm for discrete black-box problems is a multivariate EDA (see Section 2.2) whose probabilistic model is encoded with a set of neural networks. The construction of a solution of the CO problem is seen as an episodic Markov Decision Process (MDP) with a reinforcement learning algorithm adapted for our setting. A toy example illustrating both the generation of solutions and the model update is provided in Figure 1.

## 3.1. Deep Reinforcement Learning for EDAs: Setting and Architectures

The EDA framework presented above can be easily cast as a reinforcement learning problem, defined on an MDP $\mathcal{M} = (\mathcal{S}, \mathcal{A}, P, R)$ where $\mathcal{S}$ is a set of states, $\mathcal{A}$ a set of actions, $P(s'|s, a)$ is the transition probability function, $R : \mathcal{S} \to \mathbb{R}$ is the reward function assigning a scalar reward to each reached states in $\mathcal{S}$. In the setting of multivariate EDAs, $\mathcal{S}$ corresponds to incomplete solutions from $\mathcal{X}$ (i.e. $S \equiv \{(\varnothing, 0, \sigma) : \sigma \in \Omega\} \cup \{((x_{\sigma_1} \ldots x_{\sigma_k}), k, \sigma) : x \in \mathcal{X}, \sigma \in \Omega, k \in [\![1, n]\!]\}$), with $\Omega$ the set of all possible generation orders of a sequence of indices $1, \ldots, n$ and $\varnothing$ an empty sequence that defines starting states $s_0$. For a given state $s_k = (x_{\sigma_{\leq k}}, k, \sigma)$, the set of possible actions $\mathcal{A}_k \subseteq \mathcal{A}$ is the domain of the $(k+1)$-th variable of the permutation $\sigma$ (i.e., $\mathcal{A}_k \equiv \mathcal{X}_{\sigma_{k+1}}$). Thus, transitions are deterministic: for any triplet $(s, a, s')$, with $s = (x_{\sigma_{\leq k}}, k, \sigma)$ and $a \in \mathcal{X}_{\sigma_{k+1}}$, $P(s'|s, a)$ is 1 iff $s' = (x'_{\sigma_{\leq k+1}}, k+1, \sigma)$ with $x'_{\sigma_{\leq k}} = x_{\sigma_{\leq k}}$ and $x'_{\sigma_{k+1}} = a$. Finally, rewards are non-zero for states from $\mathcal{S}$ that correspond to complete solutions of the problem only (i.e., those states that contain full instantiation of $\mathcal{X}$). In that setting, our goal is to optimize a parameterized stochastic generative policy $\pi_\theta(a_k \in \mathcal{X}_{\sigma_{k+1}} | s_k = (x_{\sigma_{\leq k}}, k, \sigma))$, which defines the probability of taking action $a_k$ in state $s_k$. For the binary setting where the discrete search space is $\mathcal{X} = \{-1, 1\}^n$, we model this generative policy as a neural logistic regressor as $\pi_\theta(a_k = 1 | s_k = (x_{\sigma_{\leq k}}, k, \sigma)) = \text{sigmoid}(g_{\theta_{\dim_\sigma(k)}}(x_{\sigma_{\leq k}}))$, with $g_{\theta_i}$ a neural network with parameter $\theta_i \in \mathbb{R}^m$ and $\dim_\sigma(k)$ the bijective function that returns the index of the dimension at rank $k$ in permutation $\sigma$. For categorical domains $\mathcal{X}_i$, we encode each of their $d$ categories as a one-hot vector where $X_{i,j} = 1$ iff the represented category is $j \in [\![1, d]\!]$, and $-1$ otherwise. For these outputs, we consider a softmax

over the logits produced by $g$ to produce the corresponding categorical distribution.

Rather than relying on neural models specifically dedicated for sequences, such as recurrent networks or Transformers (which are better suited to unstructured inputs), we propose to define $g$ as a classical MLP, parameterized with a different set of parameters for each individual output of the problem. For any order of generation $\sigma$ and any step $k$, we want to feed $g$ with a fixed-size vector as input. For a given step $k$ of a permutation $\sigma$, this is done by modeling the input $x_{\sigma_{<k}}$ as a vector of size $n$, where each dimension $x_i = 0$ (resp. $x_i$ is a zero vector for the categorical domains) iff $\text{rank}_\sigma(i) = \dim_\sigma^{-1}(i) \geq k$. During training, this comes down to applying a causal mask to candidate solutions, that masks future of $k$ in permutation $\sigma$. Note that, while $\theta \in \mathbb{R}^{n \times m}$ in our architecture, our work can be naturally extended by sharing parameters of hidden layers for scaling to very large problems without facing prohibitive training costs, as described in Appendix R.

## 3.2. Deep Reinforcement Learning for EDAs: Training

Given the setting stated above, the optimization seeks to maximize the expected global reward over trajectories $\tau = (s_0, a_0, \ldots, s_{n-1}, a_{n-1}, s_n)$: $J(\theta) = \mathbb{E}_{\tau \sim \pi_\theta}[R(\tau)]$, where $R(\tau)$ in our setting corresponds the fitness $f(x)$ computed for the full candidate $x \in \mathcal{X}$ contained in the last state of $\tau$ (i.e., $R((s_0, a_0, \ldots, s_n)) = f(x)$, iff $s_n = (x, n, \sigma)$). For a given $\sigma$, this is thus equivalent to maximizing $J^\sigma(\theta) = \mathbb{E}_{x \sim \pi_\theta(x|\sigma)}[f(x)]$, where $\pi_\theta(x|\sigma)$ stands for the probability of sampling $x$ as a sequence $x = (x_{\sigma_1}, \ldots, x_{\sigma_n})$ using our generative architecture.[1] Following the policy gradient theorem (Sutton et al., 1999), we get that parameters $\theta$ can be obtained using gradient updates defined as

$$\nabla_\theta J^\sigma(\theta) = \mathbb{E}_{x \sim \pi_\theta(x|\sigma)}[f(x) \sum_{k=1}^n \nabla_\theta \log \pi_\theta(x_{\sigma_k} | x_{\sigma_{<k}}, \sigma)]. \tag{1}$$

This formulation allows us to sample candidate solutions of the problem from the current distribution $\pi_\theta(x|\sigma)$ (which corresponds to $P_t(x)$ in the EDA framework described in Section 2.2), and then estimate an update of the generative distribution by computing a weighted average of gradients of $\log \pi_\theta(x|\sigma)$, with weights depending on the respective fitness of sampled $x$ (which is the analogue of step 2 from the EDA framework in Section 2.2). However, from updates defined in (1), each sample $x$ can be used for a unique gradient step only, which can be very sample inefficient. Moreover, updates of the policy are strongly dependent on its parameterization, which can lead to hazardous moves that induce catastrophic forgetting when using such neu-

---

[1]In the following of this section, we consider a fixed arbitrary order $\sigma$ for every state of the MDP. Using random variations of $\sigma$ is the subject of the next section.

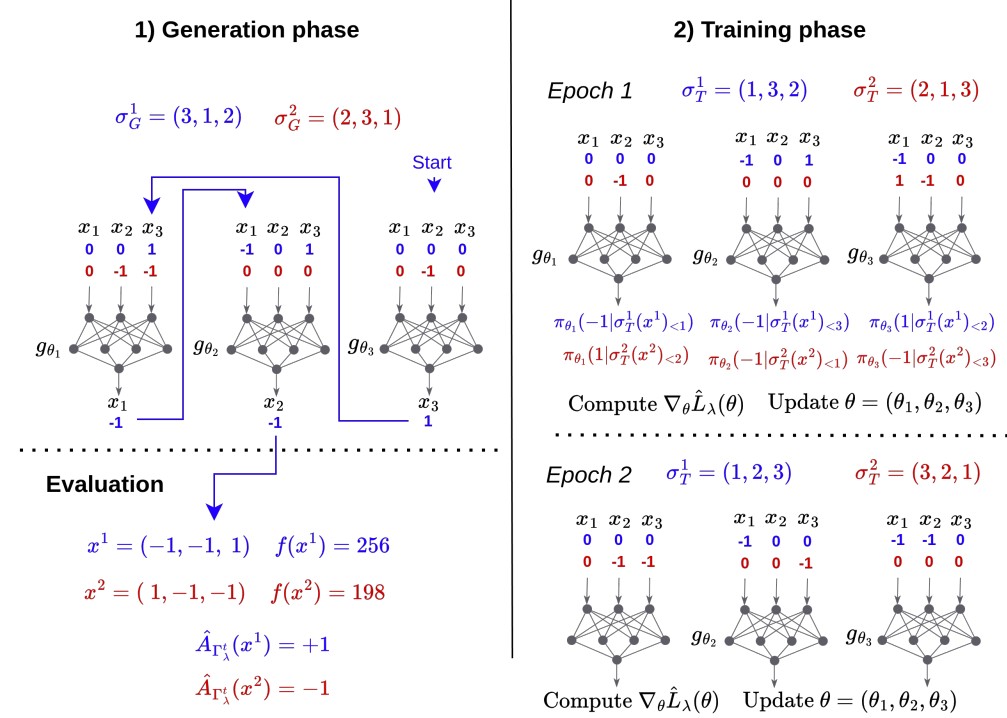

*Figure 1.* **Left.** Example of the generation, at time $t$, of a population $\Gamma_\lambda^t$ with $\lambda = 2$ individuals for a maximization problem with $n = 3$. The generation order of the first individual is indicated by blue arrows. When constructing this individual using the MDP and a given order $\sigma_G^1 = (3, 1, 2)$, we start with $x_{\sigma_G^1 < 1}^1 = (0, 0, 0)$ given as input to the neural network $g_{\theta_3}$, generating $x_3 = 1$. Then $x_{\sigma_G^1 < 2}^1 = (0, 0, 1)$ is provided to to $g_{\theta_1}$, which generates $x_1 = -1$. Finally, $x_{\sigma_G^1 < 3}^1 = (-1, 0, 1)$ is passed to $g_{\theta_2}$, which generates $x_2$, yielding the complete solution $(-1, -1, 1)$. Once all individuals in the population have been sampled, the evaluation phase starts, where advantages are computed such that $A_{\Gamma_\lambda^t}(x_{best}^i) = +1$ and $A_{\Gamma_\lambda^t}(x_{worse}^i) = -1$ (Eq. (5)). **Right.** Training phase over $E$ epochs with the $\lambda = 2$ solutions sampled at this generation. At each epoch, new generation orders $\sigma_G$ are sampled for each individual. Conditional action probabilities are then computed according to the corresponding causal masks. This allows the computation of $\hat{L}_\lambda(\theta)$ (Eq. (8)) and the update of $\theta = (\theta_1, \theta_2, \theta_3)$ by gradient ascent.

ral generators. To improve sample efficiency and stabilize training, the Proximal Policy Optimization (PPO) algorithm (Schulman et al., 2017), following TRPO (Schulman et al., 2015a), optimizes a surrogate objective function that penalizes deviations from a reference policy $\pi_{\theta_{\text{old}}}$, used for sampling, that will be denoted $\pi_{\theta^t}$ at generation $t$ of our EDA.

In our setting, the policy gradient update in (1) can be rewritten using importance sampling as an expectation under $\pi_{\theta^t}$. Approximating the state distribution $d^{\pi_\theta}$ by $d^{\pi_{\theta^t}}$, we obtain (see appendix B for details)

$$\nabla_\theta J^\sigma(\theta) \approx \mathbb{E}_{\pi_{\theta^t}(x|\sigma)} \sum_{k=1}^n \frac{\nabla_\theta \pi_\theta(x_{\sigma_k}|x_{\sigma_{<k}}, \sigma)}{\pi_{\theta^t}(x_{\sigma_k}|x_{\sigma_{<k}}, \sigma)} A^{\pi_{\theta^t}}(x_{\sigma_{<k}}, x_{\sigma_k}), \quad (2)$$

where $A^{\pi_{\theta^t}}(x_{\sigma_{<k}}, x_{\sigma_k})$ denotes the expected advantage of setting $X_{\sigma_k} = x_{\sigma_k}$ given $x_{\sigma_{<k}}$, while completing the trajectory with the reference policy. This formulation allows

multiple gradient steps for updating the policy (i.e., for obtaining $P_{t+1}$), given samples obtained using the policy (representing $P_t$) from the previous iteration $t$ of our EDA RL framework. However, the approximation in (2) (the choice of the PPO variant based on the Kullback–Leibler (KL) divergence is discussed in section F), which should be understood at the level of expected gradients, introduces an acceptable bias only when $\pi_\theta$ and $\pi_{\theta^t}$ are close, as measured by the KL divergence. Thus, following the this KL-based version of PPO, we consider the maximization of the regularized objective:

$$L^\sigma(\theta) = \mathbb{E}_{\pi_{\theta^t}(x|\sigma)} \sum_{k=1}^n \left[ \frac{\pi_\theta(x_{\sigma_k}|x_{\sigma_{<k}}, \sigma)}{\pi_{\theta^t}(x_{\sigma_k}|x_{\sigma_{<k}}, \sigma)} A^{\pi_{\theta^t}}(x_{\sigma_{<k}}, x_{\sigma_k}) \right.$$
$$\left. - \beta D_{\text{KL}}\left(\pi_{\theta^t}(\cdot|x_{\sigma_{<k}}, \sigma) \| \pi_\theta(\cdot|x_{\sigma_{<k}}, \sigma)\right) \right] \quad (3)$$

where $D_{\text{KL}}(\pi\|\pi')$ stands for the Kullback-Leibler (KL) divergence of $\pi$ from $\pi'$, and $\beta > 0$ is an adaptive penalty coefficient that controls the strength of the KL regularization. While PPO classically uses critic neural networks

to estimate advantages (e.g., using GAE (Schulman et al., 2015b)), we rather take inspiration from the GRPO approach (Shao et al., 2024), specifically dedicated for RL problems with global rewards from finite trajectories without discount, which avoids the need for a critic, by estimating scale-invariant advantages using a normalization of rewards obtained on a population of samples for a same problem.[2] Scale-invariance is particularly desirable in black-box optimization settings, as it enhances robustness to the scaling of objective values (Baluja, 1994; Doerr & Dufay, 2022; Goudet et al., 2025). Given a set of $\lambda$ candidate solutions $\Gamma_\lambda^t = \{x^i\}_{i=1}^\lambda$, each sampled from $\pi_{\theta^t}(x|\sigma)$, we consider at each iteration $t$ of the process the maximization of

$$\hat{L}_\lambda^\sigma(\theta) = \frac{1}{\lambda} \sum_{x^i \in \Gamma_\lambda^t} \sum_{k=1}^n \left[ \frac{\pi_\theta(x_{\sigma_k}^i | x_{\sigma<k}^i, \sigma)}{\pi_{\theta^t}(x_{\sigma_k}^i | x_{\sigma<k}^i, \sigma)} A_{\Gamma_\lambda^t}(x^i) \right. $$
$$\left. - \beta D_{\mathrm{KL}} \left( \pi_{\theta^t}(\cdot | x_{\sigma<k}^i, \sigma) \,\|\, \pi_\theta(\cdot | x_{\sigma<k}^i, \sigma) \right) \right], \quad (4)$$

where $A_{\Gamma_\lambda^t}(x)$ is the relative performance of candidate $x$ compared to other solutions from $\Gamma_\lambda^t$. In this paper, we consider advantages computed as

$$A_{\Gamma_\lambda^t}(x) = U \left( \frac{\mathrm{rk}(x, \Gamma_\lambda^t, f)}{\lambda - 1} \right), \quad (5)$$

where $U$ is a non-increasing utility function and $\mathrm{rk}(x^i, \Gamma_\lambda^t, f)$ is the rank of the individual $i$ in the population $\Gamma_\lambda^t$ given its fitness $f(x^i)$. Formally, $\mathrm{rk}(x, \Gamma, f) = |\{x' \in \Gamma : f(x') > f(x)\}|$. This advantage formulation, which makes the algorithm invariant under monotone transformation of the fitness function $f$, is grounded in the Information-Geometric Optimization (IGO) framework (Ollivier et al., 2017). We discuss the connexion of our approach with IGO in Appendix G.

### 3.3. Order-Invariant Reinforcement Learning for EDAs

In the previous section, we introduced a multivariate RL-EDA that uses a predetermined arbitrary generation order $\sigma$. The aim of this section is to adapt this algorithm for dealing with variations of this generation order, which we claim can strongly benefit exploration and learning in our black-box optimization setting.

Given a generation order distribution $\xi(\sigma)$, we can consider the expectation $L(\theta) = \mathbb{E}_{\sigma \sim \xi(\sigma)} L^\sigma(\theta)$ in place of using $L^\sigma(\theta)$ with a fixed known order $\sigma$. Let for convenience of the following $\sigma(x)_{<k}$ denote a masking (i.e., removing) of any dimension from $x$ whose rank in permutation $\sigma$ is greater or equal than the one of dimension $k$ (i.e., $\forall i \in [\![1, n]\!], X_i \in \sigma(X)_{<k} \iff \mathrm{rank}_\sigma(i) < \mathrm{rank}_\sigma(k)$).

---

[2]We compare a baseline that uses a critic to estimate advantages with our GRPO approach in Appendix Q.

Using this, we can rewrite the objective (3), as

$$L(\theta) = \mathbb{E}_{\sigma \sim \xi(\sigma)} \mathbb{E}_{\pi_{\theta^t}(x|\sigma)}$$
$$\sum_{k=1}^n \left[ \frac{\pi_\theta(x_k | \sigma(x)_{<k})}{\pi_{\theta^t}(x_k | \sigma(x)_{<k})} A^{\pi_{\theta^t}}(\sigma(x)_{<k}, x_k) \right.$$
$$\left. - \beta D_{\mathrm{KL}} \left( \pi_{\theta^t}(\cdot | \sigma(x)_{<k}) \,\|\, \pi_\theta(\cdot | \sigma(x)_{<k}) \right) \right]. \quad (6)$$

A notable difference in this writing compared to previous ones is that the inner sum from $k = 1$ to $n$ is taken in the original dimension ordering of the problem, rather than in the generation order. While fully equivalent, this formulation allows us to introduce a second source of variation, specifically dedicated for incentivizing order-invariance of the policy. Let $\xi(\sigma_T | \sigma_G)$ be a conditional distribution that samples a transformation $\sigma_T \in \Omega$ of a given initial permutation $\sigma_G \in \Omega$. We propose to use this transformed permutation $\sigma_T$ to train the new policy $\pi_\theta$, given samples from the old policy using the former permutation $\sigma_G$ used for generation. We get (derivation detailed in Appendix C)

$$L(\theta) = \mathbb{E}_{\sigma_G \sim \xi(\cdot), \sigma_T \sim \xi(\cdot | \sigma_G)} \mathbb{E}_{\pi_{\theta^t}(x|\sigma_G)}$$
$$\sum_{k=1}^n \left[ \frac{\pi_\theta(x_k | \sigma_T(x)_{<k})}{\pi_{\theta^t}(x_k | \sigma_G(x)_{<k})} A^{\pi_{\theta^t}}(\sigma_G(x)_{<k}, x_k) \right.$$
$$\left. - \beta D_{\mathrm{KL}} \left( \pi_{\theta^t}(\cdot | \sigma_G(x)_{<k}) \,\|\, \pi_\theta(\cdot | \sigma_T(x)_{<k}) \right) \right]. \quad (7)$$

As in the previous section, we finally consider a Monte-Carlo approximation of this quantity at each iteration, using scale normalized global advantages, given a set of $\lambda$ i.i.d. candidate solutions associated with their own order of generation $\Gamma_\lambda^t = \{(x^i, \sigma_G^i)\}_{i=1}^\lambda$. For each component $i$ in this set, an order $\sigma_G^i$ is first sampled from $\xi$, then $x^i$ is sampled from $\pi_{\theta^t}(\cdot | \sigma_G)$. We get

$$\hat{L}_\lambda(\theta) = \frac{1}{\lambda} \sum_{(x^i, \sigma_G^i) \in \Gamma_\lambda^t} \mathbb{E}_{\sigma_T \sim \xi(\cdot | \sigma_G^i)}$$
$$\sum_{k=1}^n \left[ \frac{\pi_\theta(x_k^i | \sigma_T(x^i)_{<k})}{\pi_{\theta^t}(x_k^i | \sigma_G^i(x^i)_{<k})} A_{\Gamma_\lambda^t}(x) \right.$$
$$\left. - \beta D_{\mathrm{KL}} \left( \pi_{\theta^t}(\cdot | \sigma_G^i(x^i)_{<k}) \,\|\, \pi_\theta(\cdot | \sigma_T(x^i)_{<k}) \right) \right]. \quad (8)$$

A theoretical study of the convergence of the algorithm under this objective in the infinite-data and infinite-capacity regime is provided in Appendix D. This formulation allows us to explore several variants of our training process, which we refer to as:
– $(\delta, \delta)$-RL-EDA: uses a fixed arbitrary order for both generation and training (i.e., $\xi$ and $\xi(\cdot|\sigma)$ are Dirac distributions centered on the original order $\sigma$ of the problem);
– $(\delta, \sigma)$-RL-EDA: uses a fixed arbitrary order for generation, while for training $\xi(\cdot|\sigma_G)$ is a uniform distribution;
– $(\sigma, \delta)$-RL-EDA: uses the same randomly sampled order

$\sigma_G$ for both generation and training, with $\xi$ a uniform distribution over $\Omega$ and $\xi(\cdot|\sigma_G)$ is a Dirac centered on $\sigma_G$;
– $(\sigma, \sigma)$-RL-EDA: uses two independent sources of randomness in the training process, where both the generation order $\sigma_G$ and the training order $\sigma_T$ are sampled from a uniform distribution over $\Omega$.

The pseudo-code of the full algorithm, including permutation noise during training, is provided in Appendix H (Algorithm 1). A toy example illustrating the effect of these permutation noises during training is shown in Figure 1 (right panel).

Note that incorporating varying causal graphs is also possible within this framework by using masks $\sigma(x)_{<k}$ that hide the values of non-parent variables of $x_k$ in $x$, in addition to all dimensions whose rank in $\sigma$ is greater or equal to $k$. We experiment with this structural dropout as a complement or replacement for causal masks for the different versions of the multivariate EDA in Appendices M.1 and M.2. In Appendix S, we compare our algorithm with a baseline variant, where the sequential order of generation is replaced by a Gibbs sampling, which is another way to build an order-invariant RL-EDA. We also describe in Appendix I a version called Learned-$\sigma$-RL-EDA which uses a Plackett-Luce (PL) distribution (Plackett, 1975) $\xi_w^{PL}$ to learn the generation order of solutions online. This PL distribution is parameterized by the vector $w \in \mathbb{R}^n$ trained by gradient descent with the reparameterization trick proposed by (Grover et al., 2019).

# 4. Experiments

We first consider the following NP-hard problems in this work (viewed as black-box CO problems): the Quadratic Unconstrained Binary Optimization problem (QUBO) (Kochenberger et al., 2014), the pseudo-boolean NK landscape problem (Kauffman & Weinberger, 1989), and its extension with ternary variables, referred to as NK3. For each problems $pb$, we generate instances of size $n \in \{64, 128, 256\}$. For each size, we considered different types $K$ of instances. We generate 10 instances for each tuple $(pb, n, K)$. For each problem instance, we allow a maximum budget of 10,000 objective function evaluations and perform 10 independent restarts. Details regarding the instances and experimental protocol are provided in Appendix J. The source code for the algorithm, along with the instances, complete results, and instructions for reproducing the experiments, is available at: https://github.com/GoudetOlivier/RL-EDA.

## 4.1. Comparison of the Different Versions of Reinforcement Learning Multivariate EDA

In this section, we first aim to compare the five different versions of multivariate-RL-EDA presented in Section 3.3: $(\delta, \delta)$-RL-EDA, $(\delta, \sigma)$-RL-EDA, $(\sigma, \delta)$-RL-EDA, $(\sigma, \sigma)$-RL-EDA and Learned-$\sigma$-RL-EDA. The complete hyperparameter configuration of these variants, which serves as a baseline for all experiments, is provided in Appendix K. It includes both EDA-specific and GRPO-related parameters, as well as implementation and execution details ensuring reproducibility, along with information on the computational complexity and runtime of the proposed approach. Here, we restrict this comparison to instances of the pseudo-boolean maximization problem NK with $n = 256$ and $K = 4$ (moderate ruggedness). The results presented are representative of those obtained on other instance distribution.

Figure 2a displays the evolution of average scores over 100 independent runs for the different variants (solid lines). The shaded regions around curves represent one standard deviation above and below the mean across runs. In Figure 2b, the solid lines show the evolution of the mean Hamming distance between the individuals in the population and the best solution found during the trajectory. The shaded regions indicate the standard deviation of the Hamming distance within the population at each generation (one standard deviation above and below the mean). The reported curves are averaged over the 100 independent runs.

The different multivariate variants of our EDA exhibit markedly different behavioral dynamics, despite sharing the same hyperparameters, with the only difference being the sampling distributions over orders. This highlights the critical role of these distributions during both the sampling and update phases of the multivariate RL algorithm.

The version $(\sigma, \sigma)$-RL-EDA that uses both uniform distributions of orders for sampling and training converges towards the best scores (green curve). Once the maximum is reached, we see in Figure 2b that the algorithm has converged because the average distance from the best solution encountered on the trajectory is close to 0. The comparison of this green curve with the blue curve of the $(\delta, \sigma)$-RL-EDA version highlights the contribution of sampling new orders during the EDA generation phase, because it allows maintaining a better diversity of the individuals of the population at each generation and thus allows a better exploration of the search space. It works like an ensembling method where actually different models are used at each generation to produce new solutions. But the main impact is explained when comparing the green curve with the yellow curve of the $(\sigma, \delta)$-RL-EDA version. It highlights the contribution of sampling new orders during the EDA training phase, which underscores the importance of the specific structural dropout at the input of each network induced by this random sampling of orders.

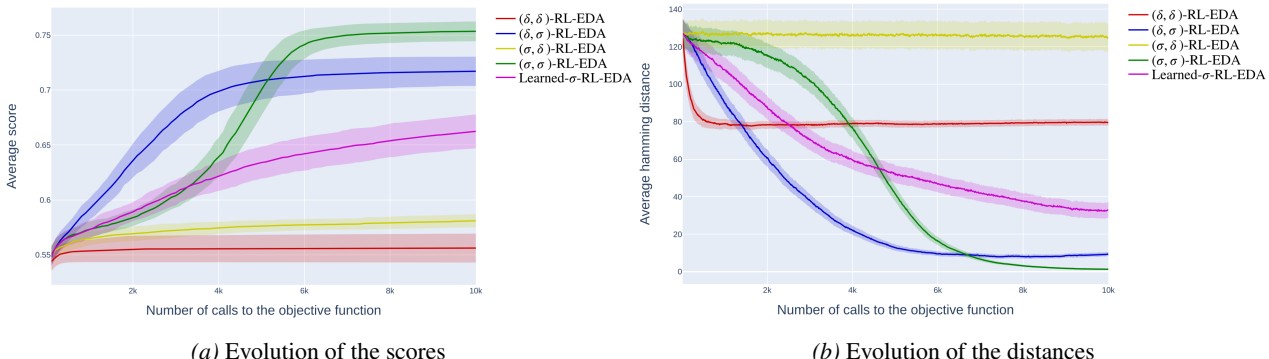

*(a)* Evolution of the scores           *(b)* Evolution of the distances

*Figure 2.* X-axis: number of calls to the objective function. Y-axis: Evolution of average scores (a) and average distances (b) obtained by the different variants of multivariate RL EDA for 100 independent runs on instances of the NK problem with $n = 256$ and $K = 4$.

Finally, the purple curves correspond to the version using a learned Plackett-Luce distribution of order. They also show a good evolution of the scores, but the model did not converge with the allocated budget, and the scores are worse than those obtained by the $(\sigma, \sigma)$-RL-EDA version (green curve). This experiment confirms that attempting to extract explicit structures in such an online search process is not beneficial when using neural estimators, since learning them is at least as hard as learning neural weights from random orderings. Instead, random resampling of new orderings for both generation and training plays a key role in discovering high-quality solutions, as it promotes exploration and enables a more effective identification of interactions between variables.

### 4.2. Validation on Discrete Black-Box Benchmarks

We evaluate the performance of our best-performing variant, $(\sigma, \sigma)$-RL-EDA identified in the previous section, against a comprehensive set of 503 algorithms, essentially composed of those available in the `Nevergrad` library (Rapin & Teytaud, 2018). In version 1.0.12 of `Nevergrad`, a total of 542 algorithms were available. We evaluated all of them on the discrete black-box problems QUBO, NK and NK3, with a time budget of one hour per instance. Among these, 500 algorithms successfully produced solutions within the given time limit for the pseudo-Boolean problems and 496 for the categorical NK3 problem. This panel includes classical metaheuristic for black-box optimization (evolutionary and memetic algorithms), as well as combinations of solving techniques driven by machine learning (e.g., adaptive portfolios). A complete description is provided in Appendix T. In addition to the algorithms available in `Nevergrad`, we include three well-known EDAs: `PBIL` (Baluja, 1994), `MIMIC` (De Bonet et al., 1996), and `BOA` (Pelikan, 2002).[3]

---
[3]For these three algorithms, we rely on the publicly available implementation at https://github.com/e5120/EDAs, using the default hyperparameter settings.

A detailed presentation of the experimental results can be found in Appendix L. As shown in Table 4 of Appendix L, $(\sigma, \sigma)$-RL-EDA frequently obtains the best performance on larger instances ($n = 128$ and $n = 256$) across the various problems considered in this work. Notably, $(\sigma, \sigma)$-RL-EDA performs well on pseudo-Boolean problems QUBO and NK, across a wide range of fitness landscape types—from smooth landscapes (e.g., NK with $K = 1$) to more rugged ones ($K = 8$)—without requiring any change to its hyperparameters, which is noteworthy. As an illustration, Figure 3 shows the evolution of the best scores (averaged over 100 runs) as a function of the number of objective function evaluations, for QUBO instances of size $n = 128$ and type $K = 5$, and NK instances of size $n = 128$ and type $K = 4$. In this figure, $(\sigma, \sigma)$-RL-EDA (green curve) is compared to the 10 competing algorithms that achieved the best results for this instance distribution among the 503 algorithms considered. We observe that our approach achieves the best performance after 10,000 objective function calls. However, it may require more evaluations to converge than competing methods and is therefore outperformed when considering shorter budgets (e.g., 1,000 evaluations). This behavior stems from the fact that $(\sigma, \sigma)$-RL-EDA maintains diversity in the sampled population, which helps prevent premature convergence to low-quality local optimum. A curriculum-based adaptation designed to accelerate convergence under limited evaluation budgets is described in Appendix P. Furthermore, extending $(\sigma, \sigma)$-RL-EDA to ternary variables (NK3 instances) also yields strong performance using the same hyperparameter configuration. Appendix M provides ablation studies and variant analyses that identify the key components contributing to the effectiveness of $(\sigma, \sigma)$-RL-EDA, including comparisons with input dropout techniques. Appendix R reports results on large-scale instances ($n = 1024$), showing that competitive performance can be maintained by adapting the algorithm with parameter sharing between the generators of the $n$ variables.

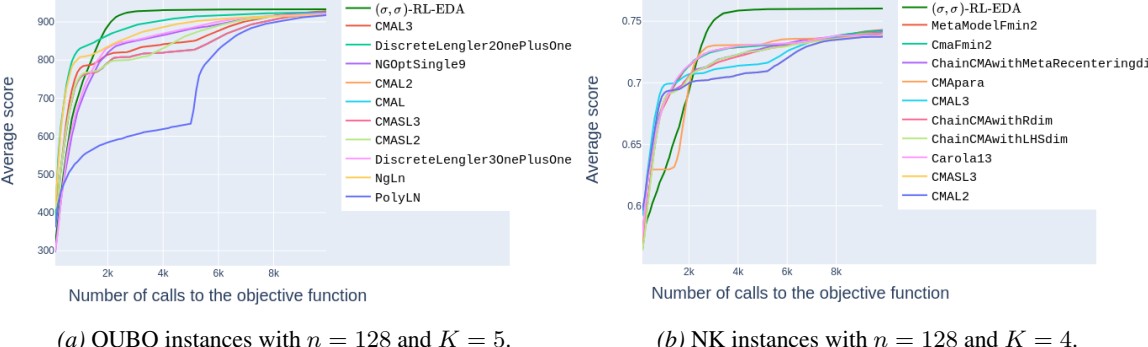

*(a)* QUBO instances with $n = 128$ and $K = 5$.  *(b)* NK instances with $n = 128$ and $K = 4$.

*Figure 3.* X-axis: number of calls to the objective function. Y-axis: Evolution of average scores.

We also compared our method with the same competitors on the real neural architecture search public dataset with binary and categorical variables (NAS-Bench-101) (Ying et al., 2019) (see Appendix O for more details). Our method, with the same hyperparameter configuration as used for the other benchmarks, achieves the best results for small budgets after 1,000 evaluations, but also for large budgets after 10,000 calls to the objective function as displayed in Figure 4.

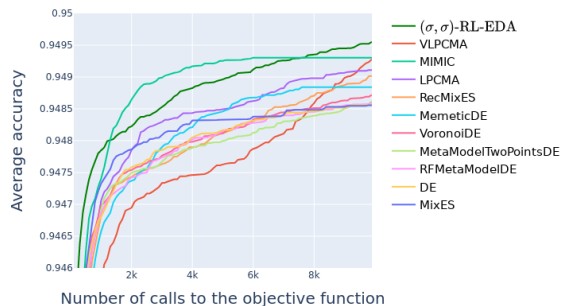

*Figure 4.* NAS-Bench-101 benchmark. X-axis: number of calls to the objective function. Y-axis: Evolution of the average accuracy of the architectures.

## 5. Conclusion

In this work, we introduce a novel discrete black-box optimization framework that leverages neural generators of candidate solutions. The model is trained using an original order-invariant reinforcement learning procedure, enhancing sample efficiency. The robustness of our method is supported by extensive empirical evaluation across a diverse set of black-box optimization problems of varying sizes. In particular, the good results obtained in a real-world application demonstrate its usefulness for practical applications. As future work, we aim to extend this approach to a multi-modal setting, for instance by employing mixtures of distributions, potentially represented through models with attraction–repulsion dynamics.

## Impact Statement

This paper presents work whose goal is to advance the field of machine learning. The code is publicly available. It can help automate and accelerate the calibration of algorithm hyperparameters.

## Acknowledgments

The authors acknowledge ANR – FRANCE (French National Research Agency) for its financial support of the COMBO project (PRC - AAPG 2023 - Axe E.2 - CE23). This work was granted access to the HPC resources of IDRIS (Grant No. AD010611887R3 and AD011014032R3) from GENCI. Some computations were also performed using the computer clusters and data storage resources of the GLiCID (Groupement Ligérien pour le calcul Intensif Distribué, www.glicid.fr).

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

# A. Related Methods for Solving Black-Box Combinatorial Problems

In this appendix, we provide a brief overview of the two principal paradigms that have been developed in the literature for addressing black-box optimization problems: (i) Bayesian optimization (BO) and surrogate-based modeling, and (ii) evolutionary algorithms (EA). We then focus more specifically on Estimation of Distribution Algorithms (EDAs), a subclass of evolutionary algorithms that iteratively use and update a generative model of promising solutions throughout the search process.

**Bayesian Optimization:** The core idea is to treat the unknown objective $f$ as a random function and place a prior over it, typically using a Gaussian Process (GP). As new evaluations are performed, this prior is updated to form a posterior distribution. The acquisition function—e.g., Expected Improvement (EI), Upper Confidence Bound (UCB), or Probability of Improvement (PI)—guides the search by quantifying the utility of evaluating new candidate solutions (Jones et al., 1998; Srinivas et al., 2012). BO is particularly effective for global optimization under tight evaluation budgets, making it well-suited for expensive black-box problems (Forrester & Keane, 2009; Frazier, 2018; Shahriari et al., 2015). **Limitations :** BO often struggles to scale effectively in high-dimensional discrete domains, particularly when GPs are used as surrogates, due to their computational complexity and modeling assumptions, even if recent advances have extended Bayesian optimization to discrete and structured domains through various adaptations: tree-structured models (Bergstra et al., 2011), relaxations of discrete variables into continuous spaces (Kandasamy et al., 2018), and surrogate models better suited to categorical or ordinal data with the use of Random Forests (Bergstra et al., 2011) instead of GP. Moreover, these methods are generally based on strong assumptions about the nature of the noise that may appear in the evaluation of the objective function, such as homoscedastic Gaussian noise, which may not hold in real-world settings, thereby compromising the robustness and reliability of the surrogate model (Wang et al., 2023). Another limitation stems from the inherently sequential nature of classical Bayesian optimization, where only one candidate point is evaluated at each iteration. This design can lead to inefficiencies in scenarios where parallel computational resources are available. Although various batch and parallel extensions have been proposed, such as parallel GP-UCB (Contal et al., 2013; González et al., 2016), these approaches often introduce additional computational overhead and require centralized coordination, which can hinder scalability and responsiveness in practical applications.

**Evolutionary Algorithms:** Metaheuristic approaches (local search, population-based algorithms...) are widely used to solve CO problems, and EAs offer several appealing characteristics. Because they avoid the overhead of building and updating surrogate models, the computational cost per iteration is typically low. EAs also demonstrate robustness to noise, as selection is often based on the ranking of individuals rather than absolute fitness values, making them resilient to stochastic perturbations and invariant under monotonic transformations of the objective. Theoretical convergence results are available for certain classes of EAs, supported by advances in runtime analysis and black-box complexity theory (Auger & Doerr, 2011; Doerr et al., 2019). **Limitations :** EAs may require more function evaluations to identify high-quality solutions compared to model-based approaches for complex problems, which can limit their sample efficiency. Some research, however, has shown that hybrid approaches—combining EAs with surrogate modeling or adaptive sampling strategies—can significantly enhance their effectiveness in scenarios with expensive evaluations (Emmerich et al., 2006; Jin, 2011).

**Estimation of Distribution Algorithms:** Like EAs, EDAs rely on population-based search, but they inherit from BO the notion of modeling structure in the search space, although their modeling goal differs. Instead of modeling the entire objective function, EDAs aim to model only the distribution of promising regions in the fitness landscape, thus avoiding the complexity of full surrogate modeling. This makes EDAs more computationally scalable in high-dimensional or discrete spaces, where standard Gaussian Process-based BO may struggle due to assumptions of smoothness, stationarity, or computational costs of inference (Frazier, 2018; Shan & Wang, 2010). The learning process in EDAs may be as simple as estimating independent univariate marginals, as in the Univariate Marginal Distribution Algorithm (UMDA) (Mühlenbein & Paaß, 1996), or as sophisticated as constructing full probabilistic graphical models, such as in the Bayesian Optimization Algorithm (BOA) (Pelikan, 2002). EDAs still benefit from recent developments (Uribe et al., 2022) that open new possible application domains, for instance, to achieve machine learning tasks (Larrañaga & Bielza, 2024). One of the principal advantages of the modeling strategy of EDAs is its ability to capture variable interactions, an essential feature in epistatic or non-separable problems, where standard EAs often fail. Several EDAs use graph structure (DAG) extraction at each generation of the process. The MIMIC algorithm (De Bonet et al., 1996) proposes constructing a first-order Markov chain on the variables, classifying them greedily using pairwise mutual information to capture their strongest statistical dependencies. The Bayesian Optimization Algorithm (BOA) (Pelikan, 2002) introduces a more expressive probabilistic model using Bayesian networks, allowing it to represent complex, higher-order interactions between variables. The Factorized Distribution Algorithm (FDA) (Lozano, 2006; Mühlenbein & Paaß, 1996) exploits prior knowledge about the

structure of the problem by explicitly incorporating domain-specific decompositions through a predefined factorization of the joint distribution. However, while these approaches can perform well on certain problems, they are fundamentally limited by the exponential growth of computational cost as problem size and dependency complexity increase. In particular, BOA-based methods not only face prohibitive model-construction costs in high-dimensional settings (Hauschild & Pelikan, 2011), but the complexity of learning accurate dependency structures can also hinder effective exploration of the search space. **Limitations :** EDAs exhibit some limitations in terms of premature convergence. Since most EDAs update their probabilistic model solely from the current population, they tend to focus the search around a single promising region, potentially losing diversity and missing other basins of attraction (Hauschild & Pelikan, 2011). To address these limitations, several diversity-preserving or niching-based EDAs have been proposed. For example, the `Multi-CMA-ES` algorithm introduces multiple co-evolving models that repel each other in the search space to maintain diversity and explore multiple optima (Karunarathne et al., 2024). Similar ideas are found in multi-population EDAs or speciation-based approaches (Yang et al., 2016).

A natural limitation is the choice of the distribution model. In the continuous case (i.e. $\mathcal{X} \subseteq \mathbb{R}^n$), a common choice is the multivariate Gaussian distribution, which encodes dependencies via its covariance matrix (e.g. `CMA-ES` (Hansen & Ostermeier, 2001)). In the discrete setting considered here, there is however no direct analogue of the Gaussian. Rather, one instead typically uses probabilistic graphical models, such as Bayesian networks (Echegoyen et al., 2008) or undirected graphical models / Markov networks (e.g. as in `DEUM` (Shakya, 2006)), which model joint dependencies via conditional probability tables or undirected cliques and permit sampling of new candidate vectors. Research on multivariate discrete EDAs has seen a notable decline in recent years because there is no equivalent of the multivariate Gaussian distribution for the discrete space. However, Benhamou et al. (2018) attempts to adapt the `CMA-ES` algorithm to the discrete case, using a multivariate Bernoulli distribution.

## B. Derivation of the PPO Update (2)

While the derivation of (2) is rather straightforward, following the proofs in (Schulman et al., 2015a), we detail here its adaptation to our notations and to our undiscounted setting, where only final rewards are considered, for completeness.

Let us first introduce some classical quantities in reinforcement learning:

- $V^\pi(s)$ is the state value function, which returns the expected cumulative return following policy $\pi$ from state $s$. In our setting, this can be defined for any given order $\sigma$ and any given state $s = (x_{\sigma_{<k}}, k-1, \sigma)$, as:

$$V^\pi(s) = V^{\pi,\sigma}(x_{\sigma_{<k}}) = \mathbb{E}_{\pi_\theta(x_{\sigma_{\geq k}}|x_{\sigma_{<k}},\sigma)} [f(x)]$$

- $Q^\pi(s,a)$ is the state-action value function, which returns the expected cumulative return from state $s$, assuming that the first action in $s$ is $a$ and then subsequent actions are sampled from $\pi$. In our setting, this can be defined for any given order $\sigma$ and any given state $s = (x_{\sigma_{<k}}, k-1, \sigma)$, and any action $a = x_{\sigma_k}$ that specifies the value for $X_{\sigma_k}$, as:

$$Q^\pi(s,a) = Q^{\pi,\sigma}(x_{\sigma_{<k}}, x_{\sigma_k}) = \mathbb{E}_{\pi_\theta(x_{\sigma_{>k}}|x_{\sigma_{\leq k}},\sigma)} [f(x)]$$

- $A^\pi(s,a)$ is the advantage function, defined as:

$$A^\pi(s,a) = A^{\pi,\sigma}(x_{\sigma_{<k}}, x_{\sigma_k}) = Q^{\pi,\sigma}(x_{\sigma_{<k}}, x_{\sigma_k}) - V^{\pi,\sigma}(x_{\sigma_{<k}})$$

We are interested in maximizing $J^\sigma(\theta) = \mathbb{E}_{\pi_\theta(x|\sigma)}[f(x)]$, while reusing samples from a previous policy to improve sample efficiency and stability.

We start by observing that, given any two policies $\pi_\theta$ and $\pi_{\theta'}$, we have:

$$\arg\max_\theta J^\sigma(\theta) = \arg\max_\theta J^\sigma(\theta) - J^\sigma(\theta'),$$

since $\theta$ does not appear in $J^\sigma(\theta')$.

Looking at $J^\sigma(\theta) - J^\sigma(\theta')$, we get:

$$
\begin{aligned}
J^\sigma(\theta) - J^\sigma(\theta') &= \mathbb{E}_{\pi_\theta(x|\sigma)}[f(x)] - \mathbb{E}_{\pi_{\theta'}(x|\sigma)}[f(x)] & (9)\\
&= \mathbb{E}_{\pi_\theta(x|\sigma)}[f(x)] - V^{\pi_{\theta'},\sigma}(\varnothing) & (10)\\
&= \mathbb{E}_{\pi_\theta(x|\sigma)}\left[f(x) - V^{\pi_{\theta'},\sigma}(\varnothing)\right] & (11)\\
&= \mathbb{E}_{\pi_\theta(x|\sigma)}\left[V^{\pi_{\theta'},\sigma}(x_{\sigma_{\le n}}) - V^{\pi_{\theta'},\sigma}(\varnothing)\right] & (12)\\
&= \mathbb{E}_{\pi_\theta(x|\sigma)}\left[\sum_{k=1}^n \left(V^{\pi_{\theta'},\sigma}(x_{\sigma_{<k+1}}) - V^{\pi_{\theta'},\sigma}(x_{\sigma_{<k}})\right)\right] & (13)\\
&= \mathbb{E}_{\pi_\theta(x|\sigma)}\left[\sum_{k=1}^n \left(Q^{\pi_{\theta'},\sigma}(x_{\sigma_{<k}}, x_{\sigma_k}) - V^{\pi_{\theta'},\sigma}(x_{\sigma_{<k}})\right)\right] & (14)\\
&= \mathbb{E}_{\pi_\theta(x|\sigma)}\left[\sum_{k=1}^n A^{\pi_{\theta'},\sigma}(x_{\sigma_{<k}}, x_{\sigma_k})\right] & (15)\\
&= \mathbb{E}_{\pi_\theta(x|\sigma)}\sum_{k=1}^n \mathbb{E}_{\pi_\theta(x_{\sigma_k}|x_{\sigma_{<k}},\sigma)}\left[A^{\pi_{\theta'},\sigma}(x_{\sigma_{<k}}, x_{\sigma_k})\right] & (16)\\
&= \mathbb{E}_{\pi_\theta(x|\sigma)}\sum_{k=1}^n \mathbb{E}_{\pi_{\theta'}(x_{\sigma_k}|x_{\sigma_{<k}},\sigma)}\frac{\pi_\theta(x_{\sigma_k}|x_{\sigma_{<k}},\sigma)}{\pi_{\theta'}(x_{\sigma_k}|x_{\sigma_{<k}},\sigma)}\left[A^{\pi_{\theta'},\sigma}(x_{\sigma_{<k}}, x_{\sigma_k})\right] & (17)
\end{aligned}
$$

where $\varnothing$ is the empty sequence (which can also be denoted as the starting point of any sequence $x_{\sigma_{<1}}$). This derivation leverages the fact that in our case, for any sequence $x$ and any policy $\pi$, $f(x) = V^{\pi,\sigma}(x_{\sigma_{\le n}})$ as the sequence is already completed after $n$ steps (we are in a terminal state, as $n$ is the dimension of our combinatorial space $\mathcal{X}$). Also, (13) exploits the fact that every term of the sum telescopes except for the two extrema that appear in (12), (14) leverages that, following definitions above, for any $x$ and any $0 < k \le n$, we have: $Q^{\pi_{\theta'},\sigma}(x_{\sigma_{<k}}, x_{\sigma_k}) = V^{\pi_{\theta'},\sigma}(x_{\sigma_{<k+1}})$.

Next, if $\pi_\theta(x|\sigma)$ is sufficiently close to $\pi'_\theta(x|\sigma)$, the idea of TRPO/PPO based approaches is to rather use samples of states from the old policy $\pi_{\theta^t}(x|\sigma)$, rather than the current one. This is done in our case by replacing $\mathbb{E}_{\pi_\theta(x|\sigma)}$ by $\mathbb{E}_{\pi_{\theta^t}(x|\sigma)}$ in (17). We obtain:

$$
J^\sigma(\theta) - J^\sigma(\theta^t) \approx L_{\theta^t}^\sigma(\theta) \tag{18}
$$

with

$$
\begin{aligned}
L_{\theta^t}^\sigma(\theta) &\triangleq \mathbb{E}_{\pi_{\theta^t}(x|\sigma)}\sum_{k=1}^n \mathbb{E}_{\pi_{\theta^t}(x_{\sigma_k}|x_{\sigma_{<k}},\sigma)}\frac{\pi_\theta(x_{\sigma_k}|x_{\sigma_{<k}},\sigma)}{\pi_{\theta^t}(x_{\sigma_k}|x_{\sigma_{<k}},\sigma)}\left[A^{\pi_{\theta^t},\sigma}(x_{\sigma_{<k}}, x_{\sigma_k})\right] & (19)\\
&= \mathbb{E}_{\pi_{\theta^t}(x|\sigma)}\sum_{k=1}^n \frac{\pi_\theta(x_{\sigma_k}|x_{\sigma_{<k}},\sigma)}{\pi_{\theta^t}(x_{\sigma_k}|x_{\sigma_{<k}},\sigma)}\left[A^{\pi_{\theta^t},\sigma}(x_{\sigma_{<k}}, x_{\sigma_k})\right] & (20)
\end{aligned}
$$

Next, we consider $\nabla_\theta L_{\theta^t}^\sigma(\theta)$ as a proxy for $\nabla_\theta(J^\sigma(\theta) - J^\sigma(\theta^t)) = \nabla_\theta J^\sigma(\theta)$, which results in (2).

## C. Derivation of the PPO Update with Varying Generation/Training Orders

In this section, we check that PPO updates, that we derived in the previous section for the case of an arbitrary fixed generation (and training) order, can be adapted to the case of varying permutations.

For the case where the training order is always the same as the generation one (i.e., $\xi(\cdot|\sigma)$ is a Dirac distribution centered on $\sigma$), the derivation of the PPO update is trivial to obtain from (20), as it suffices to take the expectation of $L_{\theta^t}^\sigma(\theta)$ depending on distribution $\xi(\cdot)$. The update can be derived by taking the gradient of $L_{\theta^t}(\theta) = \mathbb{E}_{\sigma \sim \xi(\sigma)} L_{\theta^t}^\sigma(\theta)$.

Next, we consider the more involved case, where generation and training orders can be different. For this purpose, looking

at $J^\sigma(\theta) - J^{\sigma'}(\theta')$, we get:

$$J^\sigma(\theta) - J^{\sigma'}(\theta') = \mathbb{E}_{\pi_\theta(x|\sigma)}[f(x)] - \mathbb{E}_{\pi_{\theta'}(x|\sigma')}[f(x)] \tag{21}$$

$$= \mathbb{E}_{\pi_\theta(x|\sigma)}[f(x)] - V^{\pi_{\theta'},\sigma'}(\varnothing) \tag{22}$$

$$= \mathbb{E}_{\pi_\theta(x|\sigma)}\left[f(x) - V^{\pi_{\theta'},\sigma'}(\varnothing)\right] \tag{23}$$

$$= \mathbb{E}_{\pi_\theta(x|\sigma)}\left[V^{\pi_{\theta'},\sigma'}(\sigma'(x)_{\leq dim_{\sigma'}(n)}) - V^{\pi_{\theta'},\sigma'}(\sigma'(x)_{<dim_{\sigma'}(1)})\right] \tag{24}$$

$$= \mathbb{E}_{\pi_\theta(x|\sigma)}\left[\sum_{k=1}^n \left(V^{\pi_{\theta'},\sigma'}(\sigma'(x)_{<k+1}) - V^{\pi_{\theta'},\sigma'}(\sigma'(x)_{<k})\right)\right] \tag{25}$$

$$= \mathbb{E}_{\pi_\theta(x|\sigma)}\left[\sum_{k=1}^n \left(Q^{\pi_{\theta'},\sigma'}(\sigma'(x)_{<k}, x_k) - V^{\pi_{\theta'},\sigma'}(\sigma'(x)_{<k})\right)\right] \tag{26}$$

$$= \mathbb{E}_{\pi_\theta(x|\sigma)}\left[\sum_{k=1}^n A^{\pi_{\theta'},\sigma'}(\sigma'(x)_{<k}, x_k)\right] \tag{27}$$

$$= \mathbb{E}_{\pi_\theta(x|\sigma)} \sum_{k=1}^n \mathbb{E}_{\pi_\theta(x_k|\sigma(x)_{<k},\sigma)}\left[A^{\pi_{\theta'},\sigma'}(\sigma'(x)_{<k}, x_k)\right] \tag{28}$$

$$= \mathbb{E}_{\pi_\theta(x|\sigma)} \sum_{k=1}^n \mathbb{E}_{\pi_{\theta'}(x_k|\sigma'(x)_{<k},\sigma')}\left[\frac{\pi_\theta(x_k|\sigma(x)_{<k},\sigma)}{\pi_{\theta'}(x_k|\sigma'(x)_{<k},\sigma')}A^{\pi_{\theta'},\sigma'}(\sigma'(x)_{<k}, x_k)\right] \tag{29}$$

where we switched to the notation introduced in section 3.3, that is more convenient for dealing with different orders $\sigma$ and $\sigma'$. In particular, this makes that the inner sum from $k = 1$ to $n$ enumerates index from the original problem in $\mathcal{X}$, rather than the generation order from a given permutation. This has an impact on the ordering of advantages functions in (28), but the quantities still telescope, and each advantage is aligned with the trained transition in (29). We note that importance sampling ratios do not exploit the same information, as masks do not apply on same dimensions in the numerator and denominator, but the behavior distribution is still nonzero everywhere the training distribution allocates probability mass, which is the main requirement for importance sampling techniques.

Then, given a previous behavior policy $\pi_{\theta^t}$ that sampled solutions with generation order $\sigma'$, we can train policy $\pi_\theta$, with training order $\sigma$, by considering the following approximator:

$$L_{\theta^t}^{\sigma,\sigma'}(\theta) \triangleq \mathbb{E}_{\pi_{\theta^t}(x|\sigma')} \sum_{k=1}^n \mathbb{E}_{\pi_{\theta^t}(x_k|\sigma'(x)_{<k},\sigma')}\frac{\pi_\theta(x_k|\sigma(x)_{<k},\sigma)}{\pi_{\theta^t}(x_k|\sigma'(x)_{<k},\sigma')}\left[A^{\pi_{\theta^t},\sigma'}(\sigma'(x)_{<k}, x_k)\right]$$

$$= \mathbb{E}_{\pi_{\theta^t}(x|\sigma')} \sum_{k=1}^n \frac{\pi_\theta(x_k|\sigma(x)_{<k},\sigma)}{\pi_{\theta^t}(x_k|\sigma'(x)_{<k},\sigma')}\left[A^{\pi_{\theta^t},\sigma'}(\sigma'(x)_{<k}, x_k)\right] \tag{30}$$

For any $((\pi_{\theta^t}, \sigma'), (\pi_\theta, \sigma))$, we have that: $J^\sigma(\theta) - J^{\sigma'}(\theta^t) \approx L_{\theta^t}^{\sigma,\sigma'}(\theta)$ whenever $\pi_\theta^t(\cdot|\sigma')$ remains close to $\pi_\theta(\cdot|\sigma)$.

Finally, we can take $\mathbb{E}_{\sigma\sim\xi(\sigma),\sigma'\sim\xi(\sigma'|\sigma)}L_{\theta^t}^{\sigma',\sigma}(\theta)$ as the maximization objective, with KL regularization constraints that are considered in (8).

## D. On the Convergence in the Infinite Data and Infinite Capacity Regime

In our approach, we consider at each step of our process the maximization of the quantity (see section 3.3):

$$\hat{L}_\lambda^t(\theta) = \frac{1}{\lambda} \sum_{(x^i,\sigma^i)\in\Gamma_\lambda^t} \mathbb{E}_{\sigma'\sim\xi(\sigma'|\sigma^i)} \sum_{k=1}^n \left[\frac{\pi_\theta(x_k^i|\sigma'(x^i)_{<k})}{\pi_{\theta^t}(x_k^i|\sigma^i(x^i)_{<k})}A_{\Gamma_\lambda^t}\left(x^{(i)}\right)\right.$$

$$\left. -\beta D_{\mathrm{KL}}\left(\pi_{\theta^t}(\cdot|\sigma^i(x^i)_{<k}) \,\|\, \pi_\theta(\cdot|\sigma'(x^i)_{<k})\right)\right]. \tag{31}$$

where $\Gamma_\lambda^t = \{x^{(1)}, \ldots, x^{(\lambda)}\}$ is a set of i.i.d. samples from $\pi_{\theta^t}$, and where $A_{\Gamma_\lambda^t}\left(x^{(i)}\right)$ is a ranking function of $x_i$ in the set $\Gamma_\lambda^t$ in decreasing order of fitness.

For simplicity of notation, we rewrite this quantity as:

$$\hat{L}_\lambda^t(\theta) = \frac{1}{\lambda} \sum_{i=1}^{\lambda} w_{\theta^t,\theta}\big(x^{(i)}, \sigma^{(i)}\big) A_{\Gamma_\lambda^t}\big(x^{(i)}\big) + kl_{\theta^t,\theta}\big(x^{(i)}, \sigma^{(i)}\big),$$

where:

- $w_{\theta^t,\theta}\big(x^{(i)}, \sigma^{(i)}\big) = \mathbb{E}_{\sigma' \sim \xi(\sigma'|\sigma^i)} \sum_{k=1}^{n} \left[ \frac{\pi_\theta(x_k^i | \sigma'(x^i)_{<k})}{\pi_{\theta^t}(x_k^i | \sigma^i(x^i)_{<k})} \right]$

- $kl_{\theta^t,\theta}\big(x^{(i)}, \sigma^{(i)}\big) = -\beta \mathbb{E}_{\sigma' \sim \xi(\sigma'|\sigma^i)} \sum_{k=1}^{n} \left[ D_{\mathrm{KL}}\big(\pi_{\theta^t}(\cdot | \sigma^i(x^i)_{<k}) \,\|\, \pi_\theta(\cdot | \sigma'(x^i)_{<k})\big) \right]$

We first show the following lemma, that states that $\hat{L}_\lambda^t(\theta)$ is an unbiased estimator of:

$$L_\lambda^t(\theta) = \mathbb{E}_\sigma \mathbb{E}_{x \sim \pi_{\theta^t}(\cdot|\sigma)} \left[ w_{\theta^t,\theta}\big(x, \sigma\big) \, \mathbb{E}_{\Gamma_\lambda^t \setminus \{x\}} \big[ A_{\Gamma_\lambda^t}(x) \big] + kl_{\theta^t,\theta}\big(x, \sigma\big) \right], \tag{32}$$

where $\mathbb{E}_{\Gamma_\lambda^t \setminus \{x\}} \big[ A_{\Gamma_\lambda^t}(x) \big]$ denotes the expectation of the ranking of $x$ in a set containing $\lambda - 1$ other samples from the mixture $\mathbb{E}_\sigma \pi_{\theta^t}(\cdot|\sigma)$:

**Lemma D.1.** $\mathbb{E}\big[\hat{L}_\lambda^t(\theta)\big] = \mathbb{E}_\sigma \mathbb{E}_{x \sim \pi_{\theta^t}(\cdot|\sigma)} \left[ w_{\theta^t,\theta}\big(x, \sigma\big) \, \mathbb{E}_{\Gamma_\lambda^t \setminus \{x\}} \big[ A_{\Gamma_\lambda^t}(x) \big] + kl_{\theta^t,\theta}\big(x, \sigma\big) \right]$

*Proof.* By the linearity of expectation, we have:

$$\mathbb{E}\big[\hat{L}_\lambda^t(\theta)\big] = \frac{1}{\lambda} \sum_{i=1}^{\lambda} \mathbb{E}\Big[ w_{\theta^t,\theta}\big(x^{(i)}, \sigma^{(i)}\big) A_{\Gamma_\lambda^t}\big(x^{(i)}\big) + kl_{\theta^t,\theta}\big(x^{(i)}, \sigma^{(i)}\big) \Big].$$

Then, as all $x^{(i)}$ are i.i.d., each component of the sum has the same expectation. Thus, by exchangeability, we can say that (arbitrarily taking the first sample $(x^{(1)}, \sigma^{(1)})$ from $\Gamma_\lambda^t$ as the reference, without loss of generality):

$$\mathbb{E}\big[\hat{L}_\lambda^t(\theta)\big] = \mathbb{E}\Big[ w_{\theta^t,\theta}\big(x^{(1)}, \sigma^{(1)}\big) A_{\Gamma_\lambda^t}\big(x^{(1)}\big) + kl_{\theta^t,\theta}\big(x^{(1)}, \sigma^{(1)}\big) \Big].$$

Using the law of total expectation, we obtain:

$$\mathbb{E}\Big[ w_{\theta^t,\theta}\big(x^{(1)}, \sigma^{(1)}\big) A_{\Gamma_\lambda^t}\big(x^{(1)}\big) + kl_{\theta^t,\theta}\big(x^{(1)}, \sigma^{(1)}\big) \Big] =$$
$$\mathbb{E}_{\sigma^{(1)}} \mathbb{E}_{x^{(1)} \sim \pi_{\theta^t}(\cdot|\sigma^{(1)})} \Big[ w_{\theta^t,\theta}\big(x^{(1)}, \sigma^{(1)}\big) \mathbb{E}\Big[ A_{\Gamma_\lambda^t}\big(x^{(1)}\big) \mid x^{(1)} \Big] + kl_{\theta^t,\theta}\big(x^{(1)}, \sigma^{(1)}\big) \Big].$$

Fixing $x^{(1)} = x$ corresponds to considering $x$ as one element of the set $\Gamma_\lambda^t$, and completing it with $\lambda - 1$ additional independent draws. Therefore, we have:

$$\mathbb{E}\big[ A_{\Gamma_\lambda^t}(x^{(1)}) \mid x^{(1)} = x \big] = \mathbb{E}_{\Gamma_\lambda^t \setminus \{x\}} \big[ A_{\Gamma_\lambda^t}(x) \big].$$

Thus, we finally get:

$$\mathbb{E}\big[\hat{L}_\lambda^t(\theta)\big] = \mathbb{E}_\sigma \mathbb{E}_{x \sim \pi_{\theta^t}(\cdot|\sigma)} \left[ w_{\theta^t,\theta}\big(x, \sigma\big) \, \mathbb{E}_{\Gamma_\lambda^t \setminus \{x\}} \big[ A_{\Gamma_\lambda^t}(x) \big] + kl_{\theta^t,\theta}\big(x, \sigma\big) \right],$$

which concludes the proof and indicates that $\hat{L}_\lambda^t(\theta)$ is an unbiased estimator of $L_\lambda^t(\theta)$. □

Thus, while at each epoch $t$ our algorithm seeks to maximize the stochastic estimator $\hat{L}_\lambda(\theta)$, in expectation it actually aims to optimize the theoretical objective $L_\lambda^t(\theta)$.

Following this, we observe that our surrogate scale-invariant objective $A_{\Gamma_\lambda^t}(x)$ (that we use in (8), in place of the original fitness sore from (7)), can be considered in expectation as a stationary classical reward function at each epoch $t$, depending only on constant parameters $\theta_t$.

We thus obtain a classical learning problem at each epoch $t$, where we maximize

$$\pi_\theta(x_k \mid \sigma'(x)_{<k}) \; \frac{\pi_{\theta^t}(x \mid \sigma)}{\pi_{\theta^t}(x_k \mid \sigma(x)_{<k})} \; \mathbb{E}_{\Gamma_\lambda^t \setminus \{x\}}\left[ A_{\Gamma_\lambda^t}(x) \right],$$

for any uniformly sampled tuple ($x \in \mathcal{X}, \sigma \in \Omega, \sigma' \in \Omega, k \in [\![1, n]\!]$), under the soft constraint imposed by the KL regularizer. In other words, at each epoch the conditional probability of values for dimension $k \in [\![1, n]\!]$ of solutions likely under $\pi_{\theta^t}(x \mid \sigma)$ is increased (resp. decreased) if they have a positive (resp. negative) expected signed rank among $\lambda$ samples from $\mathbb{E}_\sigma \pi_{\theta^t}(\cdot \mid \sigma)$. This means that decisions leading to high (resp. low) fitness are reinforced (resp. penalized) at each epoch. As $t \to \infty$, the distribution $\Gamma_\lambda^t$ converges asymptotically towards a degenerate set containing a single solution. If $\lambda$ is infinite, this limiting solution coincides with the global optimum of the problem (i.e., the element $x^\star \in \mathcal{X}$ such that $f(x^\star) = \max_{x \in \mathcal{X}} f(x)$).

## E. Generation/Training Permutations as Information-Preserving Input Dropout

In section D, we have shown that the quantity we consider in each maximization step is an unbiased estimator of $L_\lambda^t(\theta)$, as defined in (32):

$$L_\lambda^t(\theta) = \mathbb{E}_\sigma \mathbb{E}_{x \sim \pi_{\theta^t}(\cdot \mid \sigma)} \left[ \mathbb{E}_{\sigma' \sim \xi(\sigma' \mid \sigma)} \sum_{k=1}^n \left[ \frac{\pi_\theta(x_k \mid \sigma'(x)_{<k})}{\pi_{\theta^t}(x_k \mid \sigma(x)_{<k})} \; \mathbb{E}_{\Gamma_\lambda^t \setminus \{x\}} \left[ A_{\Gamma_\lambda^t}(x) \right] \right.\right.$$

$$\left.\left. - \beta D_{\mathrm{KL}} \left( \pi_{\theta^t}(\cdot \mid \sigma(x)_{<k}) \,\|\, \pi_\theta(\cdot \mid \sigma'(x)_{<k}) \right) \right] \right], \quad (33)$$

This formulation allows us to distinguish between the two effects of the randomness introduced in the order of generation:

- **Population Diversity**: During the first epochs, the neural generators are not prepared for order invariance. Different generation orders $\sigma$ thus induce different generation distributions $\pi(\cdot \mid \sigma)$. Uniformly sampling a new $\sigma$ from $\Omega$ for each generation therefore results in higher diversity in the populations. In that case, any estimation of the reward metric $\mathbb{E}_{\Gamma_\lambda^t \setminus \{x\}} \left[ A_{\Gamma_\lambda^t}(x) \right]$ is thus likely to have a greater variance than when using a fixed generation order (especially for low $\lambda$), as the variance of a mixture of distributions (i.e., $\mathbb{E}_\sigma \pi_{\theta^t}(\cdot, \sigma)$) is always greater or equal than the lowest variance of its components. This allows for better exploration in the first steps of the process by introducing more stochasticity in the RL returns. Moreover, this provides more diverse samples to the training process, avoiding early collapse on a particular subarea of the search space;

- **Structural Regularization**: Beyond population diversity, the second effect is a form of structural regularization. This arises from presenting, for the same candidate solution $x$, different contexts at each generation step (i.e., for each neural network $g_{\theta_k}$ in our setting). Even when the training order matches the generation order (i.e., when $\xi(\sigma' \mid \sigma)$ is a Dirac centered at $\sigma$), the process encourages the learning of order-invariant generators. In this case, the IS ratios are all equal to 1 at the start of each PPO epoch (with the KL divergence equal to 0). Nevertheless, since each individual processes dimensions in a different order, the generators are encouraged to structure their weights so as to handle arbitrary subsets of variables of any size, ultimately leading to a residual summation structure (see discussion on that point below). However, simply maintaining the same order for training as the one used for generating the training sample is usually not sufficient to efficiently prepare the generator for order-invariance, since a constant order is applied to each training sample across all iterations of the epoch. The use of a different order for each sample at each iteration of the same epoch (i.e., $\xi(\cdot \mid \sigma)$ is a uniform distribution in our experiments) provides two benefits. First, it rewards the network for making the same decision under varying contexts, thus facilitating the identification of inter-variable dependencies. Second, it steers the network toward producing, for the same decision, distributions similar to the one used for sampling despite changes in context, through the KL regularizer (which is nonzero even at the first iteration in this setting). All of this benefits sample efficiency, while also promoting generation order invariance and stability through inter-order generalization.

**About Residual Structure**   In order to further understand the effect of training order permutations on the structuring of a neural network, consider a simple distribution approximation problem via maximum likelihood estimation (MLE): $\arg\max_\theta \mathbb{E}_p[\log p_\theta(x)]$. Let $x$ be a binary sequence of size $n$, and let $p_\theta(x)$ be parameterized differently (with parameter $\theta_i$) for each dimension of $x$, as in the setting of this paper. We specifically focus on the network corresponding to the last dimension of $x$, i.e., $p_{\theta_n}$.

When optimizing the joint distribution in the original order of the sequence (from dimension 1 to $n$), $p_{\theta_n}$ is always conditioned on all preceding variables, as it predicts the last variable based on the inputs $x_1$ to $x_{n-1}$. Given $\lambda$ samples from $p$ to optimize it via MLE, the gradient updates of $p_{\theta_n}$ are computed as an average over $\lambda$ gradients of the fully informed conditional probability $p_{\theta_n}(x_n \mid x_{<n})$, while some input variables may consist only of noise with respect to the variable being decoded. The optimization process must cope with all these inputs in order to eventually identify true dependencies, despite the presence of potentially significant noise in the input.

Now, let us consider training order permutations $\sigma$, which effectively mask every variable $x_i$ whose rank in $\sigma$ is greater than the rank of $x_n$ (i.e., we set to zero each variable $x_i$ such that $\mathrm{rank}_\sigma(i) > \mathrm{rank}_\sigma(n)$ in the input of $p_{\theta_i}$). The MLE is now given for the variable $x_n$ as:

$$L = \mathbb{E}_p\mathbb{E}_\sigma[\log p_{\theta_n}(x_n|\sigma(x)_{<n})],$$

which, if the distribution of $\sigma$ is uniform, is equivalent to considering:

$$L = \mathbb{E}_p\left[\frac{(n-1)!}{n!}\log p_{\theta_n}(x_n|\varnothing) + \frac{(n-2)!}{n!}\sum_{i\in[[1,n-1]]}\log p_{\theta_n}(x_n|\{x_i\})\right.$$

$$\left. + \frac{2(n-3)!}{n!}\sum_{i\in[[1,n-1]]}\sum_{j\in[[1,n-1]],j\neq i}\log p_{\theta_n}(x_n|\{x_i,x_j\}) + \ldots + \frac{(n-1)!}{n!}\log p_{\theta\_n}(x_n|\{x_i\}_{i=1}^{n-1})\right].$$

or more compactly:

$$L = \mathbb{E}_p\sum_{k=1}^{n}\left[w_k^n\sum_{I_k\in\binom{\{1,\ldots,n-1\}}{k-1}}\log p_{\theta_n}(x_n|\{x_i\}_{i\in I_k})\right],$$

with $w_k^n = \frac{(k-1)!(n-k)!}{n!}$ the weight of a component depending on the size of its condition (i.e., number of available dimensions for decoding $x_n$), which in turn can be rewritten as:

$$L = \mathbb{E}_{p(x_n)}\sum_{k=1}^{n}\sum_{I_k\in\binom{\{1,\ldots,n-1\}}{k-1}}\left[w_k^n\mathbb{E}_{p(\{x_i\}_{i\in I_k}|x_n)}\log p_{\theta_n}(x_n|\{x_i\}_{i\in I_k})\right]$$

From this expansion, we can note a decrease of weights associated with each component of the training problem until $k = n/2$: For any $k < n/2$, $w_{k+1}^n < w_k^n$. This affects the relative learning speed of the corresponding components, simple dependencies are easier to extract. During optimization, the network thus first learns to encode the marginal probability $p_{\theta_n}(x_n \mid \varnothing)$ for $x_n$, then incrementally incorporates potential interactions with single variables through $p_{\theta_n}(x_n \mid \{x_i\})$, then with pairs of variables, and so on. As a result, the network naturally develops a form of residual structuring, where outputs are composed by aggregating contributions from different subsets of inputs.

This hierarchical learning process enables the network to more efficiently identify the parent variables that are relevant to the joint distribution, while simultaneously recognizing variables that are unrelated and contribute only noise to $p_{\theta_n}(x_n \mid \sigma(x)_{<n})$. As a result, the network becomes both more robust and sample-efficient, effectively filtering out irrelevant inputs while capturing the essential dependencies.

**Order Permutations vs Input Dropout**   We note that an alternative to permutations is input dropout, whose principle is to randomly mask any feature from the input during training. Similarly to permutation orders, input dropout can be defined as masks that set certain input variables to 0 (or to a null vector in the categorical setting). Here, we consider a mask $m \in \Omega^m$ as a binary $n \times n$ matrix that removes the entry in dimension $j$ for the decision of dimension $i$ if $m_{i,j} = 1$. We denote by $m(x)_k$ the result of applying the dropout mask $m$ to $x$, using the $k$-th row of the matrix.

As with permutations, we consider a distribution $\xi^m(\cdot)$ for dropout at generation time, and a distribution $\xi^m(\cdot \mid m)$ for dropout at training time. Given this, our objective in (33) can be naturally extended as:

$$L_\lambda^t(\theta) = \mathbb{E}_{\sigma,m} \mathbb{E}_{x \sim \pi_{\theta^t}(\cdot|\sigma,m)} \left[ \mathbb{E}_{\substack{\sigma' \sim \xi(\sigma'|\sigma^i) \\ m' \sim \xi^m(m'|m)}} \sum_{k=1}^n \left[ \frac{\pi_\theta(x_k|\sigma'(m'(x)_k)_{<k})}{\pi_{\theta^t}(x_k|\sigma(m(x)_k)_{<k})} \mathbb{E}_{\Gamma_\lambda^t \setminus \{x\}} \left[ A_{\Gamma_\lambda^t}(x) \right] \right. \right.$$
$$\left. \left. - \beta D_{\mathrm{KL}} \left( \pi_{\theta^t}(\cdot|\sigma(m(x)_k)_{<k}) \, \| \, \pi_\theta(\cdot|\sigma'(m(x)_k)_{<k}) \right) \right], \quad (34)$$

As with permutations, we can consider different distributions for the dropout mask. In this work, we mainly focus on independent Bernoulli distributions for each entry of the mask matrix, controlled by a hyperparameter $p$. We note in (34) that the dropout mask is applied prior to the causal mask arising from the variable ordering, which allows the combination of both techniques. For the training distribution $\pi_\theta$, this causal mask can be deactivated by simply implementing $\sigma'$ as a table that assigns a negative rank to each dimension.

For any configuration, we can compute the probability $P_{mask}(i,j)$ that a given dimension $j$ from the input is masked when decoding variable $i$. Depending on the setting, we have:

- With the input dropout $m$ only (using Bernoulli parameter $p$): $P_{mask}^{m_p}(i,j) = p$

- With the causal ordering mask $\sigma$ only: $P_{mask}^\sigma(i,j) = 1 - P(\mathrm{rank}_\sigma(j) < \mathrm{rank}_\sigma(i)) = 1 - \sum_{r=1}^n P(\mathrm{rank}_\sigma(i) = r)P(\mathrm{rank}_\sigma(j) < \mathrm{rank}_\sigma(i)|\mathrm{rank}_\sigma(i) = r) = 1 - \frac{1}{n} \sum_{r=1}^n \frac{r-1}{n-1} = 1 - \frac{1}{n(n-1)} \sum_{r=0}^{n-1} r = 1 - \frac{n(n-1)/2}{n(n-1)} = 0.5$

- With the input dropout $m$ and causal ordering mask combined: $P_{mask}^{m_p,\sigma}(i,j) = P_{mask}^{m_p}(i,j) + (1 - P_{mask}^{m_p}(i,j)) \times P_{mask}^\sigma(i,j) = p + (1-p)0.5 = 0.5 + 0.5p$

Thus, it is possible to set a dropout probability $p$ such that the masking probability of an input for decoding any given dimension is similar to the one induced by random permutations of variable order. However, this equivalence only holds for the marginal distribution over single inputs. To go further, let us consider the distribution $P_{\#\mathrm{available}}(k)$, for $k \in [[0, n]]$, where $k$ denotes the exact number of non-masked inputs available for decoding a given variable $i$. Depending on the setting, this distribution can differ significantly between permutations and dropout:

- With the input dropout $m$ only: $P_{\#\mathrm{available}}^{m_p}(k) = P^{m_p}(\text{number of non-masked dimensions before } n) = \binom{n-1}{k} p^{n-k-1} (1-p)^k$

- With the causal ordering mask $\sigma$ only: $P_{\#\mathrm{available}}^\sigma(k) = P(\mathrm{rank}_\sigma(i) = k + 1)$

- With the input dropout $m$ and causal ordering mask combined: $P_{\#\mathrm{available}}^{m_p,\sigma}(k) = \sum_{r=k+1}^n P(\mathrm{rank}_\sigma(i) = r)P^{m_p}(\text{number of non masked dimensions before } r) = \frac{1}{n} \sum_{i=k+1}^n \binom{i-1}{k} p^{i-k-1} (1-p)^k = \frac{1}{n} \sum_{i=0}^{n-k-1} \binom{i+k}{k} p^i (1-p)^k = \frac{1}{n} \frac{1 - I_p(n-k, k+1)}{1-p}$, with $I_p(a,b) = \frac{B(p;a,b)}{B(a,b)}$ the Regularized incomplete Beta function, $B(p; a, b)$ the Incomplete Beta function and $B(a,b)$ the Beta function.

To better illustrate the differences between these settings, Figure 5 shows the distribution of available (non-masked) input variables during neural inference. The x-axis represents $k$, the number of available inputs, and the y-axis shows the corresponding probability. In both settings, input dropout only (left) and input dropout combined with order permutations under a causal mask (right), the dropout probability $p$ has a strong impact. Without a causal mask, the distribution is binomial, with mode at $k = \lfloor (n-1)(1-p) \rfloor$. Each variable is independently available with probability $1 - p$, but this results in a small chance of observing either very small or very large contexts, which is difficult to control efficiently. Ideally, one would prefer a more evenly spread distribution, providing each variable in diverse contexts. In contrast, when combining input dropout with order permutations under a causal mask (right panel), the distribution becomes more evenly spread across $k$. This increases the variety of available contexts for each variable during inference, making it easier to learn robust dependencies. Unlike the purely binomial case, each variable can appear in both small and large contexts (for small $p$ values), which improves controllability and ensures that the model sees diverse conditioning patterns. Notably, the case

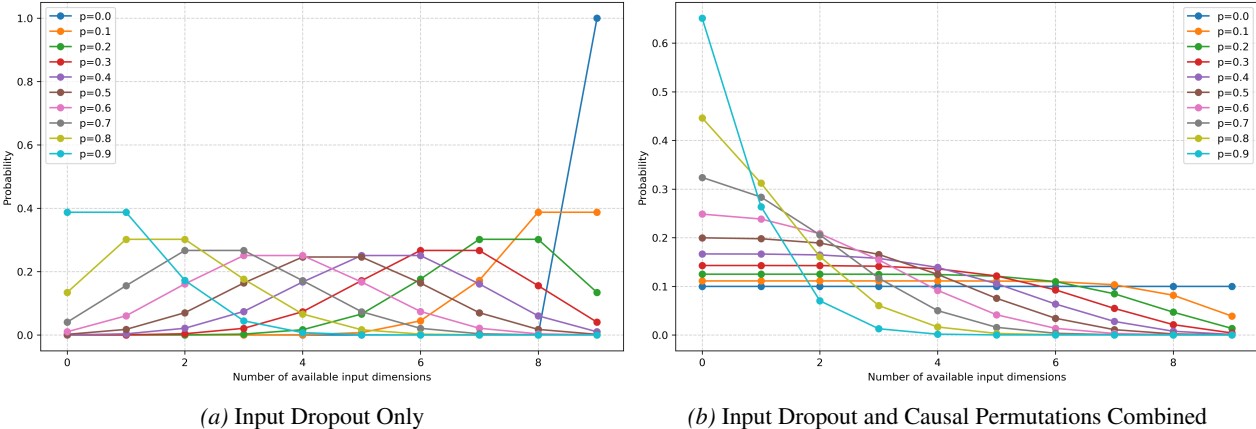

*(a)* Input Dropout Only        *(b)* Input Dropout and Causal Permutations Combined

*Figure 5.* Probability of having exactly $k$ available (non-masked) input variables during neural inference of the generation probabilities of values for any dimension. Left: input dropout without order permutations. Right: input dropout combined with order permutations.

$p = 0$ yields the most uniform distribution of group sizes, enabling more effective structural regularization as discussed above.

The effect of input dropout, either alongside or instead of our generation/training order permutations, is evaluated in Section M.1.

Finally, note that using dropout alone cannot be applied for the generation of individuals, since a sampling order must be defined. One option is a predetermined fixed order, combined with a constant causal mask and dropout. This yields a distribution similar to the binomial case above, with $k$ taken among the $i-1$ positions for the $i$-th variable. However, this approach does not fully exploit structural regularization or population diversity, which would likely require position-dependent parameters. Using varying orders combined with dropout is a potential alternative, but it does not guarantee stable convergence, as input dropout induces information loss during inference. At generation time, this can be detrimental, causing catastrophic forgetting and instability even at the optimum.

In contrast, using permutations of the generation orders without additional dropout is **information-preserving**. For any sampled generation order $\sigma$, the joint distribution $\pi_{\theta^t}(\cdot \mid \sigma)$ can fully exploit all dependencies among variables. Moreover, when the generators become fully order-invariant (which is further encouraged by training order permutations through KL regularization across different orderings), we have $\pi_{\theta^t}(\cdot \mid \sigma) = \pi_{\theta^t}(\cdot \mid \sigma')$ for any pair of generation orders $(\sigma, \sigma') \in \Omega^2$, ensuring complete consistency across all orderings.

## F. On the Choice of the PPO-KL Algorithm as our Backbone for Order-Invariant RL

As we have shown in section E, using random permutations for generation and training in our method can be viewed as a structured dropout of the input features of individuals, which enables various benefits. However, the choice of the KL version of PPO for this purpose is yet to be discussed. This is the focus of this section.

In particular, we can analyze our choices in comparison to findings from (Hausknecht & Wagener, 2022), which also discussed the role of dropout in reinforcement learning and showed that naïvely combining the standard REINFORCE updates with dropout leads to severe instability. Specifically, when the dropout masks differ between trajectory generation and policy updates, the procedure is no longer on-policy, and learning quickly collapses. They investigate PPO in this context, but only the clipped variant. Interestingly, one can observe that the PPO ratio deviates from one even at the first update step (we are no longer on-policy when sampling and training with different masks on layers' inputs). In the clipped version of PPO, this results in most gradients being clipped and therefore prevents meaningful updates. This behavior undermines the intent of clipping—designed to correct occasional overshooting—since here the mechanism blocks learning altogether from the start. To address these issues, the authors propose two strategies for making REINFORCE consistent under dropout: (1) marginalizing over dropout masks, and (2) enforcing identical dropout masks during generation and training (akin to our approach of sampling a permutation during generation and applying the same permutation during training, with $\sigma'$ drawn from a Dirac distribution). The first strategy is theoretically appealing but practically prohibitive, as

even with Monte Carlo approximations using dozens or hundreds of samples, the variance of the estimator overwhelms the learning signal. The second strategy, by contrast, is shown to be more effective and stable.

In our work, we revisit this question from a different angle. While Hausknecht et al. argue that consistency requires using the same dropout mask between rollout and update, we posit that sampling different conditioning patterns at update time can in fact be beneficial. By exposing the policy to multiple conditioning variations from the same rollout, the training process gains additional signal, thereby improving sample efficiency. To make this feasible, we rely on PPO rather than plain REINFORCE. PPO naturally tolerates updates from slightly different policies, which aligns well with our setting where updates need not be fully on-policy. Moreover, we adopt the KL-regularized version of PPO, which avoids the blocking issues observed with the clipped variant: instead of discarding gradients when ratios diverge, the KL penalty smoothly regularizes the policy towards the sampling distribution. This design choice is key to enabling effective training under random permutations.

Importantly, Hausknecht et al. developed their Dropout-Marginalized Gradient in the context of REINFORCE, which forces them to approximate, via Monte Carlo sampling, the exact dropout distribution used during rollout. This requires likelihood normalization over many sampled masks, and thus demands a prohibitively large number of samples to achieve a low-variance estimator. By contrast, in our KL-PPO framework we only need to compute expectations of gradients under the current mask distribution, without approximating the rollout distribution itself. This allows us to train efficiently with as little as a single mask sample per example and iteration, a much lighter procedure in practice.

## G. Connection with Natural Gradient and Information-Geometric Optimization Algorithm

The Information-Geometric Optimization (IGO) algorithm (Ollivier et al., 2017) is a natural gradient method that seeks to maximize a quantile-based transformation of the objective function $f$.

For our probabilistic model $\pi_\theta$ with $\theta \in \Theta$, and given a permutation $\sigma \in \Omega$, the IGO flow that defines the trajectory in the parameter space $\Theta$ to maximize the objective $\mathbb{E}_{x \sim \pi_\theta(x|\sigma)}[A_{\Gamma_\lambda^t}(x)]$ is given by (see Definition 5 in (Ollivier et al., 2017))

$$\theta^{t+\delta_t} = \theta^t + \delta_t I^{-1}(\theta^t) \sum_{i=1}^{\lambda} A_{\Gamma_\lambda^t}(x^i) \frac{\nabla \ln \pi_\theta(x^i|\sigma)}{\nabla \theta}\bigg|_{\theta=\theta^t}, \tag{35}$$

with $x^i$ for $i = 1, \ldots, \lambda$ generated by the model $\pi_{\theta^t}$ at time-step $t$ and where $I^{-1}(\theta^t)$ denotes the inverse Fisher information matrix of $\pi_{\theta^t}$.

When $\delta t$ is close to 0, and using Theorem 10 in (Ollivier et al., 2017), (35) can be rewritten as

$$\theta^{t+\delta_t} = \underset{\theta \in \Theta}{\operatorname{argmax}} \left( (1 - \delta_t \sum_{i=1}^{\lambda} A_{\Gamma_\lambda^t}(x^i)) \int \ln \pi_\theta(x|\sigma) \pi_{\theta^t}(dx) + \delta_t \sum_{i=1}^{\lambda} A_{\Gamma_\lambda^t}(x^i) \ln \pi_\theta(x^i|\sigma) \right). \tag{36}$$

When using this framework with our probabilistic model $\pi_\theta(x|\sigma) = \prod_{k=1}^{n} \pi_\theta(x_{\sigma_k}|x_{\sigma<k}, \sigma)$ this yields

$$\theta^{t+\delta_t} = \underset{\theta \in \Theta}{\operatorname{argmax}}[(1 - \delta_t \sum_{i=1}^{\lambda} A_{\Gamma_\lambda^t}(x^i)) \int \sum_{k=1}^{n} \ln \pi_\theta(x_{\sigma_k}|x_{\sigma<k}, \sigma) \pi_{\theta^t}(dx)$$

$$+ \delta_t \sum_{i=1}^{\lambda} \sum_{k=1}^{n} A_{\Gamma_\lambda^t}(x^i) \ln \pi_\theta(x^i_{\sigma_k}|x^i_{\sigma<k}, \sigma)] \tag{37}$$

As the maximization is on $\theta$ we can subtract the term $(1 - \delta_t \sum_{i=1}^{\lambda} A_{\Gamma_\lambda^t}(x^i)) \int \sum_{j=1}^{n} \ln \pi_{\theta^t}(x_{\sigma_k}|x_{\sigma<k}, \sigma) \pi_{\theta^t}(dx)$ that does not depend on $\theta$. Therefore, we have

$$\theta^{t+\delta_t} = \underset{\theta}{\arg\max}[\delta t \sum_{i=1}^{\lambda} \sum_{k=1}^{n} A_{\Gamma_\lambda^t}(x^i) \ln \pi_\theta(x_{\sigma_k}^i | x_{\sigma<k}^i, \sigma)$$

$$+ (\delta_t \sum_{i=1}^{\lambda} A_{\Gamma_\lambda^t}(x^i) - 1) \sum_{k=1}^{n} \int \ln \frac{\pi_{\theta^t}(x_{\sigma_k} | x_{\sigma<k}, \sigma)}{\pi_\theta(x_{\sigma_k} | x_{\sigma<k}, \sigma)} \pi_{\theta^t}(dx)]. \quad (38)$$

Now using the $\lambda$ samples to approximate the integral over the domain $\mathcal{X}_{\sigma<k}$, and using the fact that all conditional Markov kernels are independent we have for $k = 1, \dots, n$

$$\int \ln \frac{\pi_{\theta^t}(x_{\sigma_k} | x_{\sigma<k}, \sigma)}{\pi_\theta(x_{\sigma_k} | x_{\sigma<k}, \sigma)} \pi_{\theta^t}(dx) \approx \frac{1}{\lambda} \sum_{i=1}^{\lambda} \int \ln \frac{\pi_{\theta^t}(x_{\sigma_k} | x_{\sigma<k}^i, \sigma)}{\pi_\theta(x_{\sigma_k} | x_{\sigma<k}^i, \sigma)} \pi_{\theta^t}(dx_{\sigma_k}). \quad (39)$$

Thus, we have for $k = 1, \dots, n$

$$\int \ln \frac{\pi_{\theta^t}(x_{\sigma_k} | x_{\sigma<k}, \sigma)}{\pi_\theta(x_{\sigma_k} | x_{\sigma<k}, \sigma)} \pi_{\theta^t}(dx) \approx \frac{1}{\lambda} \sum_{i=1}^{\lambda} D_{\mathrm{KL}}\left(\pi_{\theta^t}(\cdot | x_{\sigma<k}^i, \sigma) \,\|\, \pi_\theta(\cdot | x_{\sigma<k}^i, \sigma)\right) \quad (40)$$

Using (39) and defining $\beta = \frac{1}{\lambda\delta t} - \frac{\sum_{i=1}^{\lambda} A_{\Gamma_\lambda^t}(x^i)}{\lambda}$, the maximization objective of (38) for the update of the model at each generation becomes

$$L'(\theta) = \frac{1}{\lambda} \sum_{i=1}^{\lambda} \sum_{k=1}^{n} \left[\ln \pi_\theta(x_{\sigma_k}^i | x_{\sigma<k}^i, \sigma) A_{\Gamma_\lambda^t}(x^i) - \beta D_{\mathrm{KL}}\left(\pi_{\theta^t}(\cdot | x_{\sigma<k}^i, \sigma) \,\|\, \pi_\theta(\cdot | x_{\sigma<k}^i, \sigma)\right)\right]. \quad (41)$$

The update phase of the algorithm can then be interpreted as the maximization of a weighted log-likelihood over the individuals in the current generation, regularized by a KL divergence term. This regularization penalizes excessive reductions in the entropy of the sampling distribution, thereby maintaining a degree of diversity in the population. By controlling the rate of convergence, this mechanism prevents premature collapse of the distribution onto a single high-performing individual, which could otherwise lead to early stagnation in a local optimum.

It corresponds to the surrogate objective of our GRPO-based framework given by 4 when replacing each term $\ln \pi_\theta(x_{\sigma_k}^i | x_{\sigma<k}^i, \sigma)$ by the importance sampling ratio $\frac{\pi_\theta(x_{\sigma_k}^i | x_{\sigma<k}^i, \sigma)}{\pi_{\theta^t}(x_{\sigma_k}^i | x_{\sigma<k}^i, \sigma)}$. We empirically observed that maximizing the ratio of importance sampling instead of the log-probability yields better results in our context, therefore in the following we stay with the formulation of the objective given by (4) instead of (41).

## H. Algorithm Pseudocode

In this appendix, we detail the pseudo-code of the multivariate RL EDA with Algorithm 1, which includes the four multivariate RL EDA variants presented in Section 3.3: $(\delta, \delta)$-RL-EDA, $(\delta, \sigma)$-RL-EDA, $(\sigma, \delta)$-RL-EDA and $(\sigma, \sigma)$-RL-EDA.

Until the termination criterion is met, this EDA performs the following steps at each generation $t$:

1. Draw a population $\Gamma_t = \{(x^i, \sigma_G^i)\}_{i=1}^{\lambda}$ from the joint distribution $\pi_{\theta^t}(x | \sigma_G)\xi(\sigma_G)$.

2. Order the individuals according to their fitness, and compute the advantage $A_{\Gamma_\lambda^t}(x^i)$ for each individual.

3. Update the probabilistic model by maximizing during $E$ epochs the objective

$$\hat{L}_\lambda(\theta) = \frac{1}{\lambda} \sum_{(x^i, \sigma_G^i) \in \Gamma_t} \mathbb{E}_{\sigma_T \sim \xi(\sigma_T | \sigma_G^i)} \sum_{k=1}^{n} [\frac{\pi_\theta(x_k^i | \sigma_T(x^i)_{<k})}{\pi_{\theta^t}(x_k^i | \sigma_G^i(x^i)_{<k})} A_{\Gamma_\lambda^t}(x^i) - \beta D_{\mathrm{KL}}\left(\pi_{\theta^t}(\cdot | \sigma_G^i(x^i)_{<k}) \,\|\, \pi_\theta(\cdot | \sigma_T(x^i)_{<k})\right)].$$

$$(42)$$

In practice, at each epoch in order to reduce computation time, the expectation $\mathbb{E}_{\sigma_T \sim \xi(\sigma_T | \sigma_G^i)}[.]$ is replaced by an evaluation based on a single sample.

---

**Algorithm 1** $(\sigma, \sigma)$-RL-EDA with parameters $\lambda \in \mathbb{N}^*$, $\beta \in \mathbb{R}^+$, utility function $U$, number of epochs $E$ and functional mechanism $g$.

---

**Input:** an instance $(\mathcal{X}, f)$, with $\mathcal{X} = \{-1, 1\}^n$, $f : \mathcal{X} \to \mathbb{R}$ and a number of iterations $T$.
Randomly initialize the parameters $\theta^0 = (\theta_1^0, \ldots, \theta_n^0)$.
$x^* \leftarrow \emptyset$ and $f(x^*) \leftarrow -\infty$.
**for** $t = 0, 1, 2, \ldots, T - 1$ **do**
    **for** $i = 1, 2, \ldots, \lambda$ **do**
        $x^i \leftarrow (0, \ldots, 0)$.
        Draw a permutation $\sigma_G^i \sim \xi(\sigma_G)$.
        Generate solution $x^i$ in the order of generation $\sigma_G^i$:
        **for** $k = 1, 2, \ldots, n$ **do**
            $x_{\sigma_G^i(k)}^i \sim \text{Bernoulli}(\text{sigmoid}(g_{\theta_{\sigma_G^i(k)}}(x_{\sigma_G^i < k})))$
        **end for**
    **end for**
    **for** $i = 1, 2, \ldots, \lambda$ **do**
        Compute $f(x^i)$.
        **if** $f(x^i) > f(x^*)$ **then**
            $x^* \leftarrow x^i$
        **end if**
    **end for**
    **for** $i = 1, 2, \ldots, \lambda$ **do**
        Compute $A_{\Gamma_\lambda^t}(x^i) = U\left(\frac{\text{rk}(x^i)}{\lambda - 1}\right)$.
    **end for**
    $\theta \leftarrow \theta^t$
    **for** $e = 1, 2, \ldots, E$ **do**
        **for** $i = 1, 2, \ldots, \lambda$ **do**
            $\sigma_T^{(i)} \sim \xi(\sigma_T | \sigma_G)$.
        **end for**
        Compute

$$\hat{L}_\lambda(\theta) = \frac{1}{\lambda} \sum_{(x^i, \sigma_G^i) \in \Gamma_t} \sum_{k=1}^n \left[ \frac{\pi_\theta(x_k^i | \sigma_T^{(i)}(x^i)_{<k})}{\pi_{\theta^t}(x_k^i | \sigma_G^i(x^i)_{<k})} A_{\Gamma_\lambda^t}(x^i) - \beta D_{\text{KL}}\left(\pi_{\theta^t}(\cdot | \sigma_G^i(x^i)_{<k}) \,\|\, \pi_\theta(\cdot | \sigma_T^{(i)}(x^i)_{<k})\right) \right]. \quad (43)$$

        Compute $\nabla_\theta \hat{L}_\lambda(\theta)$ and update $\theta$ with gradient ascent.
    **end for**
    $\theta^{t+1} \leftarrow \theta$
**end for**
**Output::** the best solution found $x^*$

---

# I. Multivariate EDA with Learned Order

In this appendix, we derive a version of the multivariate EDA learned with PPO, called Learned-$\sigma$-RL-EDA where we model the distribution of order with the Plackett-Luce (PL) distribution (Plackett, 1975) parameterized by the vector of scores $w = (w_1, \ldots, w_n)$ (this distribution is denoted $\xi_w^{PL}(\sigma)$ hereafter) and we use the reparameterization trick proposed by (Grover et al., 2019) to learn $w$ by gradient descent.

### I.1. Plackett-Luce Distribution

For each $\sigma \in \Omega$, and given $w \in \mathbb{R}^n$ the probability mass function of the PL distribution is given by

$$\xi_w^{PL}(\sigma) = \frac{w_{\sigma(1)}}{Z} \frac{w_{\sigma(2)}}{Z - w_{\sigma(1)}} \cdots \frac{w_{\sigma(n)}}{Z - \sum_{k=1}^{n-1} w_{\sigma(k)}}, \tag{44}$$

with $Z = \sum_{i=1}^n w_i$ a normalization constant.

Let $sort : \mathbb{R}^n \to \Omega$ be the operator mapping an n-dimensional real-valued vector to a permutation $\sigma$ corresponding to a descending ordering of its values. Let $W$ denote the matrix of absolute pairwise differences of the elements of $w$ such that $W_{ij} = |w_i - w_j|$. As shown by (Grover et al., 2019), the permutation matrix $P_{sort(w)}$ corresponding to $sort(w)$ is given by:

$$P_{\text{sort}(w)}[i,j] = \begin{cases} 1 \text{ if } j = \text{argmax}[(n + 1 - 2i)w - W\mathbb{1}] \\ 0 \text{ otherwise,} \end{cases} \tag{45}$$

where $\mathbb{1}$ denotes the column vector of all ones.

In practice to sample from $\xi_w(\sigma)$, (Grover et al., 2019) propose a method for sampling from PL distributions with parameters $w$ by sampling for $k = 1, \ldots, n$ a noise $\epsilon_k \sim \text{Gumbel}(0, 1)$ with zero mean and unit scale, then by computing $\tilde{w}$ as the vector of perturbed log-scores with entries such that $\tilde{w}_i = \ln w_i + \epsilon_i$, and lastly by applying the sort operator to the perturbed log-scores $\tilde{w}_i$. The resulting order gives a permutation $\sigma$ sampled from $\xi_w^{PL}(\sigma)$. Indeed (Grover et al., 2019) show that $\mathbb{P}(\tilde{w}_{\sigma(1} \geq \cdots \geq \tilde{w}_{\sigma(n)}) = \xi_w(\sigma)$ (see Proposition 5).

For a vector $\tilde{w}$ of perturbed log-score, the sampled permutation matrix is $P_{sort(\tilde{w})}$ corresponding to permutation $\tilde{\sigma}$, such that $\left[P_{sort(\tilde{w})}\right]_{ij} = 1$ if $i = \tilde{\sigma}(j)$ and 0 otherwise. This permutation matrix allows to compute the adjacency matrix $\tilde{M} = P_{sort(\tilde{w})}^\top B P_{sort(\tilde{w})}$ of the sampling directed acyclic graph (DAG), with $B$ the strictly upper triangular binary matrix of size $n \times n$, whose entries are defined as $b_{i,j} = 1$ if $j > i$, and $b_{i,j} = 0$ otherwise. Each column vector $m_k$ at position $k$ of $\tilde{M}$ corresponds to the binary causal mask used at step $k$ to mask the entries of $g$ (see Section 3.1).

### I.2. Plackett-Luce Reparameterization Trick

Computing the permutation matrix $P_{sort(\tilde{w})}$ from $w$ is a non-differentiable operation due to the use of the argmax function. Therefore, (Grover et al., 2019) propose to replace $P_{sort(\tilde{w})}$ by the continuous relaxation $\widehat{P}_{sort(\tilde{w})}$ using the softmax function instead of the argmax function when gradient computations are required. The $i$-th row of $\widehat{P}_{sort(w)}$ is given by

$$\widehat{P}_{sort(w)} = \text{softmax}[(n + 1 - 2i)w - W\mathbb{1}/\tau], \tag{46}$$

with $\tau > 0$ a temperature parameter (set at the value of 1 in the following).

### I.3. Learned-$\sigma$-EDA Algorithm

During the sampling phase of `Learned-σ-RL-EDA`, to generate each individual of the population, an order $\sigma^i$ is first sampled from $\xi_w^{PL}(\sigma)$, then $x^i$ is sampled from $\pi_{\theta^t}(\cdot|\sigma^i)$.

During the update phase of the EDA we maximize the following GRPO objective with respect to $(\theta, w)$:

$$\hat{L}_\lambda(\theta, w) = \frac{1}{\lambda} \sum_{(x^i, \sigma^i) \in \Gamma_t} \mathbb{E}_{\sigma' \sim \xi_w^{PL}(\sigma)} \sum_{k=1}^n [\frac{\pi_\theta(x_k^i|\sigma'(x^i)_{<k})}{\pi_{\theta^t}(x_k^i|\sigma^i(x^i)_{<k})} A_{\Gamma_\lambda^t}(x^i) - \beta D_{\text{KL}}\left(\pi_{\theta^t}(\cdot|\sigma^i(x^i)_{<k}) \| \pi_\theta(\cdot|\sigma'(x^i)_{<k})\right)]. \tag{47}$$

This maximization is performed using first-order gradient descent using $\nabla_\theta L(\theta, w)$ and $\nabla_w L(\theta, w)$ (computed with the reparameterization trick).

# J. Synthetic Dataset Generation and Experimental Protocol

We examine the following NP-hard problems in this work. For each of these problems, we generated instances of size $n \in \{64, 128, 256\}$, and for each size, we considered different types of instances.

**The Quadratic Unconstrained Binary Optimization Problem (QUBO)** aims to find a pseudo-Boolean vector $x = (x_1, \ldots, x_n)$ of size $n$ maximizing the function $f : \{-1, 1\}^n \to \mathbb{R}$ given by $f(x) = x^\top Q x$, where $Q$ is a symmetric real matrix of size $n \times n$. We generate QUBO instances using the $\text{PUBO}_i$ generator (Tari et al., 2022), which enables the creation of QUBO problems with controlled structural properties. The parameters of the $\text{PUBO}_i$ generator are set to produce six different instance types $K$ by tuning both the density of the QUBO matrix $Q$ and the relative importance of binary variables, thereby influencing the degree of non-uniformity in $Q$.

Formally, the fitness function of each instance of this QUBO problem is defined as $f(x) = \sum_{i=1}^{m} f_i(x_{i_1}, x_{i_2}, x_{i_3}, x_{i_4})$, where each sub-function $f_i$ is a quadratic function randomly selected from the set $\{\varphi_1, \ldots, \varphi_4\}$. Each $\varphi_k$ is designed to have $2k$ symmetric local optima. In $\text{PUBO}_i$, binary variables are divided into two importance classes: important and non-important variables. For each sub-function $f_i$, the four variables $x_{i_j}$ are selected according to an importance degree parameter $d$, where the probability of selecting an important variable is proportional to $d$. An additional importance co-appearance parameter $\alpha$ controls the correlation in the selection of important variables: higher $\alpha$ values increase the likelihood that two important variables co-occur within the same sub-function $f_i$. The number of sub-functions is given by $m = r \times \frac{n(n-1)}{2}$, where $r$ is a density coefficient controlling the proportion of non-zero entries in $Q$. For example, with $r = 0.05$ and $r = 0.2$, the density of $Q$ is approximately 16% and 43%, respectively, for uniform instances.

We consider three interaction configurations:

- Uniform random instances when $(d, \alpha) = (1, 1)$, corresponding to no specific important variables, i.e., a fully random QUBO structure.

- Instances with $(d, \alpha) = (10, 1)$, where important variables are 10 times more likely to be selected than non-important variables, but selections are independent.

- Instances with $(d, \alpha) = (10, 1.09)$: the selection of important variables is not independent, and the selection of important variables is concentrated.

Further details on the $\text{PUBO}_i$ generator can be found in (Tari et al., 2022). By combining parameters $r$, degree $d$ of importance of variables and parameter $\alpha$ of co-appearance, we obtain six different types of instances described in Table 1.

*Table 1.* Parameters of $\text{PUBO}_i$ instances.

| Type instance $K$ | $r$ | $d$ | $\alpha$ |
|---|---|---|---|
| 0 | 0.05 | 1 | 1 |
| 1 | 0.05 | 10 | 1 |
| 2 | 0.05 | 10 | 1.09 |
| 3 | 0.2 | 1 | 1 |
| 4 | 0.2 | 10 | 1 |
| 5 | 0.2 | 10 | 1.09 |

**The NKD model** is a natural extension of the NK model of Kauffman (Kauffman & Weinberger, 1989) to cases where variables can take more than two categorical values. It is a framework for describing fitness landscapes whose problem size and ruggedness are both parameterizable. The NKD function is defined as $f_{\text{NKD}} : \{0, 1, \ldots, D - 1\}^n \to [0, 1[$ and takes the same form as NK functions: $f_{\text{NKD}}(x) = \frac{1}{n} \sum_{i=1}^{n} \gamma_i(x_i, x_{l_{i1}}, \ldots, x_{l_{iK}})$, except that each subfunction $\gamma_i : \{0, 1, \ldots, D - 1\}^{K+1} \to [0, 1[$ is defined over categorical variables with $D$ possible values instead of binary ones. We construct instances with $D = 2$, which corresponds to the original pseudo-boolean NK problem, but we also construct instances of a categorical problem called NK3 with $D = 3$. For each variant NK or NK3 of the problem four different types of distribution of instances with $K \in \{1, 2, 4, 8\}$ are built. When $K = 1$, the interaction graph is very sparse and the landscape is smooth; when $K = 8$, the landscape becomes significantly more rugged.

Unless otherwise specified, we treat these problems as black-box problems, meaning that both the objective function and the interaction graph between variables are assumed to be unknown. For each pair $(n, K)$ and for each problem, we generated 10 different instances. For the sake of reproducibility, all these instances are available at the URL `https:`

. For each problem instance, we allow for a maximum budget of 10,000 objective function evaluations. The best solution found since the beginning of the search is recorded every 100 evaluations. For each distribution of instances, defined by the feature vector $(pb, n, K)$ (with $pb$ the problem name, $n$ the instance size and $K$ the type of instance), and for each algorithm, we compute the average performance over 10 distinct instances, each solved with 10 independent restarts using different random seeds. This procedure results in 100 independent runs per algorithm and per instance distribution, from which the average score evolution is reported. It is worth noting that, within a given distribution, the best scores obtained across the 10 instances are of comparable magnitude, which justifies averaging them to produce a single representative performance measure.

## K. Multivariate EDA Hyperparameter Configuration and Computing Time

In this appendix, we detail the hyperparameter configuration of the multivariate RL EDA presented in Section 3.3, which is used as a baseline for all experiments, and give some details on the complexity and computing time of the proposed approach.

### K.1. Hyperparameter Configuration

The population size is set by default to $\lambda = 10$ across all benchmark instances. Although fine-tuning this parameter may lead to better performance for specific distributions of problem instances, and may also depend on the instance dimension $n$, we opt for simplicity and maintain a constant value throughout this work. A sensitivity analysis of this key parameter is presented in Subsection M.4.

By default, each functional mechanism $g_{\theta_i}$ for $i = 1, \ldots, n$ is implemented as a feedforward neural network with a single hidden layer of 20 neurons, using the hyperbolic tangent activation function. This choice is particularly advantageous, as it allows the network to approximate both nonlinear and linear relationships when needed. Using single-hidden-layer neural networks for each variable strikes a practical balance between model expressiveness and computational efficiency, especially given the instance sizes considered in this study. Nevertheless, as discussed in Appendix M.7, we explore alternative configurations—such as linear models and deeper neural networks—which may offer improved performance on more complex tasks, albeit at the cost of increased computational time.

The utility function $U$ used in the advantage calculation of (5) is defined as a linearly decreasing function on the interval $[0, 1]$, specifically $U(x) = 1 - 2x$. Under this definition, the best individual $x_{\text{best}}^i$ in the current population, with $\text{rk}(x_{best}^i) = 0$, receives a reward $A_{\Gamma_\lambda^t}(x_{best}^i) = 1$, whereas the worst individual $x_{\text{worst}}^i$, with $\text{rk}(x_{worst}^i) = \lambda - 1$, receives $A_{\Gamma_\lambda^t}(x_{worse}^i) = -1$. If $\lambda$ is odd, the individual with median fitness obtains an advantage of zero. With this choice of $U$, maximizing (8) assigns the greatest weight to increasing the likelihood of generating the best individual in the population, while simultaneously decreasing the likelihood of generating the worst individual. As a result, the policy is updated so that, in the next generation $t + 1$, it tends to produce individuals that are closer to the best members of generation $t$, and farther from the worst ones. It is worth noting that a fine-tuned utility function may yield superior performance for specific distributions of problem instances. Prior research has investigated the impact of selecting appropriate utility values or importance weights. For example, in the context of the CMA-ES algorithm, (Andersson et al., 2015) showed that adapting these parameters to the distribution of instances can lead to significant performance improvements. Specifically, for smooth landscapes with a single local optimum, a utility function that assigns disproportionately high values to the very best individuals can be advantageous. Conversely, for highly deceptive landscapes, it may be beneficial to assign the highest weights to the worst-performing individuals in the population.

Regarding the coefficient for the KL regularization term, we consistently set $\beta = 1$. A sensitivity analysis of this parameter is presented in Subsection M.5. Appendix N additionally presents a variant of the algorithm in which $\beta$ is adaptively adjusted according to the local landscape ruggedness.

At each generation, the algorithm is trained for $E = 50$ epochs using the Adam optimizer (Kingma & Ba, 2014) with an initial learning rate 0.001. In practice, to avoid numerical issues in the multivariate RL EDAs, particularly division by zero when evaluating the KL divergence term or the importance sampling ratio, we apply clipping to the probability values of each conditional distribution $\pi_\theta(\cdot|\sigma'(x^i)_{<k})$. Specifically, all probabilities are clipped to lie within the interval $[\epsilon, 1 - \epsilon]$, with $\epsilon = 0.001$.

Table 2 summarizes all hyperparameters used in the multivariate RL EDA.

*Table 2.* Hyperparameters settings for $(\sigma, \sigma)$-RL-EDA

| Parameter | Description | Value |
|---|---|---|
| | EDA parameters | |
| $\lambda$ | Size of the population | 10 |
| $L$ | Number of hidden layers in $g$ | 1 |
| $n_l$ | Number of neurons in hidden layer | 20 |
| $\epsilon$ | Probability threshold coefficient | 0.001 |
| | PPO parameters | |
| $U$ | Utility function | $U(x) = 1 - 2x$ |
| $\beta$ | KL penalty parameter | 1 |
| $E$ | Number of training epochs | 50 |
| $l_r$ | Learning rate of Adam optimizer | 0.001 |

### K.2. Time and Space Complexity

The overall time complexity of the proposed RL-EDA algorithm can be decomposed into two main components:

1. Solution Generation: For each iteration $t$ of the EDA, a population of size $\lambda$ is sampled. Each solution has $n$ variables generated sequentially by neural networks with one hidden layer of size $h$. The cost per forward pass for one variable is $O(nh)$. For $\lambda$ solutions with $n$ variables per solution $O(\lambda n^2 h)$.

2. Policy Update (Training) : For $E$ epochs per generation, each epoch recomputes masked inputs and performs gradient updates $O(E\lambda n^2 h)$

Hence the total complexity for $T$ generations is $O(T.E.\lambda.n^2 h)$. Note that a classical BOA (Pelikan, 2002) typically leads to $O(n^3)$.

Concerning space complexity, using the $n$ small NNs in the standard version, we get an $O(n^2 h)$ space complexity, which decreases to $O(nh)$ for the shared-parameter variant (see Appendix R).

### K.3. Computing Time

The multivariate RL EDA algorithm is implemented in Python 3.7 with PyTorch 2.5 library for tensor calculation with CUDA 12.4. The source code is available at the URL `https://github.com/GoudetOlivier/RL-EDA`. It is specifically designed to run on GPU devices.

When using the hyperparameters described in Table 2, the time required to process a single QUBO instance of size $n = 128$, with a budget of 10,000 calls to the objective function, corresponding to 1,000 generations of the algorithm when $\lambda = 10$, is approximately 11.5 minutes on a single Intel(R) Xeon(R) Silver 4208 CPU at 2.10GHz, and 5 minutes on an Nvidia V100 GPU device (including the 10,000 objective function evaluations). The code is also adapted to process batches of multiple instances of the same size in parallel, which greatly benefits from GPU parallelism. In particular, it can process 100 QUBO instances of size $n = 128$, each with a budget of 10,000 objective function calls, in 20 minutes on a single V100 GPU device.

We have also implemented a version of the algorithm with shared parameters in the architecture, called $(\sigma, \sigma)$-RL-EDA-share-params (see Appendix R), which scales better in term of CPU/GPU usage.

Table 3 provides more details on wall-clock times required to solve QUBO instances of different sizes with a budget of 10,000 evaluations for the standard version $(\sigma, \sigma)$-RL-EDA and the version with shared parameters $(\sigma, \sigma)$-RL-EDA-share-params, in comparison with BOA EDA and a strong Nevergrad baseline CMApara. Times are given in seconds and evaluated for CPU on Xeon(R) Silver 4208 at 2.10GHz and for GPU on an Nvidia V100 GPU device. Note that a runtime of 2940 seconds to solve a big instance of size $n = 256$ with a budget of 10,000 on a CPU with $(\sigma, \sigma)$-RL-EDA remains acceptable, as it corresponds to only 0.29 seconds per generated solution, and for a black box problem such as neural architecture search (see Appendix O), the time required to evaluate a single solution is generally much more costly.

These times are provided for indicative purposes only, as the main criterion used to assess the performance of a black-box algorithm is typically the best score obtained within a limited number of calls to the objective function—a criterion that is

*Table 3.* CPU/GPU wall-clock times required to solve a QUBO instance with a budget of 10,000 calls to the objective function.

| Size instance | CPU time (s) | GPU time (s) |
|---|---|---|
| CMApara (Nevergrad) | | |
| 64 | 61 | - |
| 128 | 78 | - |
| 256 | 141 | - |
| Multivariate BOA EDA | | |
| 64 | 460 | - |
| 128 | 2100 | - |
| 256 | 9310 | - |
| $(\sigma, \sigma)$-RL-EDA | | |
| 64 | 450 | 210 |
| 128 | 690 | 300 |
| 256 | 2940 | 420 |
| $(\sigma, \sigma)$-RL-EDA-share-params | | |
| 64 | 300 | 195 |
| 128 | 420 | 255 |
| 256 | 780 | 300 |

precisely retained in our experimental analyses and benchmark comparisons.

## L. Global Experimental Results

Table 4 presents a selection of these results, comparing $(\sigma, \sigma)$-RL-EDA to three other EDAs of the same category: PBIL, MIMIC and BOA. The final columns report the performance of the best algorithm among all Nevergrad algorithms. For each algorithm, we report the average score obtained after 10,000 calls to the objective function, over 100 independent runs. Based on this average score, the algorithms are ranked, and their ranking among all competitors is reported.

To facilitate comparison between our proposed algorithm, $(\sigma, \sigma)$-RL-EDA, and the best-performing competing methods, we conducted statistical significance tests. In Table 4, a star next to the results of $(\sigma, \sigma)$-RL-EDA indicates that its average performance over 100 runs is statistically significantly better than that of the best other competing algorithm. Conversely, a star next to a competing algorithm denotes that it significantly outperforms $(\sigma, \sigma)$-RL-EDA on average. Statistical significance is assessed using a two-sample t-test with a p-value threshold of 0.001.

We observe in Table 4 that $(\sigma, \sigma)$-RL-EDA consistently outperforms the other EDAs for instances of size $n = 128$ and $n = 256$. Interestingly, among the three competing EDAs, the univariate PBIL algorithm achieves the best results.[4] This confirms empirical findings previously reported by (Doerr & Dufay, 2022), which suggest that univariate EDAs can sometimes match or even surpass the performance of more complex multivariate EDAs. One possible explanation is that the number of parameters to be learned in multivariate models such as MIMIC and BOA increases rapidly with instance size, potentially slowing convergence compared to the simpler PBIL.

We also observe in this table that $(\sigma, \sigma)$-RL-EDA is not the best on small instances with $n = 64$ in comparison with the best competitors of the Nevergrad library. This is because $(\sigma, \sigma)$-RL-EDA with this set of hyperparameters converges too quickly for these instances, as discussed in Appendix P. On the other hand, $(\sigma, \sigma)$-RL-EDA consistently outperforms all competitors for this instance size with a budget of 1,000 evaluations instead of 10,000.

In addition to the global results table, we also provide plots showing the evolution of the best scores (averaged over 100 runs) as a function of the number of objective function evaluations. In each plot, the curve for $(\sigma, \sigma)$-RL-EDA is always displayed in green and placed first in the legend, for consistency. It is compared against the 10 best-performing competing algorithms, listed in the legend from best to worst.

Here, we present these curves only for the different instance types of size $n = 128$ from the pseudo-Boolean QUBO problem (Figure 6) and the categorical NK3 problem (Figure 7).[5] Note that for the QUBO instance distribution with $n = 128$ and

---

[4]Since PBIL is designed specifically for pseudo-Boolean optimization, it was not evaluated on NK3 instances involving variables with three categorical values

[5]All plots for all instance distributions are available at the URL https://github.com/GoudetOlivier/RL-EDA.

*Table 4.* Global rankings and average scores obtained by $(\sigma, \sigma)$-RL-EDA and the other EDAs (PBIL, MIMIC, and BOA) are reported. The last columns present the ranking and average score of the best-performing method among the 500 additional algorithms considered (496 for NK3 problems). Rankings are computed over all 504 algorithms (499 for NK3 problems) by comparing the best score achieved after 10,000 objective function evaluations, averaged across 100 independent runs. Bold values highlight the best results among all competing methods. A star associated with the results obtained by $(\sigma, \sigma)$-RL-EDA indicates that it is significantly better on average (over 100 runs) than the best other competitor. A star associated with a result obtained by another algorithm indicates that it is significantly better on average (over 100 runs) than $(\sigma, \sigma)$-RL-EDA. A difference on the average scores is said statistically significant according to a t-test with p-value 0.001.

| Instances | | | Methods | | | | | | | | | | |
|---|---|---|---|---|---|---|---|---|---|---|---|---|---|
| | | | $(\sigma,\sigma)$-RL-EDA | | PBIL | | MIMIC | | BOA | | Best method (others) | | | |
| Pb | $n$ | $K$ | Rank | Score | Rank | Score | Rank | Score | Rank | Score | Name | Rank | Score |
| QUBO | 64 | 0 | 33/504 | 200.8 | 61/504 | 199.8 | 249/504 | 188.2 | 267/504 | 184.6 | **CMAL3** | **1/504** | **206.2*** |
| QUBO | 64 | 1 | 82/504 | 148.8 | 91/504 | 147.8 | 134/504 | 146.0 | 140/504 | 145.4 | **CMApara** | **1/504** | **154.3*** |
| QUBO | 64 | 2 | 115/504 | 138.1 | 88/504 | 139.1 | 119/504 | 137.6 | 154/504 | 137.4 | **DiscreteDE** | **1/504** | **143.4*** |
| QUBO | 64 | 3 | 79/504 | 411.2 | 89/504 | 410.4 | 264/504 | 379.4 | 266/504 | 377.5 | **ChainCMAwithRsqrt** | **1/504** | **430.7*** |
| QUBO | 64 | 4 | 113/504 | 326.1 | 80/504 | 329.7 | 264/504 | 311.7 | 275/504 | 309.7 | **CMApara** | **1/504** | **344.2*** |
| QUBO | 64 | 5 | 76/504 | 309.4 | 65/504 | 310.0 | 241/504 | 298.3 | 260/504 | 295.9 | **CMApara** | **1/504** | **319.3*** |
| QUBO | 128 | 0 | **1/504** | **593.7*** | 65/504 | 570.8 | 256/504 | 504.4 | 224/504 | 517.2 | DiscreteLengler2OnePlusOne | 2/504 | 587.7 |
| QUBO | 128 | 1 | 2/504 | 449.2 | 21/504 | 438.3 | 241/504 | 408.4 | 226/504 | 413.0 | **CMApara** | **1/504** | **453.8*** |
| QUBO | 128 | 2 | **1/504** | **437.1** | 19/504 | 427.5 | 237/504 | 398.9 | 222/504 | 403.7 | CMAL3 | 2/504 | 435.4 |
| QUBO | 128 | 3 | **1/504** | **1227.2*** | 78/504 | 1177.8 | 257/504 | 1034.7 | 253/504 | 1046.1 | Wiz | 2/504 | 1207.2 |
| QUBO | 128 | 4 | 2/504 | 955.4 | 17/504 | 934.5 | 265/504 | 842.8 | 253/504 | 857.3 | **CMApara** | **1/504** | **964.9*** |
| QUBO | 128 | 5 | **1/504** | **933.3*** | 53/504 | 907.6 | 263/504 | 817.2 | 249/504 | 830.9 | CMAL3 | 2/504 | 928.6 |
| QUBO | 256 | 0 | **1/504** | **1697.7*** | 46/504 | 1570.4 | 199/504 | 1317.4 | 99/504 | 1422.4 | NLOPT_LN_PRAXIS | 2/504 | 1607.1 |
| QUBO | 256 | 1 | **1/504** | **1367.7*** | 3/504 | 1290.5 | 197/504 | 1105.2 | 92/504 | 1197.0 | BigLognormalDiscreteOnePlusOne | 2/504 | 1301.4 |
| QUBO | 256 | 2 | **1/504** | **1304.1*** | 12/504 | 1230.9 | 187/504 | 1073.0 | 92/504 | 1154.4 | SVMMetaModelLogNormal | 2/504 | 1233.8 |
| QUBO | 256 | 3 | **1/504** | **3436.8*** | 53/504 | 3208.6 | 196/504 | 2650.7 | 148/504 | 2854.3 | RLSOnePlusOne | 2/504 | 3316.5 |
| QUBO | 256 | 4 | **1/504** | **2769.0*** | 35/504 | 2597.5 | 208/504 | 2219.0 | 134/504 | 2391.5 | DiscreteLengler2OnePlusOne | 2/504 | 2617.1 |
| QUBO | 256 | 5 | **1/504** | **2730.1*** | 41/504 | 2557.0 | 185/504 | 2206.6 | 141/504 | 2349.2 | SVM1MetaModelLogNormal | 2/504 | 2605.1 |
| NK | 64 | 1 | 29/504 | 0.7103 | 52/504 | 0.7096 | 126/504 | 0.7050 | 236/504 | 0.7008 | **CMApara** | **1/504** | **0.7119** |
| NK | 64 | 2 | 23/504 | 0.742 | 57/504 | 0.7391 | 146/504 | 0.7317 | 204/504 | 0.7273 | **CMApara** | **1/504** | **0.7459** |
| NK | 64 | 4 | 12/504 | 0.7523 | 40/504 | 0.7463 | 146/504 | 0.7330 | 179/504 | 0.7311 | **ChainCMAwithMetaRecenteringsqrt** | **1/504** | **0.756*** |
| NK | 64 | 8 | 18/504 | 0.7379 | 34/504 | 0.7330 | 262/504 | 0.7088 | 308/504 | 0.6932 | **ChainCMAwithLHSsqrt** | **1/504** | **0.749*** |
| NK | 128 | 1 | **1/504** | **0.7100** | 4/504 | 0.7061 | 158/504 | 0.6958 | 206/504 | 0.6941 | CMApara | 2/504 | 0.7074 |
| NK | 128 | 2 | **1/504** | **0.7375*** | 2/504 | 0.7305 | 140/504 | 0.7138 | 128/504 | 0.7139 | CMApara | 3/504 | 0.7304 |
| NK | 128 | 4 | **1/504** | **0.7603*** | 2/504 | 0.7464 | 202/504 | 0.7190 | 124/504 | 0.7252 | MetaModelFmin2 | 3/504 | 0.743 |
| NK | 128 | 8 | **1/504** | **0.7369*** | 2/504 | 0.7266 | 354/504 | 0.6372 | 387/504 | 0.6071 | CMAL3 | 3/504 | 0.7256 |
| NK | 256 | 1 | **1/504** | **0.7071*** | 2/504 | 0.7014 | 111/504 | 0.6810 | 87/504 | 0.6869 | CMApara | 3/504 | 0.6989 |
| NK | 256 | 2 | **1/504** | **0.7364*** | 2/504 | 0.7248 | 98/504 | 0.7004 | 60/504 | 0.7100 | MetaModelFmin2 | 3/504 | 0.7218 |
| NK | 256 | 4 | **1/504** | **0.7534*** | 2/504 | 0.7336 | 104/504 | 0.7006 | 189/504 | 0.6895 | MetaModelFmin2 | 3/504 | 0.7295 |
| NK | 256 | 8 | **1/504** | **0.7232*** | 2/504 | 0.7171 | 384/504 | 0.5798 | 389/504 | 0.5730 | LognormalDiscreteOnePlusOne | 3/504 | 0.7166 |
| NK3 | 64 | 1 | **1/499** | **0.7818*** | - | - | 70/499 | 0.7659 | 115/499 | 0.7635 | DiscreteDE | 2/499 | 0.7772 |
| NK3 | 64 | 2 | **1/499** | **0.8095*** | - | - | 7/499 | 0.7857 | 73/499 | 0.7779 | DiscreteLengler3OnePlusOne | 2/499 | 0.788 |
| NK3 | 64 | 4 | **1/499** | **0.8004*** | - | - | 137/499 | 0.7622 | 153/499 | 0.7570 | BigLognormalDiscreteOnePlusOne | 2/499 | 0.7790 |
| NK3 | 64 | 8 | 62/499 | 0.7473 | - | - | 359/499 | 0.6407 | 357/499 | 0.6420 | **NLOPT_GN_DIRECT_L** | **1/499** | **0.7547*** |
| NK3 | 128 | 1 | **1/499** | **0.7876** | - | - | 62/499 | 0.7599 | 103/499 | 0.7537 | DiscreteLengler3OnePlusOne | 2/499 | 0.7800 |
| NK3 | 128 | 2 | **1/499** | **0.7986*** | - | - | 58/499 | 0.7635 | 111/499 | 0.7527 | BigLognormalDiscreteOnePlusOne | 2/499 | 0.7820 |
| NK3 | 128 | 4 | **1/499** | **0.7847*** | - | - | 123/499 | 0.7374 | 129/499 | 0.7311 | Neural1MetaModelLogNormal | 2/499 | 0.7740 |
| NK3 | 128 | 8 | 62/499 | 0.7373 | - | - | 376/499 | 0.5986 | 344/499 | 0.6008 | **NLOPT_GN_DIRECT_L** | **1/499** | **0.7542*** |
| NK3 | 256 | 1 | **1/499** | **0.7763*** | - | - | 55/499 | 0.7360 | 62/499 | 0.7247 | NGOpt | 2/499 | 0.7542 |
| NK3 | 256 | 2 | **1/499** | **0.7801*** | - | - | 53/499 | 0.7391 | 69/499 | 0.7236 | RF1MetaModelLogNormal | 2/499 | 0.7600 |
| NK3 | 256 | 4 | **1/499** | **0.7615*** | - | - | 147/499 | 0.6784 | 69/499 | 0.7091 | SVM1MetaModelLogNormal | 2/499 | 0.7522 |
| NK3 | 256 | 8 | 43/499 | 0.7213 | - | - | 361/499 | 0.5704 | 401/499 | 0.5692 | **RLSOnePlusOne** | **1/499** | **0.7362*** |

$K = 3$ (Figure 6d), the 10 other best algorithms, which are variants of the meta-algorithm NGOpt, exhibit overlapping performance curves. This is because they all selected the same low-level algorithm, DiscreteLenglerOnePlusOne, based on the characteristics of the instance.

When comparing the evolution curves of $(\sigma, \sigma)$-RL-EDA across these two problems, we observe markedly different behaviors. For QUBO problems (Figure 6), $(\sigma, \sigma)$-RL-EDA quickly reaches a good solution and then stagnates for the remainder of the budget. The best scores are typically achieved after approximately 3,000 to 4,000 evaluations, suggesting that the full budget of 10,000 does not benefit $(\sigma, \sigma)$-RL-EDA, but rather favors competing algorithms.

In contrast, for NK3 instances (Figure 7), $(\sigma, \sigma)$-RL-EDA requires significantly more time to converge. The algorithm exhibits an "S"-shaped curve, indicating a delayed learning phase before generating high-quality solutions. This behavior becomes more pronounced as the as the interaction degree increases (i.e., with higher $K$ values), likely due to the increased difficulty in modeling variable interactions in NK3 compared to QUBO. Notably, for the most complex instances ($K = 8$), $(\sigma, \sigma)$-RL-EDA fails to converge within the allocated budget, explaining its poor performance re-

ported in Table 4 for this distribution. Meta-algorithms from the Nevergrad library that incorporate neural networks (NeuralMetaModelLogNormal) or random forests (NRFMetaModelLogNormal) achieve good results more rapidly. On the other hand, when $(\sigma, \sigma)$-RL-EDA has sufficient time to converge—as in landscapes with $K = 2$ or $K = 4$—it achieves significantly better average scores than its competitors by the end of the search.

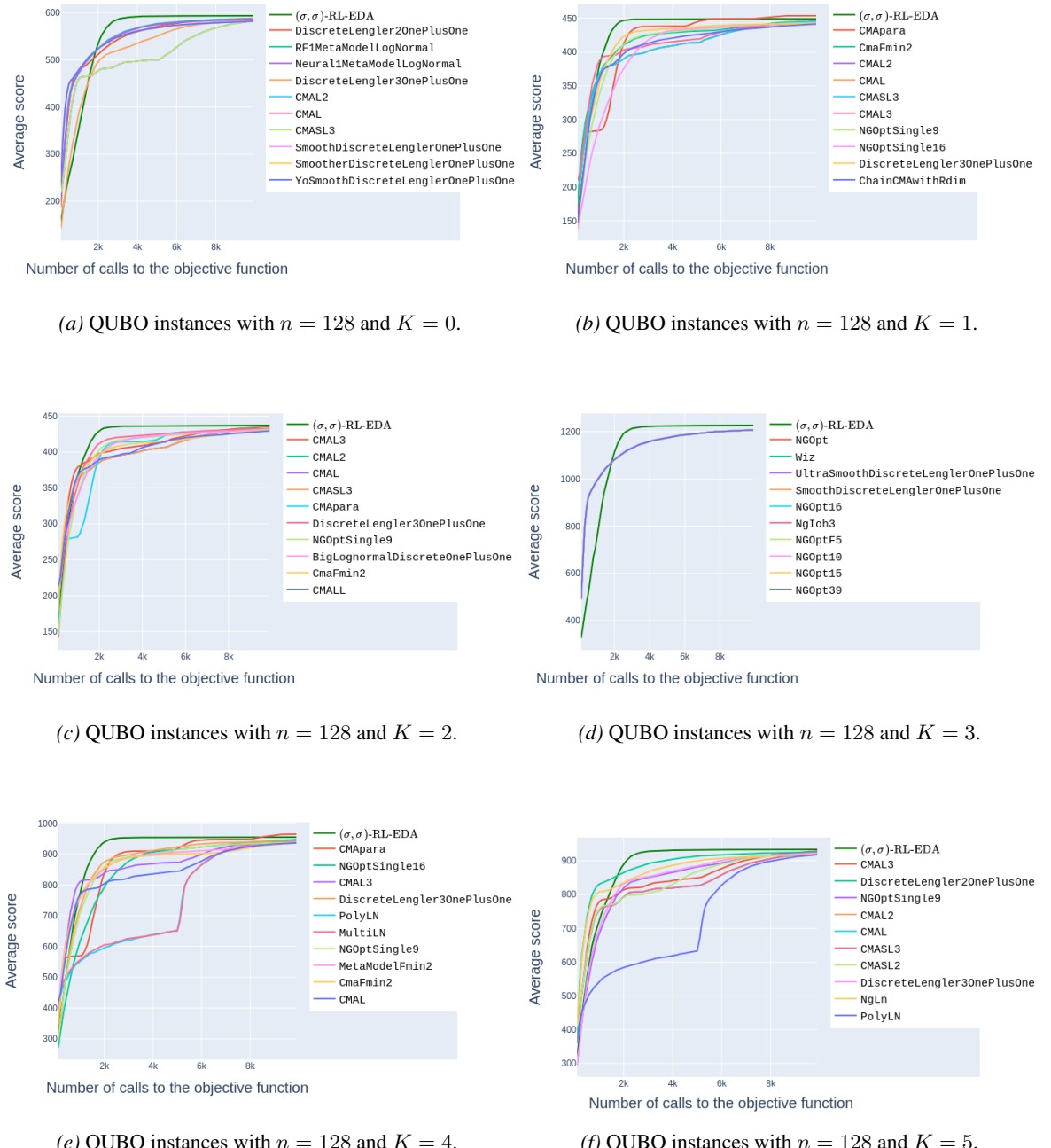

*(a)* QUBO instances with $n = 128$ and $K = 0$.

*(b)* QUBO instances with $n = 128$ and $K = 1$.

*(c)* QUBO instances with $n = 128$ and $K = 2$.

*(d)* QUBO instances with $n = 128$ and $K = 3$.

*(e)* QUBO instances with $n = 128$ and $K = 4$.

*(f)* QUBO instances with $n = 128$ and $K = 5$.

*Figure 6.* Evolution of the average scores with respect to the number of calls to the objective function obtained by $(\sigma, \sigma)$-RL-EDA and the best 10 other competitors for the different types of QUBO instances with $n = 128$.

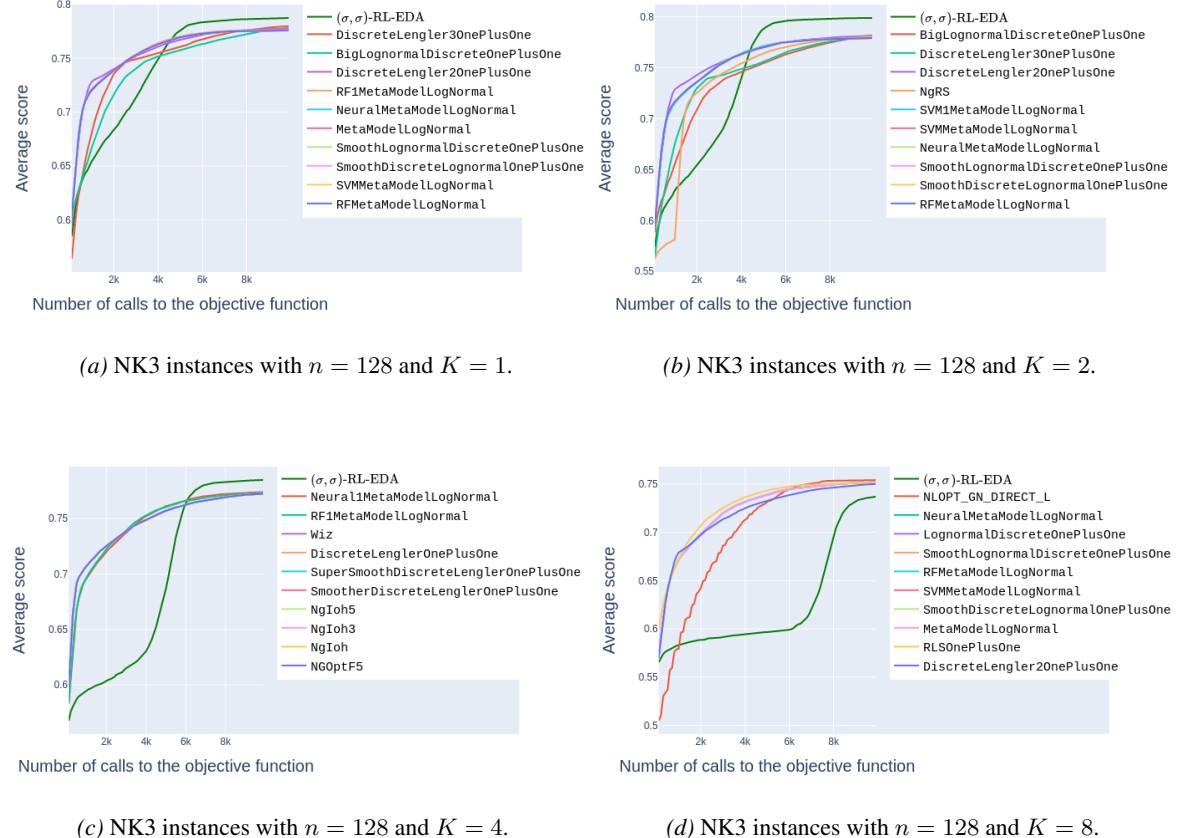

*(a)* NK3 instances with $n = 128$ and $K = 1$.

*(b)* NK3 instances with $n = 128$ and $K = 2$.

*(c)* NK3 instances with $n = 128$ and $K = 4$.

*(d)* NK3 instances with $n = 128$ and $K = 8$.

*Figure 7.* Evolution of the average scores with respect to the number of calls to the objective function obtained by $(\sigma, \sigma)$-RL-EDA and the best 10 other competitors for the different types of NK3 instances with $n = 128$.

## M. Ablation Studies and Sensitivity Analyses

In this appendix, we first present two ablation studies aimed at evaluating the impact of the order-invariant reinforcement learning framework used in $(\sigma, \sigma)$-RL-EDA (see Section 3.3), which could be partially or fully replaced by naive structural dropout during sampling and/or training.

We also investigate the influence of incorporating a known variable interaction graph on the performance of $(\sigma, \sigma)$-RL-EDA.

Furthermore, we conduct a sensitivity analysis of key parameters within the multivariate RL EDA framework, specifically examining the effects of the population size ($\lambda$), the KL divergence penalization coefficient ($\beta$), various configurations of the $g$ mechanisms employed in the multivariate generative model, the number of training epochs $E$ at each iteration $t$ of the EDA, and the learning rate $l_r$ of Adam optimizer.

### M.1. Impact of Using Additional Structural Dropout for Generation and Training

In this appendix, we aim to test variants of the multivariate RL EDA presented in Section 3.3 ($(\delta, \delta)$-RL-EDA, $(\delta, \sigma)$-RL-EDA, $(\sigma, \delta)$-RL-EDA, $(\sigma, \sigma)$-RL-EDA), augmented with additional structural dropout for sampling and training (following the objective (34) combining input dropout and order permutations described in section E).

During the generation phase (respectively the training phase) of the EDA, we add a probability $p_G \in \{0.0, 0.25, 0.5, 0.75\}$ (respectively $p_T \in \{0.0, 0.25, 0.5, 0.75\}$) that a parent variable in the causal mask is set to zero. Therefore, we test 16 different configurations of structural dropout for each multivariate RL variant.

First, we see in Figure 9a that adding structural dropout during the sampling phase and the training phase can be very

beneficial in particular for the variant $(\delta, \delta)$-RL-EDA with a fixed order for both generation and sampling. It helps the model have more diversity during the generation phase of the EDA and to better capture dependencies between variables during the update phase.

By contrast, adding these structural dropouts for the variant $(\sigma, \sigma)$-RL-EDA in Figure 9d does not improve the results compared to the reference version with $p_G = 0.0$ and $p_T = 0.0$ (green solid line), because this version already benefits from structural dropout for sampling and training induced by its double random order sampling process.

Overall, we observe that the reference version $(\sigma, \sigma)$-RL-EDA without structural dropout performs better, with an average score of 0.753, than all variants across the different combinations of dropout levels used for sampling and training (the best other variant obtains an average score of 0.747). The difference in score is statistically significant according to a t-test with p-value 0.001. It should be noted that it is difficult to achieve an average score that is 0.006 higher, given that the score is already very good for this type of instance. This suggests that the dropout distribution induced by double-order sampling is more advantageous than fine-tuning specific structural dropout values for the generation and update phases of the EDA.

We confirm this results on the large QUBO instances with $n = 256$ and $K = 5$ (see Figure 9. On this distribution of instances our reference variant $(\sigma, \sigma)$-RL-EDA with $p_G = 0.0$ and $p_T = 0.0$ (green solid line in SubFigure 8d) obtains a score of 2730 on average, while the best other variant $(\sigma, \delta)$-RL-EDA with dropout ratios $p_G = 0.5$ and $p_T = 0.5$ obtains a score of 2709 on average. The difference in score is statistically significant according to a t-test with p-value 0.1.

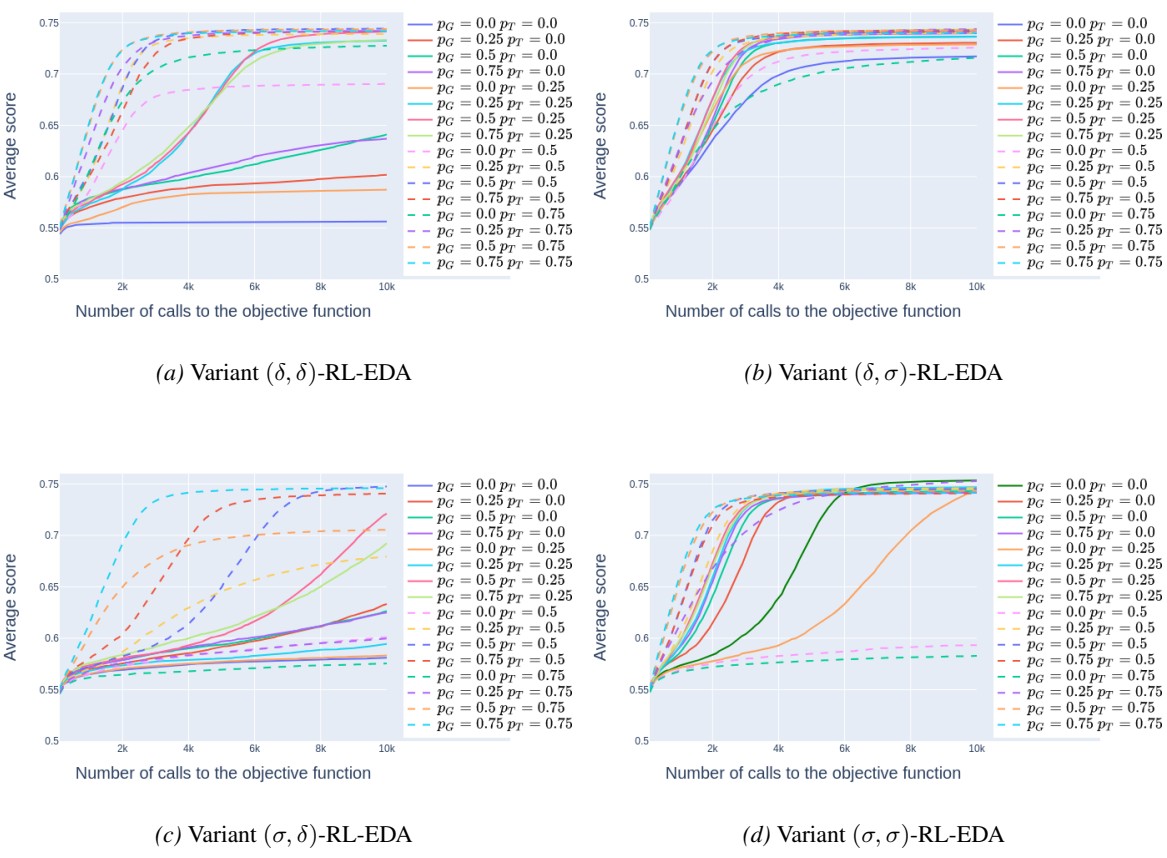

*(a) Variant $(\delta, \delta)$-RL-EDA*

*(b) Variant $(\delta, \sigma)$-RL-EDA*

*(c) Variant $(\sigma, \delta)$-RL-EDA*

*(d) Variant $(\sigma, \sigma)$-RL-EDA*

*Figure 8.* Evolution of the average scores with respect to the number of calls to the objective function, obtained by the four different versions of the multivariate RL EDA with additional structural dropout for sampling and training for the instances of the NK landscape problem with $n = 256$ and $K = 4$.

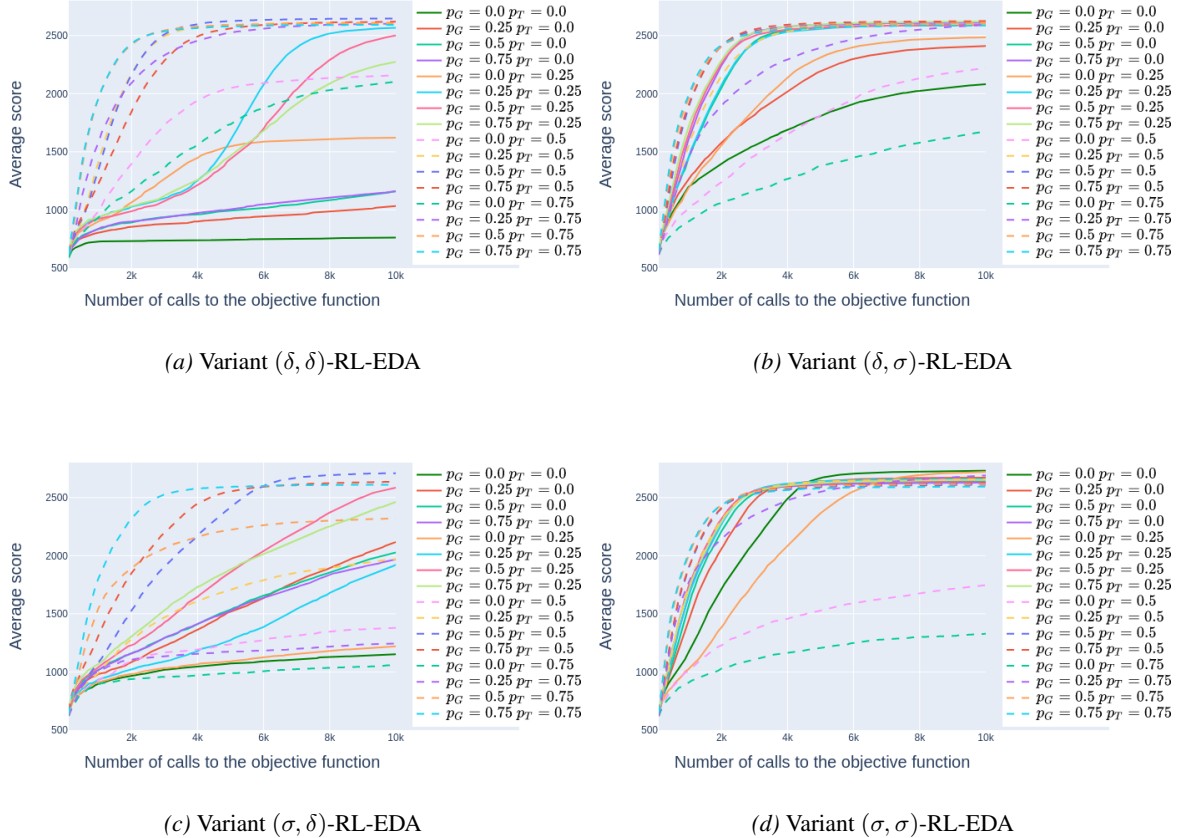

*(a)* Variant $(\delta, \delta)$-RL-EDA

*(b)* Variant $(\delta, \sigma)$-RL-EDA

*(c)* Variant $(\sigma, \delta)$-RL-EDA

*(d)* Variant $(\sigma, \sigma)$-RL-EDA

*Figure 9.* Evolution of the average scores with respect to the number of calls to the objective function, obtained by the four different versions of the multivariate RL EDA with additional structural dropout for sampling and training for the instances of the QUBO problem with $n = 256$ and $K = 5$.

## M.2. Impact of Using Structural Dropout Instead of Causal Mask During Training

In this appendix, we seek to verify whether the causal mask used during the EDA training phase can be completely replaced by a structural dropout with a probability $p_T \in \{0.0, 0.25, 0.5, 0.75\}$ for variants with fixed or random orders during generation. These variants without causal mask during training are called $(\delta, p)$-RL-EDA and $(\sigma, p)$-RL-EDA. We also retain the different structural dropout ratios for generation $p_G \in \{0.0, 0.25, 0.5, 0.75\}$ which is complementary to the mandatory causal mask for generation.

We observe in Figure 10a that the variant $(\delta, p)$-RL-EDA can achieve at best the same results than the variant $(\delta, \sigma)$-RL-EDA using fix causal mask during training (see Figure 9a). Symmetrically, the variant $(\sigma, p)$-RL-EDA obtain also at best the same results than the variant $(\sigma, \delta)$-RL-EDA (see Figure 9c). However these variants obtain worse results than the reference version $(\sigma, \sigma)$-RL-EDA (green solid line in Figure 9d), which confirms the utility of the specific double uniform distribution of random orders used during the sampling and training phase of the EDA, instead of finely tuned structural dropout rates in this context. We confirm this results on large QUBO instances with $n = 256$ and $K = 5$ (see Figure 11), when comparing the results obtain on these plots with those obtain by the reference version $(\sigma, \sigma)$-RL-EDA on the same distribution of instances (green solid line in Figure 8d).

## M.3. Impact of Using a Known Interaction Graph Between Variables

In scenarios where the interaction graph (IG) between variables is assumed to be known—i.e., a gray-box setting (Santana, 2017) —the causal masks used in $(\sigma, \sigma)$-RL-EDA can be adapted to respect these structural constraints.

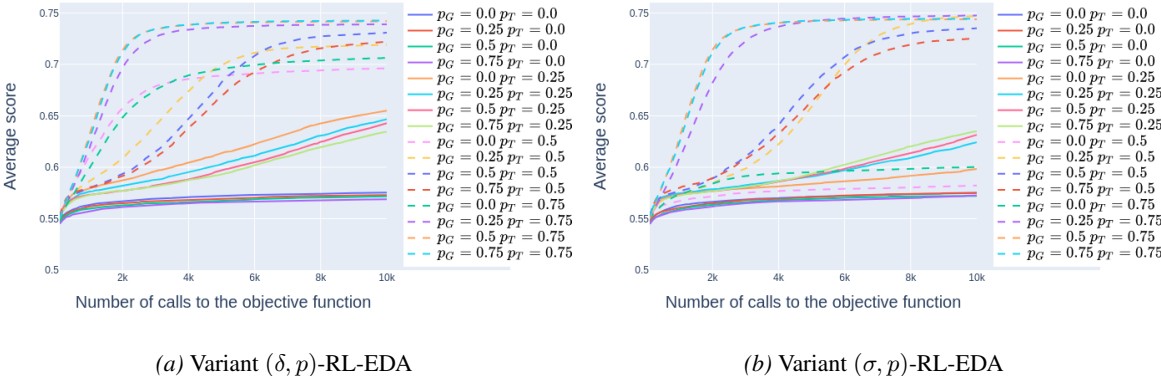

*(a) Variant $(\delta, p)$-RL-EDA*  *(b) Variant $(\sigma, p)$-RL-EDA*

*Figure 10.* Evolution of the average scores with respect to the number of calls to the objective function for the variants $(\delta, p)$-RL-EDA and $(\sigma, p)$-RL-EDA for the instances of the NK landscape problem with $n = 256$ and $K = 4$.

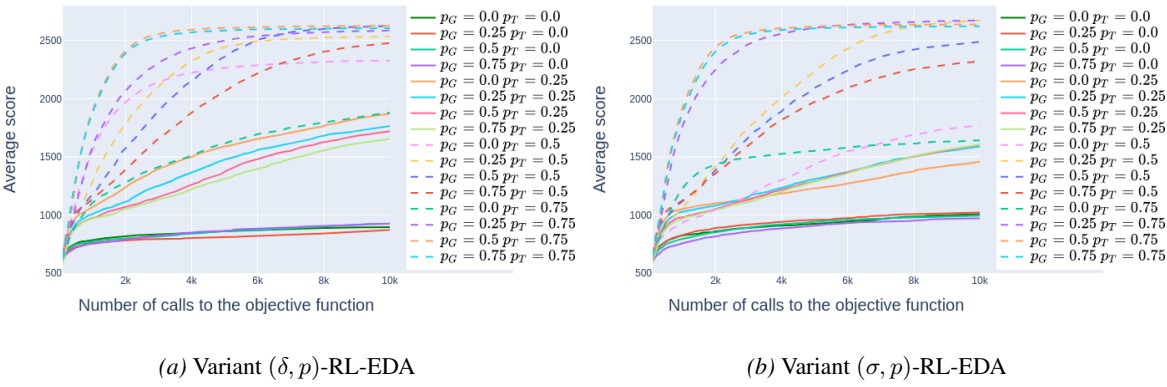

*(a) Variant $(\delta, p)$-RL-EDA*  *(b) Variant $(\sigma, p)$-RL-EDA*

*Figure 11.* Evolution of the average scores with respect to the number of calls to the objective function for the variants $(\delta, p)$-RL-EDA and $(\sigma, p)$-RL-EDA for the instances of the QUBO problem with $n = 256$ and $K = 5$.

Let $A$ denote the symmetric binary adjacency matrix of the interaction graph, where $a_{i,j} = 1$ indicates that variables $X_i$ and $X_j$ interact in the evaluation of the objective function $f$. For example, in the QUBO problem, the objective function is defined as $f(x) = x^\top Q x$, where $Q$ is a symmetric real matrix of size $n \times n$ and coefficients $q_{ij}$. In this case, the adjacency matrix $A$ is constructed such that $a_{ij} = 1$ if $q_{ij} \neq 0$, and 0 otherwise.

Each causal mask $\sigma(x)_{<k}$ (see Section 3.3) is then adapted to hide values of non-adjacent variables in the interaction graph (corresponding to zero coefficients in the adjacency matrix $A$), in addition to every dimension whose rank in $\sigma$ is greater or equal than $k$.

Figure 12 shows the evolution of average scores across 100 independent runs of $(\sigma, \sigma)$-RL-EDA, comparing the case with an unknown IG (green curve) to the case with a known IG (blue curve). When comparing the green and blue curves, we observe that providing the interaction graph between variables helps guide the algorithm more effectively at the beginning of the search. Indeed, $(\sigma, \sigma)$-RL-EDA with a known IG reaches high-quality solutions more rapidly. However, it is noteworthy that the green curve eventually outperforms the blue curve, suggesting that constraining the learning process to the predefined interaction graph may become limiting. Toward the end of the search, generating optimal solutions may benefit from discovering new relationships between variables that are not encoded in the known interaction graph used to compute the objective function. This phenomenon can be attributed to the fact that the learned model of $(\sigma, \sigma)$-RL-EDA is not designed to model the full objective function, but rather to approximate the distribution of high-quality solutions within a specific region of the search space.

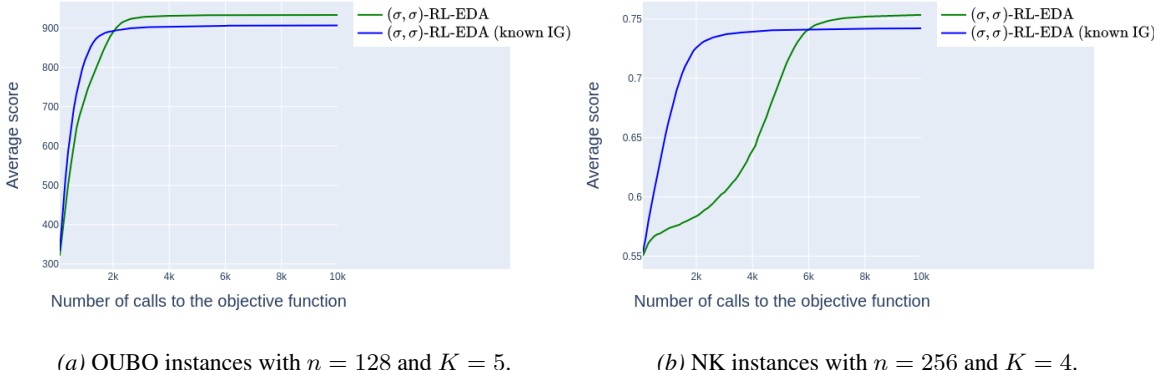

*(a)* QUBO instances with $n = 128$ and $K = 5$.      *(b)* NK instances with $n = 256$ and $K = 4$.

*Figure 12.* Evolution of the average scores with respect to the number of calls to the objective function, obtained by $(\sigma, \sigma)$-RL-EDA with and without known interaction graph.

## M.4. Sensitivity to the Population Size

Figure 13 shows the score evolution curves for $(\sigma, \sigma)$-RL-EDA with different population sizes.

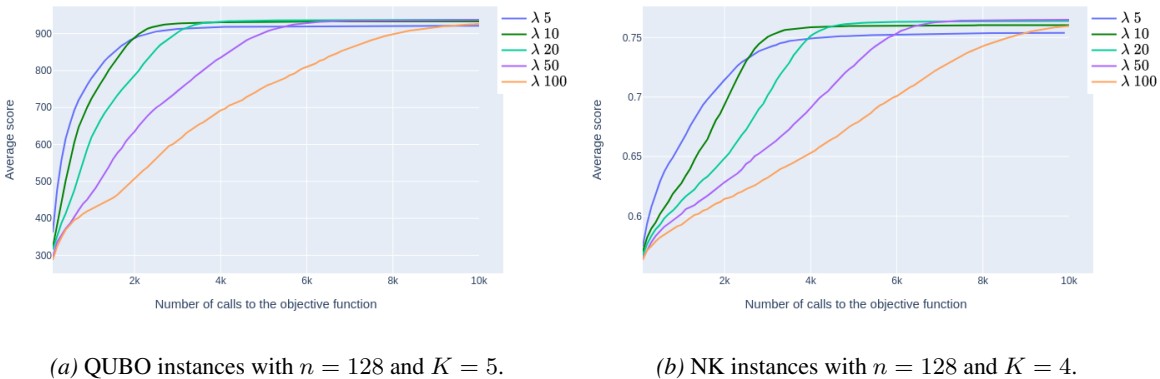

*(a)* QUBO instances with $n = 128$ and $K = 5$.      *(b)* NK instances with $n = 128$ and $K = 4$.

*Figure 13.* Sensitivity to the population size in $(\sigma, \sigma)$-RL-EDA.

Our analysis reveals that, for the considered instance distributions, a smaller population size tends to promote faster convergence in terms of the number of objective function evaluations. However, this accelerated convergence often comes at the expense of reduced exploration, which can lead the algorithm to suboptimal local solutions. Increasing the population size to $\lambda = 20$ or $\lambda = 50$ improves the average performance previously reported for NK instances with $n = 128$ and $K = 4$ (Figure 13b). In contrast, as shown in Figure 13a, the population size appears to have a negligible impact on performance for QUBO instances.

## M.5. Sensitivity to the KL Penalty Coefficient

Figure 14 shows the score evolution curves of $(\sigma, \sigma)$-RL-EDA for different values of the KL penalty coefficient $\beta$. By default, this coefficient is set to 1 in $(\sigma, \sigma)$-RL-EDA (green curve). It controls the amplitude of the KL regularization term included in the objective function during the update phase of $(\sigma, \sigma)$-RL-EDA (see Equation 8).

We observe that low values of $\beta$ lead to faster convergence in terms of objective function evaluations. However, this often results in premature convergence to suboptimal solutions due to insufficient exploration. Conversely, higher values of $\beta$ help maintain the initial high entropy of the solution distribution for a longer period, thereby promoting broader

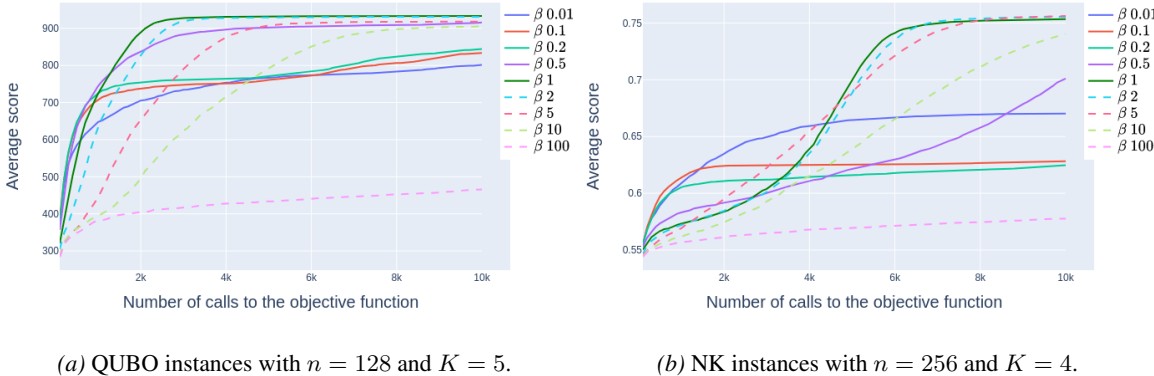

*(a)* QUBO instances with $n = 128$ and $K = 5$.  *(b)* NK instances with $n = 256$ and $K = 4$.

*Figure 14.* Sensitivity to the KL penalty coefficient $\beta$ in $(\sigma, \sigma)$-`RL-EDA`.

exploration. Nevertheless, excessively high values—such as $\beta = 100$—can hinder the algorithm's ability to converge toward high-quality solutions. These results highlight the critical role of $\beta$ in balancing exploration and exploitation. For the instance distributions considered and given the evaluation budget, setting $\beta$ within the range $[1, 5]$ appears to offer a satisfactory trade-off.

### M.6. Sensitivity to the Logistic Regression Models used in the Markov Kernels

Figure 15 shows the score evolution of $(\sigma, \sigma)$-`RL-EDA` for different logistic regression models $g$ used in the generative process of each variable conditioned on the others (see Section 3.1).

The blue curve corresponds to the univariate model, where each variable is generated independently of the others. This model converges the fastest, due to its limited number of parameters. The red curve represents the use of linear logistic regression models. Interestingly, the performance obtained with linear models is even lower than that of the univariate model. This result suggests that it may be preferable to omit interaction modeling entirely rather than attempt to capture complex dependencies using an overly simplistic linear model.

We also evaluate several variants using neural networks of varying depth—specifically with 1, 2, and 4 hidden layers—for each variable. All configurations perform similarly on NK instances with $K = 4$ (Figure 15a), where variable interactions are relatively simple. However, for the more complex categorical NK3 problem with $K = 8$ (Figure 15b), deeper architectures (e.g., the four-hidden-layer model, shown by the orange curve) outperform simpler ones such as the single hidden layer (green curve). This suggests that increased model capacity is beneficial for capturing more complex dependencies. Nevertheless, this improvement comes with increased computational and memory requirements.

### M.7. Sensitivity to the Number of Training Epochs at each Generation

Figure 16 shows the score evolution curves of $(\sigma, \sigma)$-`RL-EDA` for different values of the number of training epochs $E$ (number of permutations) at each iteration $t$ of the RL EDA. By default, this coefficient is set to 50 in $(\sigma, \sigma)$-`RL-EDA` (green curve).

In Figure 16, we observe that the higher the value of the parameter $E$, the better the long-term results. However, increasing $E$ increases the algorithm's runtime, which is 5min, 6min, 8min, 11min, 20min, 35min to process 100 instances of size $n = 128$ on a V100 GPU card when $E$ is equal to 1, 5, 10, 20, 50 and 100 respectively.

### M.8. Sensitivity to the Learning Rate

Figure 17 illustrates the evolution of the average score obtained by $(\sigma, \sigma)$-`RL-EDA` for different values of the learning rate $l_r$ used in the Adam optimizer to update the RL-EDA model at each generation. By default, $l_r$ is set to $10^{-3}$ (green curve). As shown in Figure 17, standard Adam learning rates, namely $10^{-3}$ and $10^{-4}$, yield the best performance across both NK and QUBO problem instances. In contrast, excessively small or large learning rates lead to degraded performance.

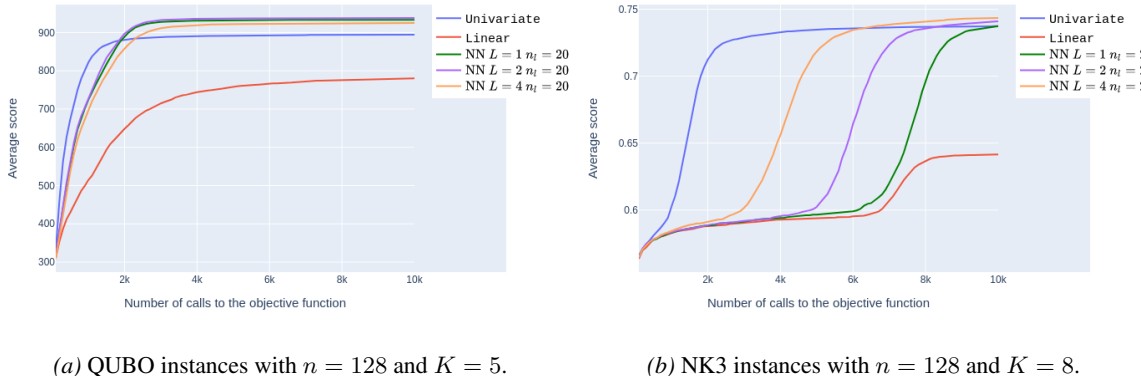

*(a)* QUBO instances with $n = 128$ and $K = 5$.

*(b)* NK3 instances with $n = 128$ and $K = 8$.

*Figure 15.* Sensitivity to the logistic regression models used in each conditional generative network of $(\sigma, \sigma)-\text{RL-EDA}$. NN corresponds to neural network. $L$ is the number of hidden layer in each neural network and $n_l$ is the number of neurons in each hidden layer.

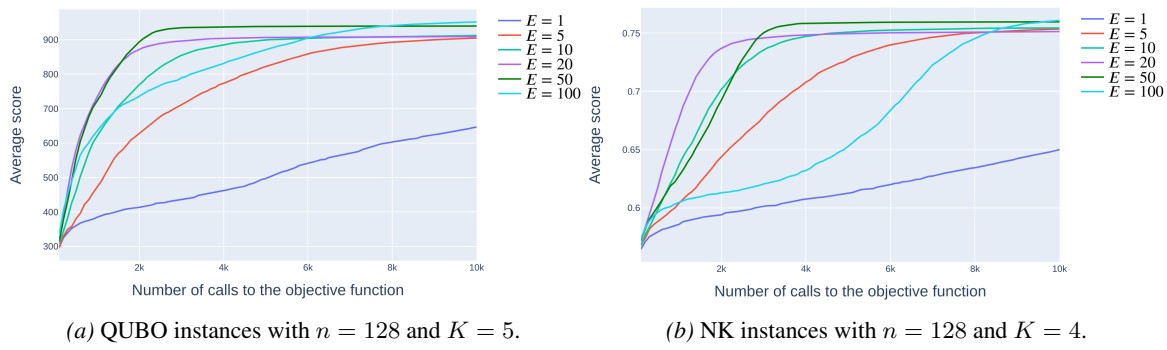

*(a)* QUBO instances with $n = 128$ and $K = 5$.

*(b)* NK instances with $n = 128$ and $K = 4$.

*Figure 16.* Sensitivity to the number $E$ of training epochs at each generation

.

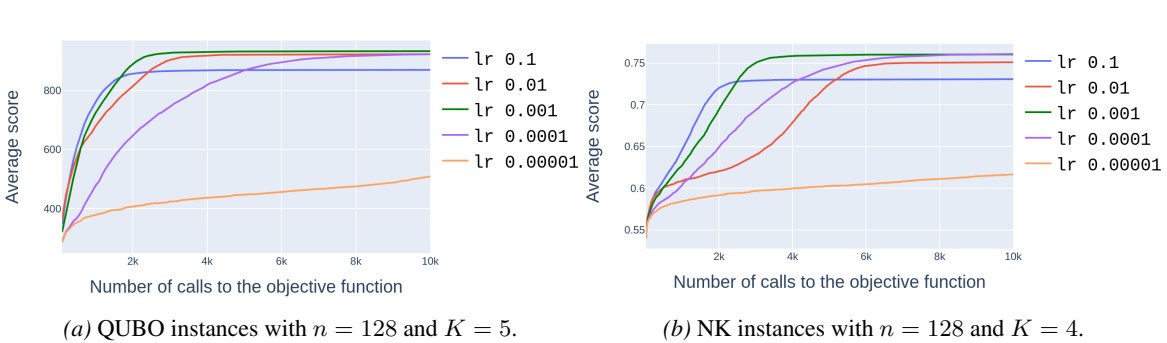

*(a)* QUBO instances with $n = 128$ and $K = 5$.

*(b)* NK instances with $n = 128$ and $K = 4$.

*Figure 17.* Sensitivity of $(\sigma, \sigma)-\text{RL-EDA}$ to the learning rate $l_r$ of the Adam optimizer.

.

# N. Adaptive KL Penalty Coefficient According to Landscape Ruggedness

Empirically, we observe that tuning the parameter $\beta$, which controls the KL regularization term in $(\sigma, \sigma)-\text{RL-EDA}$, on a per-instance basis significantly impacts performance. In particular, smaller values of $\beta$ accelerate convergence on *smooth* landscapes, whereas larger values help avoid premature convergence to poor local optima on *rugged* landscapes.

To capture this phenomenon, we introduce a *ruggedness criterion* $R$, defined as the Pearson correlation between Hamming distances and absolute fitness differences computed over an archive of solutions gathered during the last ten iterations. Intuitively, high correlation corresponds to smoother landscapes, while low correlation indicates increased ruggedness.

Based on this measure, we propose an adaptive variant of $(\sigma, \sigma)$-`RL-EDA`, where $\beta$ is updated every 10 generations according to

$$\beta \leftarrow \frac{0.12}{R + 10^{-7}}.$$

As reported in Table 5, this adaptive scheme consistently improves performance on QUBO instances while maintaining stable behavior on NK benchmarks when all other hyperparameters are kept fixed.

*Table 5.* Performance comparison between baseline and adaptive $\beta$. We report per-instance results along with averages and relative improvements ($\Delta$). Relative improvements are computed as (Adaptive − Baseline)/Baseline. Higher is better.

| Instances | | | Baseline | Adaptive | $\Delta(\%)$ |
|---|---|---|---|---|---|
| QUBO | 64 | 0 | 201 | 206 | +2.49 |
| QUBO | 64 | 1 | 148 | 150 | +1.35 |
| QUBO | 64 | 2 | 138 | 141 | +2.17 |
| QUBO | 64 | 3 | 410 | 430 | +4.88 |
| QUBO | 64 | 4 | 326 | 335 | +2.76 |
| QUBO | 64 | 5 | 309 | 316 | +2.27 |
| Avg | | | 255.33 | 263.00 | +3.00 |
| QUBO | 128 | 0 | 594 | 596 | +0.34 |
| QUBO | 128 | 1 | 449 | 449 | +0.00 |
| QUBO | 128 | 2 | 437 | 437 | +0.00 |
| QUBO | 128 | 3 | 1227 | 1240 | +1.06 |
| QUBO | 128 | 4 | 955 | 960 | +0.52 |
| QUBO | 128 | 5 | 933 | 942 | +0.96 |
| Avg | | | 765.83 | 770.67 | +0.63 |
| NK | 64 | 1 | 0.71 | 0.71 | 0.00 |
| NK | 64 | 2 | 0.74 | 0.74 | 0.00 |
| NK | 64 | 4 | 0.75 | 0.76 | +1.33 |
| NK | 64 | 8 | 0.74 | 0.75 | +1.35 |
| Avg | | | 0.735 | 0.740 | +0.68 |
| NK | 128 | 1 | 0.71 | 0.70 | -1.41 |
| NK | 128 | 2 | 0.73 | 0.73 | 0.00 |
| NK | 128 | 4 | 0.76 | 0.76 | 0.00 |
| NK | 128 | 8 | 0.74 | 0.74 | 0.00 |
| Avg | | | 0.735 | 0.733 | -0.34 |

# O. Results on NAS-Bench-101 Real Dataset

The neural architecture search public dataset (Ying et al., 2019) (full NAS-Bench-101 available at `https://github.com/google-research/nasbench`), is a table that maps neural network architectures to their testing metrics. Each architecture is encoded by a set of 26 variables: 21 binary variables and 5 categorical variables, each of which can take one of three values. The resulting search space is therefore $\mathcal{X} = \{0,1\}^{21} \times \{0,1,2\}^5$, with $|\mathcal{X}| \approx 510$ million. However, as noted in (Ying et al., 2019), a substantial fraction of these configurations correspond to invalid models, which are assigned a testing accuracy of 0 in the dataset. The number of valid architectures, those with an accuracy strictly greater than 0, amounts to 423,000. The goal of this benchmark is to find architectures with high test accuracy.

On this benchmark, we run $(\sigma, \sigma)$-`RL-EDA` with the same hyperparameters used for the synthetic datasets, and compare it with the same competitors as described in the previous subsection, with the same maximum budget of 10,000 calls to the objective function. When solving the problem with the Nevergrad library, it is parametrized with unordered categorical variables using *choice* parameters. Figure 18 displays the evolution of the average best accuracy of $(\sigma, \sigma)$-`RL-EDA` in comparison with the 10 best competing baselines, with respect to the number of calls to the objective function. The evolution curves are averaged over 100 independent runs.

The $(\sigma, \sigma)$-`RL-EDA` algorithm (green line) achieves the best performance both with a budget of 10,000 objective-function

evaluations and under a shorter budget of only 1,000 evaluations.

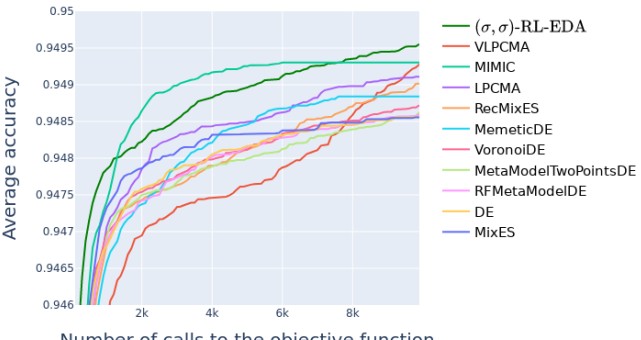

*Figure 18.* NAS-Bench-101 benchmark. X-axis: number of calls to the objective function. Y-axis: Evolution of the average accuracy of the architectures.

## P. Early-budget Behavior and Adaptation of the Algorithm in this Context

Table 6 shows the results of the same experiments as those conducted in Section 4.2 whose full results are reported in Appendix L, except that the scores are reported after only 1,000 calls to the objective function (i.e., for a small budget) instead of 10,000. In this case, we observe that our algorithm $(\sigma, \sigma)$-RL-EDA achieves strong performance for small binary instances of size 64. However, for more complex instances, larger in size or with more categories, even though it often obtains the best scores after 10,000 evaluations, as shown in Figure 3 and in Table 4 in Appendix L, it takes longer to converge than other methods, which explains the poorer results observed in this table 6. This is because our EDAs maintain a high degree of diversity in the population at the start of the search, so as not to get stuck too quickly in a local optimum, as we can see in Figure 2. In this regard, we can see that the other EDAs also behave in the same way, as the multivariate EDAs MIMIC and BOA also see their scores deteriorate when the number of iterations is very low. A certain number of evaluations are also required for this type of multivariate model in order to properly learn the complex interactions between variables.

**Curriculum Adaptation**    To overcome this problem, we propose adapting the algorithm with a curriculum approach, so that the model is univariate at the start of the search to quickly find high-quality solutions at the beginning, then gradually switches to a multivariate mode. To do this, we revisit the idea of structured dropout introduced in Appendix M.1. At the very beginning of the search, the dropout probabilities $p_G^0$ and $p_T^0$ are set to 1, which means that all input variables of all networks are masked, and therefore the model is completely univariate. Then we introduce a coefficient $\rho < 1$ that multiplies these probabilities at each iteration $t$ of the EDA, with the equations $p_G^{t+1} = \rho \times p_G^t$ and $p_T^{t+1} = \rho \times p_T^t$, so as to decrease them during the search. When the iteration index $t$ tends towards infinity, $p_G^t$ and $p_T^t$ both tend towards 0, which makes the algorithm revert to the standard multivariate model $(\sigma, \sigma)$-RL-EDA. In the following figures, we therefore propose a sensitivity analysis for this coefficient $\rho$ for NK and QUBO instances of size 128. We also set $\lambda = 5$ and $E = 100$ instead $\lambda = 10$ and $E = 50$ in order to accelerate convergence of the algorithm.

In Figure 19, we observe that when $\rho$ is close to 1, i.e., when the model is close to the univariate model, the network converges quickly at the beginning but reaches lower final scores. Conversely, a lower value of $\rho$ leads to lower scores at the beginning, but these scores end up being higher at the end of the search, because the model learns without loss of information of the context of the full joint distribution, generating higher-quality solutions.

Using a value of $\rho = 0.993$ seems to be a good compromise for obtaining high-quality solutions quickly, as well as good results when the model converges.

However, in order to create an effective version when the budget is limited, and knowing that some of Nevergrad's competitors' algorithms are specially optimized for this purpose, we create a version called Fast-$(\sigma, \sigma)$-RL-EDA with $\rho = 0.999$ and run it with a budget of 1,000 calls to the objective function. Table 7 shows the results obtained by this variant Fast-$(\sigma, \sigma)$-RL-EDA in comparison with the other methods. We observe in this Table that the version Fast-$(\sigma, \sigma)$-RL-EDA frequently obtains the best results for instances of size $n = 64$ and $n = 128$ for all types of instance distributions, and results close to those obtained by the best competitors for instances of size 256.

*Table 6.* Global rankings and average scores obtained by $(\sigma, \sigma)$-RL-EDA and the other EDAs (PBIL, MIMIC, and BOA) are reported. The last columns present the ranking and average score of the best-performing method among the 500 additional algorithms considered (496 for NK3 problems). Rankings are computed over all 504 algorithms (499 for NK3 problems) by comparing the best score achieved **with short budget after 1,000 objective function evaluations**, averaged across 100 independent runs. Bold values highlight the best results among all competing methods. A star associated with the results obtained by $(\sigma, \sigma)$-RL-EDA indicates that it is significantly better on average (over 100 runs) than the best other competitor. A star associated with a result obtained by another algorithm indicates that it is significantly better on average (over 100 runs) than $(\sigma, \sigma)$-RL-EDA. A difference in average scores is said statistically significant according to a t-test with p-value 0.001.

| Instances | | | Methods | | | | | | | | | | |
|---|---|---|---|---|---|---|---|---|---|---|---|---|---|
| | | | $(\sigma, \sigma)$-RL-EDA | | PBIL | | MIMIC | | BOA | | Best method (others) | | | |
| Pb | $n$ | $K$ | Rank | Score | Rank | Score | Rank | Score | Rank | Score | Name | Rank | Score |
| QUBO | 64 | 0 | **1/504** | **195.9\*** | 243/504 | 159.7 | 336/504 | 139.0 | 362/504 | 117.9 | Carola4 | 2/504 | 187.4 |
| QUBO | 64 | 1 | **1/504** | **147.7\*** | 127/504 | 137.3 | 225/504 | 129.6 | 343/504 | 113.8 | LargeCMA | 2/504 | 144.7 |
| QUBO | 64 | 2 | **1/504** | **136.6** | 123/504 | 131.3 | 191/504 | 127.5 | 345/504 | 114.5 | LargeCMA | 2/504 | 136.4 |
| QUBO | 64 | 3 | **1/504** | **394.8\*** | 275/504 | 318.9 | 346/504 | 271.2 | 359/504 | 234.9 | Carola4 | 2/504 | 378.6 |
| QUBO | 64 | 4 | **1/504** | **324.3\*** | 174/504 | 287.3 | 306/504 | 261.1 | 352/504 | 230.7 | FCarola6 | 2/504 | 314.2 |
| QUBO | 64 | 5 | **1/504** | **304.3\*** | 197/504 | 266.7 | 282/504 | 252.0 | 353/504 | 226.4 | NgLglr | 2/504 | 292.5 |
| QUBO | 128 | 0 | 251/504 | 354.7 | 315/504 | 327.0 | 355/504 | 248.0 | 366/504 | 224.2 | **NLOPT_LN_PRAXIS** | **1/504** | **517.2\*** |
| QUBO | 128 | 1 | 81/504 | 381.8 | 246/504 | 316.7 | 340/504 | 266.5 | 362/504 | 223.3 | **DiscreteLengler2OnePlusOne** | **1/504** | **406.1\*** |
| QUBO | 128 | 2 | 80/504 | 367.0 | 246/504 | 319.7 | 340/504 | 269.6 | 361/504 | 229.7 | **NgIohLn** | **1/504** | **399.3\*** |
| QUBO | 128 | 3 | 249/504 | 749.54 | 320/504 | 661.4 | 354/504 | 506.6 | 368/504 | 442.7 | **NLOPT_LN_PRAXIS** | **1/504** | **1034.5\*** |
| QUBO | 128 | 4 | 98/504 | 775.9 | 257/504 | 645.1 | 361/504 | 448.5 | 254/504 | 857.3 | **DiscreteLengler2OnePlusOne** | **1/504** | **845.8\*** |
| QUBO | 128 | 5 | 179/504 | 724.3 | 246/504 | 629.9 | 346/504 | 523.8 | 365/504 | 440.8 | **DiscreteLengler2OnePlusOne** | **1/504** | **830.8\*** |
| QUBO | 256 | 0 | 359/504 | 491.5 | 326/504 | 623.8 | 364/504 | 460.1 | 369/504 | 418.6 | **Carola1** | **1/504** | **1365.2\*** |
| QUBO | 256 | 1 | 328/504 | 599.0 | 278/504 | 648.7 | 358/504 | 485.2 | 367/504 | 439.0 | **NLOPT_LN_PRAXIS** | **1/504** | **1150.1\*** |
| QUBO | 256 | 2 | 334/504 | 582.0 | 359/504 | 485.1 | 359/504 | 485.1 | 366/504 | 427. | **NgLglr** | **1/504** | **1083.0\*** |
| QUBO | 256 | 3 | 359/504 | 992.6 | 327/504 | 1262.9 | 365/504 | 929.0 | 368/504 | 845.3 | **NLOPT_LN_PRAXIS** | **1/504** | **2666.7\*** |
| QUBO | 256 | 4 | 334/504 | 1168.7 | 271/504 | 1324.6 | 358/504 | 978.8 | 367/504 | 856.0 | **NLOPT_LN_PRAXIS** | **1/504** | **2280.8\*** |
| QUBO | 256 | 5 | 335/504 | 1169.0 | 284/504 | 1303.5 | 360/504 | 977.1 | 366/504 | 882.2 | **NgLglr** | **1/504** | **2208.7\*** |
| NK | 64 | 1 | **1/504** | **0.7095\*** | 123/504 | 0.6876 | 90/504 | 0.6953 | 108/504 | 0.6914 | Neural1MetaModelD | 2/504 | 0.7000 |
| NK | 64 | 2 | **1/504** | **0.7378\*** | 162/504 | 0.6994 | 141/504 | 0.7029 | 200/504 | 0.6937 | LargeDiagCMA | 2/504 | 0.7225 |
| NK | 64 | 4 | **1/504** | **0.7341\*** | 308/504 | 0.6695 | 334/504 | 0.6545 | 342/504 | 0.6456 | CmaFmin2 | 2/504 | 0.7187 |
| NK | 64 | 8 | 335/504 | 0.6362 | 355/504 | 0.6287 | 359/504 | 0.6243 | 363/504 | 0.6190 | **DSsubspace** | **1/504** | **0.7166\*** |
| NK | 128 | 1 | 134/504 | 0.6658 | 137/504 | 0.6616 | 137/504 | 0.6616 | 236/504 | 0.6447 | **RF1MetaModelD** | **1/504** | **0.6883\*** |
| NK | 128 | 2 | 144/504 | 0.6609 | 220/504 | 0.6501 | 207/504 | 0.6530 | 315/504 | 0.6322 | **RF1MetaModelD** | **1/504** | **0.6988** |
| NK | 128 | 4 | 316/504 | 0.6277 | 318/504 | 0.6220 | 342/504 | 0.6105 | 355/504 | 0.6011 | **Quad1MetaModelD** | **1/504** | **0.7006\*** |
| NK | 128 | 8 | 359/504 | 0.5863 | 353/504 | 0.5898 | 361/504 | 0.5839 | 360/504 | 0.5844 | **NLOPT_LN_NELDERMEAD** | **1/504** | **0.6978\*** |
| NK | 256 | 1 | 264/504 | 0.5983 | 173/504 | 0.6094 | 124/504 | 0.6209 | 282/504 | 0.5966 | **NLOPT_LN_NELDERMEAD** | **1/504** | **0.6754\*** |
| NK | 256 | 2 | 314/504 | 0.5923 | 240/504 | 0.6071 | 213/504 | 0.6103 | 319/504 | 0.5901 | **LargeDiagCMA** | **1/504** | **0.6809\*** |
| NK | 256 | 4 | 352/504 | 0.5732 | 318/504 | 0.5859 | 351/504 | 0.5742 | 353/504 | 0.5696 | **NLOPT_LN_NELDERMEAD** | **1/504** | **0.6847\*** |
| NK | 256 | 8 | 362/504 | 0.5598 | 352 | 0.5632 | 364/504 | 0.5595 | 401/504 | 0.5581 | **NLOPT_LN_NELDERMEAD** | **1/504** | **0.6760\*** |
| NK3 | 64 | 1 | 52/499 | 0.7228 | - | - | 71/499 | 0.7140 | 45/499 | 0.7318 | SmallLognormalDiscreteOnePlusOne | **1/499** | **0.7419\*** |
| NK3 | 64 | 2 | 128/499 | 0.7012 | - | - | 202/499 | 0.6804 | 153/499 | 0.6934 | **NgLglr** | **1/499** | **0.7477\*** |
| NK3 | 64 | 4 | 272/499 | 0.6385 | - | - | 319/499 | 0.6252 | 318/499 | 0.6252 | **NgIohLn** | **1/499** | **0.7358\*** |
| NK3 | 64 | 8 | 311/499 | 0.6201 | - | - | 335/499 | 0.6172 | 320/499 | 0.6182 | **RLSOnePlusOne** | **1/499** | **0.7185\*** |
| NK3 | 128 | 1 | 128/499 | 0.6543 | - | - | 116/499 | 0.6689 | 101/499 | 0.6846 | **DiscreteLengler2OnePlusOne** | **1/499** | **0.7280\*** |
| NK3 | 128 | 2 | 159/499 | 0.6295 | - | - | 208/499 | 0.6223 | 157/499 | 0.6332 | **DiscreteLengler2OnePlusOne** | **1/499** | **0.7285\*** |
| NK3 | 128 | 4 | 249/499 | 0.5946 | - | - | 271/499 | 0.5874 | 268/499 | 0.5887 | **NGOptF5** | **1/499** | **0.7072\*** |
| NK3 | 128 | 8 | 286/499 | 0.5832 | - | - | 328/499 | 0.5826 | 285/499 | 0.5833 | **Carola10** | **1/499** | **0.6918\*** |
| NK3 | 256 | 1 | 127/499 | 0.6143 | - | - | 117/499 | 0.6200 | 110/499 | 0.6322 | **NGOptF5** | **1/499** | **0.7049\*** |
| NK3 | 256 | 2 | 212/499 | 0.5846 | - | - | 209/499 | 0.5847 | 159/499 | 0.5915 | **NGOptF5** | **1/499** | **0.7052\*** |
| NK3 | 256 | 4 | 234/499 | 0.5679 | - | - | 282/499 | 0.5621 | 274/499 | 0.5633 | **Carola1** | **1/499** | **0.6998\*** |
| NK3 | 256 | 8 | 314/499 | 0.5595 | - | - | 360/499 | 0.5582 | 363/499 | 0.5581 | **Cobyla** | **1/499** | **0.6779\*** |

# Q. Comparison with a Variant Using a Critic Neural Network

In this appendix, we compare $(\sigma, \sigma)$-RL-EDA with an alternative version using a critic neural network to compute advantages instead of the GRPO advantages described in Section 3.2 and given by (5).

This new variant called $(\sigma, \sigma)$-RL-EDA-Critic uses exactly the same algorithm, except that the advantages for individual $i$ at time step $k$ of the MDP are computed as

$$A^{\pi_{\theta t}}(\sigma^i(x^i)_{<k}, x_k^i) = \alpha(f(x^i) - \hat{V}(\sigma^i(x^i)_{<k}, x_k^i)), \tag{48}$$

with $f(x^i)$ the final score of the complete solution $x^i$ and $\hat{V}(\sigma^i(x^i)_{<k}, x_k^i)$ an estimate of the state value $(\sigma^i(x^i)_{<k}, x_k^i)$ given by a critic neural network composed of a set $(g_{\theta_1^c}, g_{\theta_2^c}, \ldots, g_{\theta_n^c})$ of $n$ neural networks (one for each variable), with exactly the same architecture as the set $(g_{\theta_1}, g_{\theta_2}, \ldots, g_{\theta_n})$ of generative neural networks used to build solutions, except that the sigmoid activation function is replaced by an identity function in order to output a value in $\mathbb{R}$. $\alpha$ is a hyperparameter

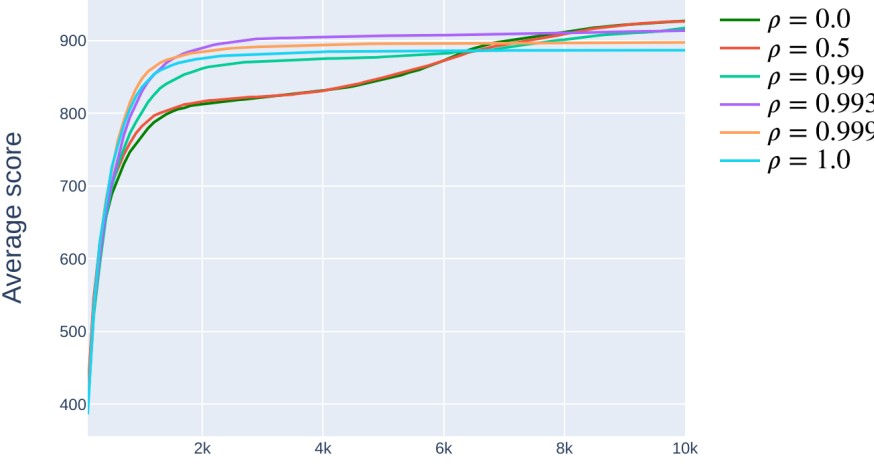

*(a)* QUBO instances with $n = 128$ and $K = 5$.

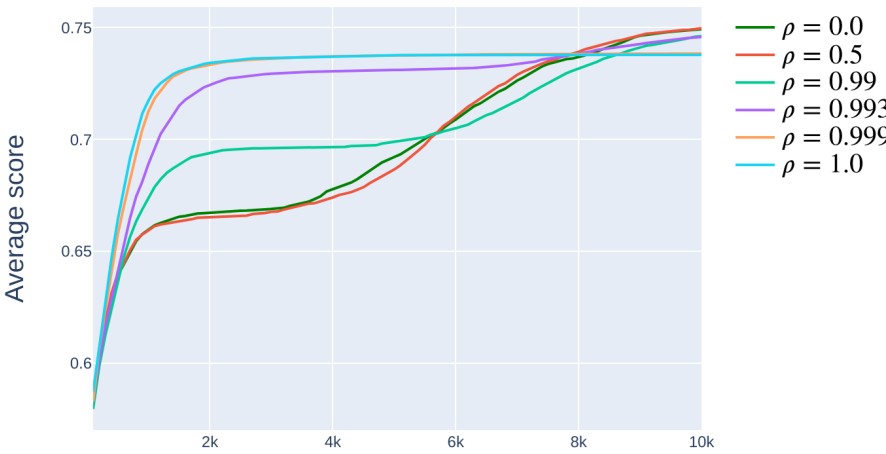

*(b)* NK instances with $n = 128$ and $K = 4$.

*Figure 19.* Sensitivity to the parameter $\rho$ in $(\sigma, \sigma)$-RL-EDA.

used to adjust the impact of the advantages on the learning process.

At each iteration $t$ of the EDA, at the beginning of the update phase, the n neural networks of the critic are trained in parallel during $E$ epochs to minimize the mean squared error between $f(x^i)$ and $\hat{V}(\sigma^i(x^i)_{<k}, x_k^i)$ at time step $k$ and for each individual $i$.

For each dataset, we evaluated several values of the hyperparameter $\alpha$ from the set $10^{-4}, 10^{-3}, 10^{-2}, 10^{-1}, 1, 10, 100$. The best performance was obtained with $\alpha = 10$ for the NK and NK3 datasets, and with $\alpha = 0.001$ for the QUBO datasets. Table 8 reports the results obtained by the variant incorporating a critic, denoted $(\sigma, \sigma)$-RL-EDA-Critic, in comparison with the standard version $(\sigma, \sigma)$-RL-EDA. Overall, the standard version outperforms the critic-based variant in most settings.

Furthermore, the critic-based approach exhibits two major drawbacks:

1. it requires nearly twice the computational time required to run the standard version, as the critic must be trained at each

*Table 7.* Global rankings and average scores obtained by **Fast-$(\sigma,\sigma)$-RL-EDA** and the other EDAs (`PBIL`, `MIMIC`, and `BOA`) are reported. The last columns present the ranking and average score of the best-performing method among the 500 additional algorithms considered (496 for NK3 problems). Rankings are computed over all 504 algorithms (499 for NK3 problems) by comparing the best score achieved **with short budget after 1,000 objective function evaluations**, averaged across 100 independent runs. Bold values highlight the best results among all competing methods. A star associated with the results obtained by `Fast-`$(\sigma,\sigma)$`-RL-EDA` indicates that it is significantly better on average (over 100 runs) than the best competing method. A star associated with a result obtained by another algorithm indicates that it is significantly better on average (over 100 runs) than `Fast-`$(\sigma,\sigma)$`-RL-EDA`. A difference in average scores is considered statistically significant according to a t-test with p-value 0.001.

| Instances | | | Methods | | | | | | | | | |
|---|---|---|---|---|---|---|---|---|---|---|---|---|
| | | | `Fast-`$(\sigma,\sigma)$`-RL-EDA` | | `PBIL` | | `MIMIC` | | `BOA` | | Best method (others) | | |
| Pb | $n$ | $K$ | Rank | Score | Rank | Score | Rank | Score | Rank | Score | Name | Rank | Score |
| QUBO | 64 | 0 | **1/504** | **189.5*** | 243/504 | 159.7 | 336/504 | 139.0 | 362/504 | 117.9 | Carola4 | 2/504 | 187.4 |
| QUBO | 64 | 1 | 17/504 | 142.6 | 127/504 | 137.3 | 225/504 | 129.6 | 343/504 | 113.8 | **LargeCMA** | **1/504** | **144.7*** |
| QUBO | 64 | 2 | 21/504 | 134.3 | 123/504 | 131.3 | 191/504 | 127.5 | 345/504 | 114.5 | **LargeCMA** | **1/504** | **136.4*** |
| QUBO | 64 | 3 | **1/504** | **396.3*** | 275/504 | 318.9 | 346/504 | 271.2 | 359/504 | 234.9 | Carola4 | 2/504 | 378.6 |
| QUBO | 64 | 4 | **1/504** | **316.5** | 174/504 | 287.3 | 306/504 | 261.1 | 352/504 | 230.7 | FCarola6 | 2/504 | 314.2 |
| QUBO | 64 | 5 | **1/504** | **297.9*** | 197/504 | 266.7 | 282/504 | 252.0 | 353/504 | 226.4 | NgLglr | 2/504 | 292.5 |
| QUBO | 128 | 0 | **1/504** | **525.6*** | 315/504 | 327.0 | 355/504 | 248.0 | 366/504 | 224.2 | NLOPT_LN_PRAXIS | 2/504 | 517.2 |
| QUBO | 128 | 1 | **1/504** | **417.7*** | 246/504 | 316.7 | 340/504 | 266.5 | 362/504 | 223.3 | Dis.Lengler2 1+1 | 2/504 | 406.1 |
| QUBO | 128 | 2 | **1/504** | **402.9*** | 246/504 | 319.7 | 340/504 | 269.6 | 361/504 | 229.7 | NgIohLn | 2/504 | 399.3 |
| QUBO | 128 | 3 | **1/504** | **1065.3*** | 320/504 | 661.4 | 354/504 | 506.6 | 368/504 | 442.7 | NLOPT_LN_PRAXIS | 2/504 | 1034.5 |
| QUBO | 128 | 4 | **1/504** | **881.6*** | 257/504 | 645.1 | 361/504 | 448.5 | 254/504 | 857.3 | Dis.Lengler2 1+1 | 2/504 | 845.8 |
| QUBO | 128 | 5 | **1/504** | **847.9*** | 262/504 | 629.9 | 346/504 | 523.8 | 365/504 | 440.8 | Dis.Lengler2 1+1 | 2/504 | 830.8 |
| QUBO | 256 | 0 | 49/504 | 1188.9 | 326/504 | 623.8 | 364/504 | 460.1 | 369/504 | 418.6 | **Carola1** | **1/504** | **1365.2*** |
| QUBO | 256 | 1 | 43/504 | 1089.5 | 278/504 | 648.7 | 358/504 | 485.2 | 367/504 | 439.0 | **NLOPT_LN_PRAXIS** | **1/504** | **1150.1*** |
| QUBO | 256 | 2 | 31/504 | 1033.6 | 359/504 | 485.1 | 359/504 | 485.1 | 366/504 | 427. | **NgLglr** | **1/504** | **1083.0*** |
| QUBO | 256 | 3 | 110/504 | 2400.6 | 327/504 | 1262.9 | 365/504 | 929.0 | 368/504 | 845.3 | **NLOPT_LN_PRAXIS** | **1/504** | **2666.7*** |
| QUBO | 256 | 4 | 42/504 | 2233.4 | 271/504 | 1324.6 | 358/504 | 978.8 | 367/504 | 856.0 | **NLOPT_LN_PRAXIS** | **1/504** | **2280.8*** |
| QUBO | 256 | 5 | 42/504 | 2105.2 | 284/504 | 1303.5 | 360/504 | 977.1 | 366/504 | 882.2 | **NgLglr** | **1/504** | **2208.7*** |
| NK | 64 | 1 | **1/504** | **0.7051*** | 123/504 | 0.6876 | 90/504 | 0.6953 | 108/504 | 0.6914 | Neural1MetaModelD | 2/504 | 0.7000 |
| NK | 64 | 2 | **1/504** | **0.7302*** | 162/504 | 0.6994 | 141/504 | 0.7029 | 200/504 | 0.6937 | LargeDiagCMA | 2/504 | 0.7225 |
| NK | 64 | 4 | **1/504** | **0.7320*** | 308/504 | 0.6695 | 334/504 | 0.6545 | 342/504 | 0.6456 | CmaFmin2 | 2/504 | 0.7187 |
| NK | 64 | 8 | 4/504 | 0.7120 | 355/504 | 0.6287 | 359/504 | 0.6243 | 363/504 | 0.6190 | **DSsubspace** | **1/504** | **0.7166** |
| NK | 128 | 1 | **1/504** | **0.6976*** | 137/504 | 0.6616 | 137/504 | 0.6616 | 236/504 | 0.6447 | RF1MetaModelD | 2/504 | 0.6883 |
| NK | 128 | 2 | **1/504** | **0.7146*** | 220/504 | 0.6501 | 207/504 | 0.6530 | 315/504 | 0.6322 | RF1MetaModelD | 2/504 | 0.6988 |
| NK | 128 | 4 | **1/504** | **0.7124** | 318/504 | 0.6220 | 342/504 | 0.6105 | 355/504 | 0.6011 | Quad1MetaModelD | 2/504 | 0.7006 |
| NK | 128 | 8 | 138/504 | 0.6567 | 353/504 | 0.5898 | 361/504 | 0.5839 | 360/504 | 0.5844 | **NLOPT_LN_NELDERMEAD** | **1/504** | **0.6978*** |
| NK | 256 | 1 | 16/504 | 0.6694 | 173/504 | 0.6094 | 124/504 | 0.6209 | 282/504 | 0.5966 | **NLOPT_LN_NELDERMEAD** | **1/504** | **0.6754*** |
| NK | 256 | 2 | 40/504 | 0.6793 | 240/504 | 0.6071 | 213/504 | 0.6103 | 319/504 | 0.5901 | **LargeDiagCMA** | **1/504** | **0.6809*** |
| NK | 256 | 4 | 71/504 | 0.6565 | 318/504 | 0.5859 | 351/504 | 0.5742 | 353/504 | 0.5696 | **NLOPT_LN_NELDERMEAD** | **1/504** | **0.6847*** |
| NK | 256 | 8 | 164/504 | 0.6057 | 352 | 0.5632 | 364/504 | 0.5595 | 401/504 | 0.5581 | **NLOPT_LN_NELDERMEAD** | **1/504** | **0.6760*** |
| NK3 | 64 | 1 | **1/499** | **0.7593*** | - | - | 71/499 | 0.7140 | 45/499 | 0.7318 | SmallLognormalDiscreteOnePlusOne | 2/499 | 0.7419 |
| NK3 | 64 | 2 | **1/499** | **0.7702*** | - | - | 202/499 | 0.6804 | 153/499 | 0.6934 | NgLglr | 1/499 | 0.7477 |
| NK3 | 64 | 4 | **1/499** | **0.7547*** | - | - | 319/499 | 0.6252 | 318/499 | 0.6252 | NgIohLn | 2/499 | 0.7358 |
| NK3 | 64 | 8 | 2/499 | 0.7169 | - | - | 335/499 | 0.6172 | 320/499 | 0.6182 | **RLSOnePlusOne** | **1/499** | **0.7185*** |
| NK3 | 128 | 1 | **1/499** | **0.7482*** | - | - | 116/499 | 0.6689 | 101/499 | 0.6846 | Dis.Lengler2 1+1 | 2/499 | 0.7280 |
| NK3 | 128 | 2 | **1/499** | **0.7457*** | - | - | 208/499 | 0.6223 | 157/499 | 0.6332 | Dis.Lengler2 1+1 | 2/499 | 0.7285 |
| NK3 | 128 | 4 | **1/499** | **0.7277*** | - | - | 271/499 | 0.5874 | 268/499 | 0.5887 | NGOptF5 | 2/499 | 0.7072 |
| NK3 | 128 | 8 | 46/499 | 0.6746 | - | - | 328/499 | 0.5826 | 285/499 | 0.5833 | **Carola10** | **1/499** | **0.6918*** |
| NK3 | 256 | 1 | 28/499 | 0.6999 | - | - | 117/499 | 0.6200 | 110/499 | 0.6322 | **NGOptF5** | **1/499** | **0.7049*** |
| NK3 | 256 | 2 | 44/499 | 0.6925 | - | - | 209/499 | 0.5847 | 159/499 | 0.5915 | **NGOptF5** | **1/499** | **0.7052*** |
| NK3 | 256 | 4 | 101/499 | 0.6571 | - | - | 282/499 | 0.5621 | 274/499 | 0.5633 | **Carola1** | **1/499** | **0.6998*** |
| NK3 | 256 | 8 | 132/499 | 0.6045 | - | - | 360/499 | 0.5582 | 363/499 | 0.5581 | **Cobyla** | **1/499** | **0.6779*** |

generation;

2. it makes the algorithm sensitive to the scale of fitness values, thereby reducing its robustness to the diverse instance distributions encountered.

# R. Variant Sharing Parameters of Hidden Layers for Scaling to Large Problems

In this appendix, we introduce a variant of the $(\sigma,\sigma)$-`RL-EDA` algorithm using a single MLP $g^n(\theta)$ with 2 hidden layers of 100 neurons and $n$ outputs, instead of the set $(g_{\theta_1}, g_{\theta_2}, \ldots, g_{\theta_n})$ of $n$ MLP, each with a single hidden layer of 20 neurons (see Section 3.1). In $g_\theta^n$, each of the $n$ outputs produces the probability of the value of each variable conditionally on the values of the other variables. This variant, which employs a single MLP denoted $g_\theta^n$, is called $(\sigma,\sigma)$-`RL-EDA-share-params`. All other hyperparameters are identical to those used in the standard version.

Table 9 reports the results obtained by $(\sigma,\sigma)$-`RL-EDA-share-params` in comparison with the standard version $(\sigma,\sigma)$-`RL-EDA`. Overall, the standard version $(\sigma,\sigma)$-`RL-EDA` is generally slightly more effective than

*Table 8.* Average scores obtained by $(\sigma, \sigma)$-RL-EDA and its variant $(\sigma, \sigma)$-RL-EDA-critic. Bold values highlight the best results. A star associated with the results indicates that it is significantly better on average (over 100 runs). A difference in the average scores is said statistically significant according to a t-test with p-value 0.001.

| Instances | | | Methods | |
|---|---|---|---|---|
| Pb | $n$ | $K$ | $(\sigma, \sigma)$-RL-EDA | $(\sigma, \sigma)$-RL-EDA-critic |
| QUBO | 64 | 0 | **200.8*** | 195.3 |
| QUBO | 64 | 1 | **148.8*** | 146.7 |
| QUBO | 64 | 2 | **138.1*** | 134.7 |
| QUBO | 64 | 3 | **411.2*** | 407.7 |
| QUBO | 64 | 4 | 326.1 | **326.4** |
| QUBO | 64 | 5 | **309.4*** | 303.6 |
| QUBO | 128 | 0 | **593.7*** | 560.7 |
| QUBO | 128 | 1 | **449.2*** | 432.5 |
| QUBO | 128 | 2 | **437.1*** | 412.6 |
| QUBO | 128 | 3 | **1227.2*** | 943.2 |
| QUBO | 128 | 4 | **955.4*** | 781.9 |
| QUBO | 128 | 5 | **933.3*** | 768.4 |
| QUBO | 256 | 0 | **1697.7*** | 1119.0 |
| QUBO | 256 | 1 | **1367.7*** | 596.9 |
| QUBO | 256 | 2 | **1304.1*** | 944.8 |
| QUBO | 256 | 3 | **3436.8*** | 1645.9 |
| QUBO | 256 | 4 | **2769.0*** | 1500.4 |
| QUBO | 256 | 5 | **2730.1*** | 1491.6 |
| NK | 64 | 1 | **0.7103** | 0.7099 |
| NK | 64 | 2 | **0.7420** | 0.7413 |
| NK | 64 | 4 | **0.7523** | 0.7495 |
| NK | 64 | 8 | 0.7379 | **0.7420*** |
| NK | 128 | 1 | **0.7100** | 0.7086 |
| NK | 128 | 2 | **0.7375*** | 0.7355 |
| NK | 128 | 4 | **0.7603** | 0.7574 |
| NK | 128 | 8 | 0.7369 | **0.7408*** |
| NK | 256 | 1 | **0.7071** | 0.706 |
| NK | 256 | 2 | **0.7364*** | 0.7349 |
| NK | 256 | 4 | **0.7534** | 0.7527 |
| NK | 256 | 8 | **0.7232*** | 0.696 |
| NK3 | 64 | 1 | 0.7818 | **0.7861*** |
| NK3 | 64 | 2 | 0.8095 | **0.8114** |
| NK3 | 64 | 4 | 0.8004 | **0.8016** |
| NK3 | 64 | 8 | **0.7473** | 0.7416 |
| NK3 | 128 | 1 | 0.7876 | **0.7957*** |
| NK3 | 128 | 2 | 0.7986 | **0.8101*** |
| NK3 | 128 | 4 | 0.7847 | **0.7988*** |
| NK3 | 128 | 8 | **0.7373*** | 0.6031 |
| NK3 | 256 | 1 | 0.7763 | **0.7811*** |
| NK3 | 256 | 2 | 0.7801 | **0.7802** |
| NK3 | 256 | 4 | **0.7615*** | 0.6342 |
| NK3 | 256 | 8 | **0.7213*** | 0.5723 |

$(\sigma, \sigma)$-RL-EDA-share-params on these small and medium-sized datasets. This suggests that employing one MLP per variable contributes to a more stable learning process during the search for these sizes of instances.

Then, we generated a new dataset of larger instances with 10 NK instances of size $n = 1028$ and $K = 8$, and launched the two variants $(\sigma, \sigma)$-RL-EDA and $(\sigma, \sigma)$-RL-EDA-share-params on these large instances.

First of all, we notice that the variant sharing parameters scales very well in terms of computational time required in comparison with the standard version. It required 1.5 hours to process 10 instances of size $n = 1028$ with a budget of 10,000 evaluations for the variant $(\sigma, \sigma)$-RL-EDA-share-params, while it required more than 10 hours for the standard version $(\sigma, \sigma)$-RL-EDA to perform the same task, which can be explained by the much lower number of parameters to be learned for the version with shared parameters.

Figure 20 shows the evolution of the average results over 100 runs of the two variants $(\sigma, \sigma)$-RL-EDA and

$(\sigma, \sigma)$-RL-EDA-share-params on these 10 large instances with 10 independent restarts on each instance, in comparison with the best other Nevegrad competitors as well as the other EDAs.

We then notice that our standard version $(\sigma, \sigma)$-RL-EDA also performs very poorly. This is because each generation of variables is produced by a small network with a hidden layer of size 20 that takes the other 1023 variables as input. This becomes insufficient to properly model the complex interactions between the variables.

However, we can see that the new variant sharing parameters called $(\sigma, \sigma)$-RL-EDA-share-params yields very good results (green dotted line), in comparison with the other best competitors.

*Table 9.* Average scores obtained by $(\sigma, \sigma)$-RL-EDA and its variant $(\sigma, \sigma)$-RL-EDA-share-params. Bold values highlight the best results. A star associated with the results indicates that it is significantly better on average (over 100 runs). A difference on the average scores is said statistically significant according to a t-test with p-value 0.001.

| Instances | | | Methods | |
|---|---|---|---|---|
| Pb | $n$ | $K$ | $(\sigma, \sigma)$-RL-EDA | $(\sigma, \sigma)$-RL-EDA-share-params |
| QUBO | 64 | 0 | **200.8** | 198.7 |
| QUBO | 64 | 1 | **148.8*** | 145.9 |
| QUBO | 64 | 2 | **138.1** | 137.9 |
| QUBO | 64 | 3 | 411.2 | **415.4** |
| QUBO | 64 | 4 | **326.1*** | 323.6 |
| QUBO | 64 | 5 | **309.4** | 309.26 |
| QUBO | 128 | 0 | **593.7*** | 584.8 |
| QUBO | 128 | 1 | **449.2*** | 437.9 |
| QUBO | 128 | 2 | **437.1*** | 429.5 |
| QUBO | 128 | 3 | **1227.2*** | 1211.3 |
| QUBO | 128 | 4 | **955.4*** | 944.6 |
| QUBO | 128 | 5 | **933.3*** | 920.9 |
| QUBO | 256 | 0 | **1697.7*** | 1669.6 |
| QUBO | 256 | 1 | **1367.7*** | 1337.4 |
| QUBO | 256 | 2 | **1304.1*** | 1272.9 |
| QUBO | 256 | 3 | **3436.8*** | 3400.9 |
| QUBO | 256 | 4 | **2769.0*** | 2696.9 |
| QUBO | 256 | 5 | **2730.1*** | 2654.6 |
| NK | 64 | 1 | 0.7103 | 0.7103 |
| NK | 64 | 2 | **0.7420** | 0.7402 |
| NK | 64 | 4 | **0.7523** | 0.7521 |
| NK | 64 | 8 | **0.7379** | 0.7367 |
| NK | 128 | 1 | **0.7100** | 0.7094 |
| NK | 128 | 2 | **0.7375*** | 0.7360 |
| NK | 128 | 4 | **0.7603** | 0.7569 |
| NK | 128 | 8 | **0.7369** | 0.7331 |
| NK | 256 | 1 | **0.7071** | 0.7065 |
| NK | 256 | 2 | **0.7364*** | 0.7352 |
| NK | 256 | 4 | **0.7534*** | 0.7478 |
| NK | 256 | 8 | 0.7232 | **0.7243** |
| NK3 | 64 | 1 | 0.7818 | **0.7835*** |
| NK3 | 64 | 2 | **0.8095** | 0.8079 |
| NK3 | 64 | 4 | **0.8004** | 0.7933 |
| NK3 | 64 | 8 | 0.7473 | **0.7478** |
| NK3 | 128 | 1 | **0.7876** | 0.7816 |
| NK3 | 128 | 2 | **0.7986** | 0.7908 |
| NK3 | 128 | 4 | **0.7847** | 0.7715 |
| NK3 | 128 | 8 | **0.7373*** | 0.7271 |
| NK3 | 256 | 1 | **0.7763*** | 0.7553 |
| NK3 | 256 | 2 | **0.7801** | 0.7601 |
| NK3 | 256 | 4 | **0.7615*** | 0.7434 |
| NK3 | 256 | 8 | **0.7213*** | 0.7105 |

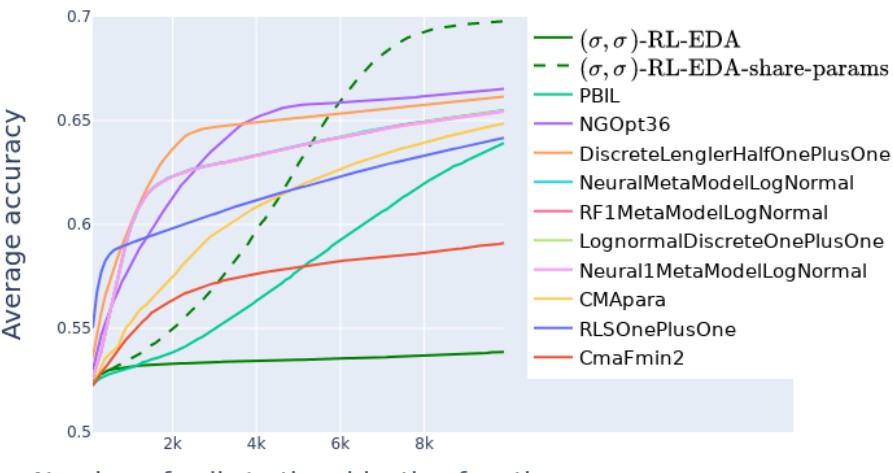

*Figure 20.* NK instances with $n = 1024$ and $K = 8$.

## S. Ablation Study: Non Autoregressive Generation (Using Gibbs Sampling)

In this appendix, we propose a baseline variant of the algorithm, where the sequential order of generation is replaced by Gibbs sampling. This is another way to build an order-invariant RL EDA.

In this case, in order to construct a complete solution $x$, we start with $x^0 = (x_1^0, x_2^0, \ldots, x_n^0) = (0, 0, \ldots, 0)$, then at iteration $\ell$, given a sample $x^{(\ell)} = (x_1^{(\ell)}, x_2^{(\ell)}, \ldots, x_n^{(\ell)})$, to obtain the next sample $x^{(\ell+1)} = (x_1^{(\ell+1)}, x_2^{(\ell+1)}, \ldots, x_n^{(\ell+1)})$, we sample each component $x_j^{(\ell+1)}$ conditioned on all other variables sampled so far, such that $\pi_\theta(x_j^{(\ell+1)} = 1 | x_{-j}^{(\ell)}) = \text{sigmoid}(g_{\theta_j}(x_{-j}^{(\ell)}))$, with $x_{-j}^{(\ell)} = (x_1^{(\ell)}, \ldots, x_{j-1}^{(\ell)}, 0, x_{j+1}^{(\ell)}, \ldots, x_n^{(\ell+1)})$ corresponding to the vector $x_j^{(\ell)}$, but with a 0 in position $j$. When $\ell \to \infty$, this process allows to obtain a sampling of the joint multivariate distribution. However in practice we restrict this sampling to $G$ iterations.

In this variant without order, during training, the GRPO objective to maximize becomes

$$\hat{L}_\lambda(\theta) = \frac{1}{\lambda} \sum_{(x^i, \sigma^i) \in \Gamma_\lambda^t} \sum_{k=1}^n \left[ \frac{\pi_\theta(x_k^i | x_{-k}^i)}{\pi_{\theta^t}(x_k^i | x_{-k}^i)} A_{\Gamma_\lambda^t}(x) - \beta D_{\text{KL}} \left( \pi_{\theta^t}(\cdot | x_{-k}^i) \| \pi_\theta(\cdot | x_{-k}^i) \right) \right]. \tag{49}$$

Figure 21 displays the results obtained by this variant with Gibbs sampling for different values of $G$ of iterations during sampling. The figure indicates that increasing the value of $G$ leads to improved performance. However, these gains rapidly plateau and remain below those achieved by the standard variant $(\sigma, \sigma)-\text{RL-EDA}$, which employs a sequential generation of variables based on randomly sampled permutation masks during both training and inference, which allows better uncovering of dependency relationships between variables (see section E for discussions about what can bring samplings of different causal masks at train time, in terms of residual structuring of the networks).

Moreover, the Gibbs-sampling version incurs substantially higher computational costs during the variable-generation phase, as it requires multiple re-samplings of each variable. In contrast, the standard approach assigns a value to each variable only once, resulting in significantly lower computational overhead.

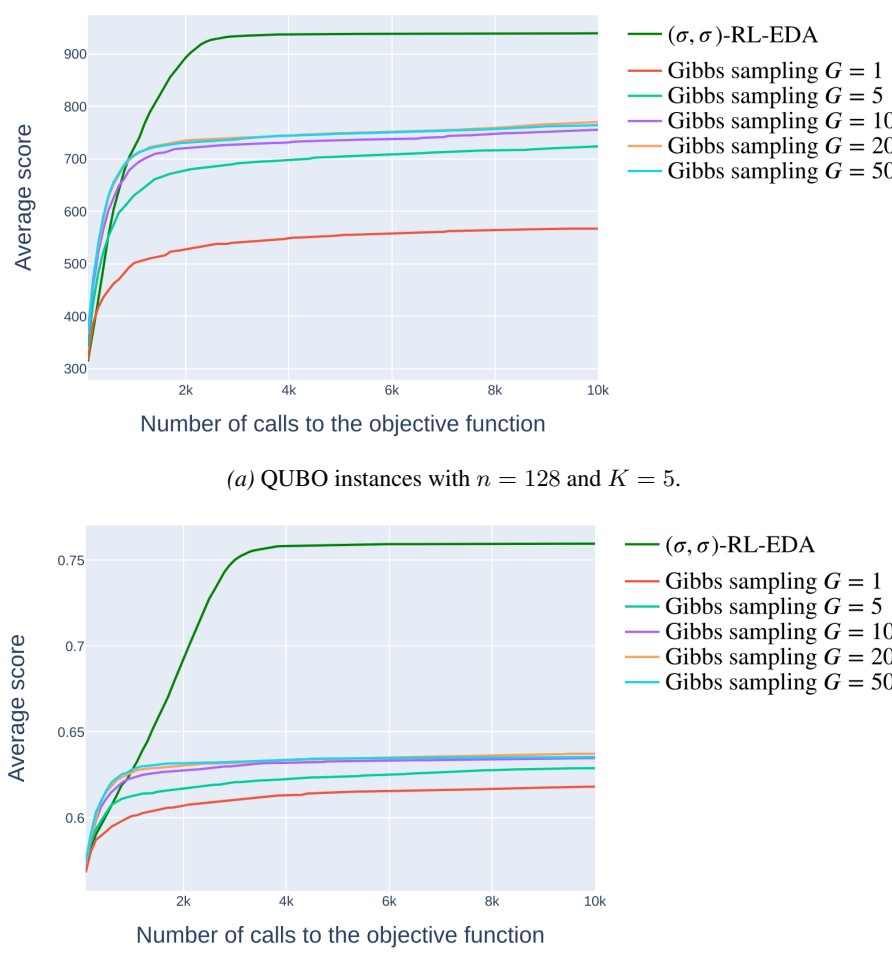

*(a)* QUBO instances with $n = 128$ and $K = 5$.

*(b)* NK instances with $n = 128$ and $K = 4$.

*Figure 21.* Evolution of the score of a variant with Gibbs sampling with different numbers of iterations $G$ for the sampling, in comparison with the standard version $(\sigma, \sigma)$-RL-EDA

## T. Nevergrad Competing Algorithms

It is important to note that some algorithms in the library are primarily designed for continuous optimization—such as various variants of Particle Swarm Optimization (PSO) and CMA-ES— and are not expected to perform competitively on discrete problems. Nevertheless, Nevergrad (Rapin & Teytaud, 2018) also includes a wide range of algorithms specifically tailored for large-scale discrete black-box optimization. The algorithms of the Nevergrad library can be grouped into the following categories:

- **Memetic and Genetic Algorithms**, such as cGA and discretememetic.

- **Discrete** $(1 + 1)$ **Evolutionary Algorithms**, including variants with adaptive mutation rates like DiscreteLengler2OnePlusOne and FastGADiscreteOnePlusOne.

- **Differential Evolution algorithms**, e.g., DiscreteDE, LhsHSDE.

- **Chaining Algorithms**, which are meta-algorithms applying several baseline algorithms in sequence, such as ChainDEwithLHS30, Carola1, ..., Carola15.

- **Portfolio Algorithms**, including NGOpt, NgIoh, and Wiz, which select low-level algorithms based on problem dimension and budget.

- **Adaptive Portfolio Algorithms**, which test several algorithms during early search phases before selecting one for later stages, e.g., `PolyLN`.

- **Learning Meta-Models**, which approximate the optimum using supervised models (e.g., random forests, neural networks, SVMs) trained on the best solutions generated by low-level algorithms. Examples include `RF1MetaModel`, `Neural1MetaModelOnePlusOne`, and `SVM1MetaModelD`.

Here we provide the complete list of all competing algorithms from version 1.0.12 of the `Nevergrad` library used in the experiments (sorted by name). Detailed documentation and source code for these algorithms are available at https://facebookresearch.github.io/nevergrad.

AdaptiveDiscreteOnePlusOne, AlmostRotationInvariantDE, AlmostRotationInvariantDEAndBigPop, AnisoEMNA, AnisoEMNATBPSA, AnisotropicAdaptiveDiscreteOnePlusOne, ASCMADEthird, AvgHammersleySearch, AvgHammersleySearchPlusMiddlePoint, AvgMetaRecenteringNoHull, AvgRandomSearch, BAR, BAR2, BAR3, BAR4, BFGS, BFGSCMA, BFGSCMAPlus, BigLognormalDiscreteOnePlusOne, BPRotationInvariantDE, Carola1, Carola2, Carola3, Carola4, Carola5, Carola6, Carola7, Carola8, Carola9, Carola10, Carola11, Carola13, Carola14, Carola15, CauchyLHSSearch, CauchyOnePlusOne, CauchyRandomSearch, CauchyScrHammersleySearch, cGA, ChainCMAPowell, ChainCMASQP, ChainCMAwithLHS, ChainCMAwithLHS30, ChainCMAwithLHSdim, ChainCMAwithLHSsqrt, ChainCMAwithMetaRecentering, ChainCMAwithMetaRecentering30, ChainCMAwithMetaRecenteringdim, ChainCMAwithMetaRecenteringsqrt, ChainCMAwithR, ChainCMAwithR30, ChainCMAwithRdim, ChainCMAwithRsqrt, ChainDE, ChainDEwithLHS, ChainDEwithLHS30, ChainDEwithLHSdim, ChainDEwithLHSsqrt, ChainDEwithMetaRecentering, ChainDEwithMetaRecentering30, ChainDEwithMetaRecenteringdim, ChainDEwithMetaRecenteringsqrt, ChainDEwithMetaTuneRecentering, ChainDEwithMetaTuneRecentering30, ChainDEwithMetaTuneRecenteringdim, ChainDEwithMetaTuneRecenteringsqrt, ChainDEwithR, ChainDEwithR30, ChainDEwithRdim, ChainDEwithRsqrt, ChainDiagonalCMAPowell, ChainDSPowell, ChainMetaModelDSSQP, ChainMetaModelPowell, ChainMetaModelSQP, ChainNaiveTBPSACMAPowell, ChainNaiveTBPSAPowell, ChainPSOwithLHS, ChainPSOwithLHS30, ChainPSOwithLHSdim, ChainPSOwithLHSsqrt, ChainPSOwithMetaRecentering, ChainPSOwithMetaRecentering30, ChainPSOwithMetaRecenteringdim, ChainPSOwithMetaRecenteringsqrt, ChainPSOwithR, ChainPSOwithR30, ChainPSOwithRdim, ChainPSOwithRsqrt, ChoiceBase, CLengler, CM, CMA, CMAbounded, CmaFmin2, CMAL, CMAL2, CMAL3, CMALL, CMALn, CMALS, CMALYS, CMandAS2, CMandAS3, CMApara, CMARS, CMASL, CMASL2, CMASL3, CMAsmall, CMAstd, CMAtuning, Cobyla, CSEC, CSEC10, CSEC11, DE, DiagonalCMA, DiscreteBSOOnePlusOne, DiscreteDE, DiscreteDoerrOnePlusOne, DiscreteLengler2OnePlusOne, DiscreteLengler3OnePlusOne, DiscreteLenglerFourthOnePlusOne, DiscreteLenglerHalfOnePlusOne, DiscreteLenglerOnePlusOne, DiscreteLenglerOnePlusOneT, discretememetic, DiscreteNoisy13Splits, DiscreteOnePlusOne, DiscreteOnePlusOneT, DoubleFastGADiscreteOnePlusOne, DoubleFastGAOptimisticNoisyDiscreteOnePlusOne, DS2, DS3p, DS4, DS5, DS6, DS8, DS9, DS14, DSbase, DSproba, DSsubspace, ECMA, EDA, EDCMA, ES, F2SQPCMA, F3SQPCMA, FastGADiscreteOnePlusOne, FastGANoisyDiscreteOnePlusOne, FastGAOptimisticNoisyDiscreteOnePlusOne, FCarola6, FCMA, FCMAp13, FCMAs03, file, ForceMultiCobyla, FSQPCMA, GeneticDE, HaltonSearch, HaltonSearchPlusMiddlePoint, HammersleySearch, HammersleySearchPlusMiddlePoint, HSDE, HugeLognormalDiscreteOnePlusOne, HullAvgMetaRecentering, HullAvgMetaTuneRecentering, HullCenterHullAvgCauchyLHSSearch, HullCenterHullAvgCauchyScrHammersleySearch, HullCenterHullAvgLargeHammersleySearch, HullCenterHullAvgLHSSearch, HullCenterHullAvgRandomSearch, HullCenterHullAvgScrHaltonSearch, HullCenterHullAvgScrHaltonSearchPlusMiddlePoint, HullCenterHullAvgScrHammersleySearch, HullCenterHullAvgScrHammersleySearchPlusMiddlePoint, IsoEMNA, IsoEMNATBPSA, LargeCMA, LargeDiagCMA, LargeHaltonSearch, LBFGSB, LhsDE, LhsHSDE, LHSSearch, LocalBFGS, LogBFGSCMA, LogBFGSCMAPlus, LogMultiBFGS, LogMultiBFGSPlus, LognormalDiscreteOnePlusOne, LogSQPCMA, LogSQPCMAPlus, LPCMA, LPSDE, LQODE, LQOTPDE, LSCMA, LSDE, ManyLN, MaxRecombiningDiscreteLenglerOnePlusOne, MemeticDE, MetaCauchyRecentering, MetaCMA, MetaModel, MetaModelDE, MetaModelDiagonalCMA, MetaModelDSproba, MetaModelFmin2, MetaModelLogNormal, MetaModelOnePlusOne, MetaModelPSO, MetaModelQODE, MetaModelTwoPointsDE, MetaNGOpt10, MetaRecentering, MetaTuneRecentering, MicroCMA, MicroSPSA, MicroSQP, MilliCMA, MiniDE, MiniLhsDE, MiniQrDE, MinRecombiningDiscreteLenglerOnePlusOne, MixDeterministicRL, MixES, MultiBFGS, MultiBFGSPlus, MultiCMA, MultiCobyla, MultiCobylaPlus, MultiDiscrete, MultiDS, MultiLN, MultiScaleCMA, MultiSQP, MultiSQPPlus, MutDE, NaiveAnisoEMNA, NaiveAnisoEMNATBPSA, NaiveIsoEMNA, NaiveIsoEMNATBPSA, NaiveTBPSA, NelderMead, Neural1MetaModel, Neural1MetaModelD, Neural1MetaModelE, Neural1MetaModelLogNormal, NeuralMetaModel, NeuralMetaModelDE, NeuralMetaModelLogNormal, NeuralMetaModelTwoPointsDE, NgDS, NgDS11, NgDS2, NgDS3, NGDSRW, NgIoh, NgIoh2, NgIoh3, NgIoh4, NgIoh5, NgIoh6, NgIoh7, NgIoh8, NgIoh9, NgIoh10, NgIoh11, NgIoh12, NgIoh12b, NgIoh13, NgIoh13b, NgIoh14, NgIoh14b, NgIoh15, NgIoh15b, NgIoh16, NgIoh17, NgIoh18, NgIoh19, NgIoh20, NgIoh21, NgIohLn, NgIohMLn, NgIohRS, NgIohRW2, NgIohTuned, NgLglr, NgLn, NGO, NGOpt, NGOpt10, NGOpt15, NGOpt16, NGOpt36, NGOpt39, NGOpt4, NGOpt8, NGOptBase, NGOptDSBase, NGOptF, NGOptF2, NGOptF3, NGOptF5, NGOptRW, NGOptSingle16, NGOptSingle25, NGOptSingle9, NgRS, NLOPT_GN_CRS2_LM, NLOPT_GN_DIRECT, NLOPT_GN_DIRECT_L, NLOPT_GN_ESCH, NLOPT_GN_ISRES, NLOPT_LN_NELDERMEAD, NLOPT_LN_PRAXIS, NLOPT_LN_SBPLX, Noisy13Splits, NoisyBandit, NoisyDE, NoisyDiscreteOnePlusOne, NoisyOnePlusOne, NoisyRL1, NoisyRL2, NoisyRL3, NonNSGAIIES, OldCMA, OLNDiscreteOnePlusOne, OnePlusLambda, OnePlusOne, OnePointDE, OnePtRecombiningDiscreteLenglerOnePlusOne, OpoDE, OpoTinyDE, OptimisticDiscreteOnePlusOne, OptimisticNoisyOnePlusOne, ORandomSearch, OScrHammersleySearch, ParametrizationDE, ParaPortfolio, pCarola6, PCarola6, PolyCMA, PolyLN, Portfolio, PortfolioDiscreteOnePlusOne, PortfolioDiscreteOnePlusOneT, PortfolioNoisyDiscreteOnePlusOne, PortfolioOptimisticNoisyDiscreteOnePlusOne, Powell, PSO, QNDE, QODE, QOPSO, QORandomSearch, QORealSpacePSO, QOScrHammersleySearch, QOTPDE, QrDE, Quad1MetaModel, Quad1MetaModelD, Quad1MetaModelE, RandomScaleRandomSearch, RandomScaleRandomSearchPlusMiddlePoint, RandomSearch, RandomSearchPlusMiddlePoint, RandRecombiningDiscreteLenglerOnePlusOne, RandRecombiningDiscreteLognormalOnePlusOne, RBFGS, RealSpacePSO, RecES, RecMixES, RecMutDE, RecombiningDiscreteLenglerOnePlusOne, RecombiningDiscreteLognormalOnePlusOne, RecombiningGA, RecombiningOptimisticNoisyDiscreteOnePlusOne, RecombiningPortfolioDiscreteOnePlusOne, RecombiningPortfolioOptimisticNoisyDiscreteOnePlusOne, RescaledCMA, RescaleScrHammersleySearch, RF1MetaModel, RF1MetaModelD, RF1MetaModelE, RF1MetaModelLogNormal, RFMetaModel, RFMetaModelDE, RFMetaModelLogNormal, RFMetaModelOnePlusOne, RFMetaModelPSO, RFMetaModelTwoPointsDE, RLSOnePlusOne, RotatedRecombiningGA,

RotatedTwoPointsDE, RotationInvariantDE, RPowell, RSQP, SADiscreteLenglerOnePlusOneExp09, SADiscreteLenglerOnePlusOneExp099, SADiscreteLenglerOnePlusOne-Exp09Auto, SADiscreteLenglerOnePlusOneLin1, SADiscreteLenglerOnePlusOneLin100, SADiscreteLenglerOnePlusOneLinAuto, SADiscreteOnePlusOneExp09, SADiscreteOnePlusOneExp099, SADiscreteOnePlusOneLin100, ScrHaltonSearch, ScrHaltonSearchPlusMiddlePoint, ScrHammersleySearch, ScrHammersleySearchPlusMiddlePoint, SDiagonalCMA, Shiwa, SmallLognormalDiscreteOnePlusOne, SmoothAdaptiveDiscreteOnePlusOne, SmoothDiscreteLenglerOnePlusOne, SmoothDiscreteLognormalOnePlusOne, SmoothDiscreteOnePlusOne, SmoothElitistRandRecombiningDiscreteLenglerOnePlusOne, SmoothElitistRandRecombiningDiscreteLognormalOnePlusOne, SmoothElitistRecombiningDiscreteLenglerOnePlusOne, SmootherDiscreteLenglerOnePlusOne, SmoothLognormalDiscreteOnePlusOne, SmoothPortfolioDiscreteOnePlusOne, SmoothRecombiningDiscreteLenglerOnePlusOne, SmoothRecombiningPortfolioDiscreteOnePlusOne, SODE, SOPSO, SparseDiscreteOnePlusOne, SparseDoubleFastGADiscreteOnePlusOne, SparseOrNot, SpecialRL, SplitCMA, SplitDE, SplitPSO, SplitQODE, SplitSQOPSO, SplitTwoPointsDE, SPQODE, SPSA, SQOPSO, SQOPSODCMA, SQOPSODCMA20, SQORealSpacePSO, SQP, SQPCMA, SQPCMAPlus, SqrtBFGSCMA, SqrtBFGSCMAPlus, SqrtMultiBFGS, SqrtMultiBFGSPlus, SqrtSQPCMA, SqrtSQPCMAPlus, StupidRandom, SuperSmoothDiscreteLenglerOnePlusOne, SuperSmoothElitistRecombiningDiscreteLenglerOnePlusOne, SuperSmoothRecombiningDiscreteLenglerOnePlusOne, SuperSmoothRecombiningDiscreteLognormalOnePlusOne, SuperSmoothTinyLognormalDiscreteOnePlusOne, SVM1MetaModel, SVM1MetaModelD, SVM1MetaModelE, SVM1MetaModelLogNormal, SVMMetaModel, SVMMetaModelDE, SVMMetaModelLogNormal, SVMMetaModelPSO, SVMMetaModelTwoPointsDE, TBPSA, TEAvgCauchyLHSSearch, TEAvgCauchyScrHammersleySearch, TEAvgLHSSearch, TEAvgRandomSearch, TEAvgScrHammersleySearch, TEAvgScrHammersleySearchPlusMiddlePoint, TinyCMA, TinyLhsDE, TinyLognormalDiscreteOnePlusOne, TinyQODE, TinySPSA, TinySQP, TripleCMA, TripleDiagonalCMA, TripleOnePlusOne, TwoPointsDE, TwoPtRecombiningDiscreteLenglerOnePlusOne, UltraSmoothDiscreteLenglerOnePlusOne, UltraSmoothElitistRecombiningDiscreteLenglerOnePlusOne, UltraSmoothElitistRecombiningDiscreteLognormalOnePlusOne, UltraSmoothRecombiningDiscreteLenglerOnePlusOne, VastDE, VastLengler, VLPCMA, VoronoiDE, Wiz, XLognormalDiscreteOnePlusOne, XSmallLognormalDiscreteOnePlusOne, YoSmoothDiscreteLenglerOnePlusOne, Zero.

## U. LLM Usage Declaration

During the preparation of this manuscript, we used Large Language Models to assist with text clarity, grammar, and formulation, particularly in polishing the abstract and certain explanatory sentences. The scientific content, experimental design, results, and interpretations were entirely conceived and written by the authors. LLMs were not used to generate original ideas, proofs, or analyses; its contribution was limited to language refinement.

