# OpenReview forum: "Black-Box Combinatorial Optimization with Order-Invariant Reinforcement Learning"
_ICML.cc/2026/Conference — ICML 2026 regular_

### Official Review · Reviewer_DovL · 2026-02-24

**Soundness:** 3
**Presentation:** 3
**Significance:** 3
**Originality:** 3
**Overall Recommendation:** 5
**Confidence:** 2

**Summary:**

The paper proposes a new reinforcement learning–based algorithm for black-box combinatorial optimization. Their method is an estimation-of-distribution (EOD) approach in which a distribution over candidate solutions is learned iteratively: at each step, a set of candidates is sampled from the current distribution, the fittest solutions are selected, and this feedback is used to update the distribution. Classical EOD algorithms often assume a (partial) order over the variables. In contrast, the proposed method is order-agnostic: the order in which variables are considered is randomized. The authors parameterize the solution distribution using an MLP-based multivariate autoregressive generative model and train it with an adaptation of Group Relative Policy Optimization (GRPO). Through numerical experiments, they show that their approach often achieves the best performance relative to classical black-box combinatorial optimization benchmarks.

**Compliance With Llm Reviewing Policy:**

Affirmed.

**Final Justification:**

The authors have addressed my concerns, and I see clear value on the method presented in the paper. I therefore recommand acceptance.

**Key Questions For Authors:**

- Can you comment on the running time of your approach compared to the other methods?

**Limitations:**

yes

**Strengths And Weaknesses:**

Strengths:
* The problem considered is important.
* To my knowledge, the paper is among the first to use neural autoregressive models to represent distributions in this setting.
* The order invariant generation idea appears to have a significant empirical impact and seems to encourage more exploration than Bayesian network based approaches.
* The paper is well written and easy to read.

Weaknesses:
* The main idea of removing the dependence on a fixed variable ordering is appealing, but most of the methods used in the paper are adaptations of existing techniques, and the paper’s fundamental methodological contribution is therefore somewhat limited. That said, I appreciate the empirical gains achieved by combining these methods, and overall I lean more toward acceptance than rejection.

---

> ### Author Rebuttal · Authors · 2026-03-30
>
> We would like to thank the reviewer for their insightful comments.
>
> Addressing weaknesses:
>
> Our contribution goes beyond standard policy-gradient EDAs by introducing **the first order-invariant multivariate generator and a permutation-regularized GRPO objective**, which together enable stable learning of dependency structures without assuming any fixed factorization. To our knowledge, we are the **first to apply reinforcement learning to combinatorial optimization using an autoregressive construction model**, and we show that strong performance in a fully black-box, data-scarce setting **relies critically on order-invariant training and permutation-based regularization**.
>
> Beyond introducing this framework, we provide **a principled analysis of its key components and properties**, explaining why these mechanisms are essential for effective and robust optimization. This analysis covers the effect of variable generation orders (Appendix C), convergence in the infinite‑data regime (Appendix D), the role of structural dropout (Appendix E), and the connection to natural gradient methods and the Information Geometric Optimization framework described by Ollivier et al. (2017) with the use of rank‑based rewards as employed in the well‑known CMA‑ES algorithm (Appendix G).
>
> Answer to question :
>
> Many thanks for the suggestion regarding running time of our approach. **Appendix L provides direct computational-cost measurements**, including wall-clock training time, CPU/GPU runtimes, and memory/compute complexity, and compares them to traditional multivariate BOA EDA. In particular, Appendix L reports real runtimes for multiple instance sizes and shows that our method remains competitive or faster this multivariate EDA despite using RL updates. We will make this information more visible in the main text to clarify that the comparison is not limited to function-evaluation counts but also includes concrete training-time and resource analyses.

---

> > ### Author Rebuttal · Reviewer_DovL · 2026-04-02
> >
> > Thank you for your feedback. My questions have been addressed.

---

### Official Review · Reviewer_nWUX · 2026-03-09

**Soundness:** 2
**Presentation:** 2
**Significance:** 2
**Originality:** 2
**Overall Recommendation:** 4
**Confidence:** 3

**Summary:**

The paper presents an order-invariant RL framework for black-box combinatorial optimization, framed as an EDA whose sampling distribution is represented by a neural autoregressive policy. Instead of building solutions in a single fixed variable order, it uses random orders during generation and training to improve exploration. The model is trained with a GRPO-style, rank-based update, so it does not require a separate critic and is less sensitive to objective scaling. The approach is evaluated on several standard benchmarks and compared against a broad range of baseline methods.

**Compliance With Llm Reviewing Policy:**

Affirmed.

**Final Justification:**

I appreciate the rebuttal and the additional clarifications. The authors have sufficiently addressed my concerns. While I still see some limitations in terms of novelty and practical performance under tighter evaluation budgets, I now think the strengths of the paper are significant enough to offset these weaknesses. For this reason, I am updating my score to weak accept. I also recommend that the authors move the faster variant into the main text, as it helps clarify the practical performance of the approach in lower-budget settings.

**Key Questions For Authors:**

Please see the Weaknesses section for questions.

**Limitations:**

Yes

**Strengths And Weaknesses:**

**Strengths**

- The paper provides a clear and well-justified connection between EDAs and policy-gradient methods.
- It provides a strong experimental evaluation, with thorough ablations and a broad baseline comparison showing improvements over several baselines.


**Weaknesses**

- The novelty appears limited, as the method largely builds on well-known policy-gradient style updates for EDAs, and the empirical improvements over strong baselines are not fully convincing.

- Scalability is also a main concern: the proposed architecture degrades for large $n$, requires $O(n^2)$ memory, and the shared-parameter variant improves runtime only by changing the model, while still not matching simple baselines.

- With a limited evaluation budget, the proposed method can perform poorly; it would therefore be informative to report the average score as a function of wall-clock time (not just the number of evaluations).

- Although the authors use one fixed setting for simplicity, the method still relies on manual tuning, with no clear way to automatically adjust hyperparameters such as the utility weights.

---

> ### Author Rebuttal · Authors · 2026-03-30
>
> We would like to thank the reviewer for their insightful comments.
>
> Addressing weaknesses:
>
> 1. **Our contribution goes beyond standard policy-gradient EDAs by introducing the first order-invariant multivariate generator and a permutation-regularized GRPO objective**, which together enable stable learning of dependency structures without assuming any fixed factorization. The empirical results, summarized in the main paper and detailed in Appendix M (where we compare against 500+ strong Nevergrad baselines), show that the method consistently ranks among the top performers, especially on medium and large-scale discrete problems. Note also that an increase in score from 0.74 to 0.76 (as observed for example for NK instances with N = 128 and K = 4) represents a substantial improvement for this class of optimization problems.
>
> 2. We respectfully disagree with this assessment regarding the fact that the shared‑parameter variant does not match simple baselines. As shown in Figure 19 (Appendix R), **the shared‑parameter variant (green dotted line) clearly outperforms all baseline methods on large instances with n = 1024**. This demonstrates that our order-invariant RL approach  continues to scale effectively in the large‑$n$ regime. Furthermore, the shared‑parameter variant has a memory requirement which scales in O(n). For one NK instance of size n = 1024 it requires only 700 MB of memory. This indicates that the approach is still highly usable in practice even to deal with large instances.
>
> 3. First, we acknowledge that our algorithm may require more evaluations to surpass certain baselines in some cases as seen in Figure 2. However, **this behavior is expected and does not undermine the practicality of our approach for several reasons**:
> - First, performance depends on instance characteristics: on QUBO or NK landscapes with small size (n=64)  our method is the best with a low budget  (see Table 5 in Appendix P for detailed results with low budget of 1,000 evaluations).  A similar trend appears in our real‑world NAS experiment (Appendix O), where our approach is the top performer for small budgets (Figure 3).
> - Second, EDAs are designed to start with maximum entropy in the sampling distribution and gradually reduce entropy to avoid premature convergence to local optima. Our RL-based approach follows this principle, which explains slower early progress but ensures robustness in high-dimensional, multimodal landscapes.
> - Third, comparing against 500 algorithms means some metaheuristics will always appear better for a specific evaluation budget; it may be easy to design a heuristic that converges very quickly but gets trapped in poor local optima. Our method prioritizes exploration–exploitation balance and long-term performance.
> - Fourth, learning a multivariate model is inherently more complex than univariate models like PBIL, which explains the initial gap. To overcome this problem, we introduce in Appendix P a variant of our algorithm called Fast-(σ,σ)-RL-EDA which use a curriculum approach, so that the model is univariate at the start of the search in order to find a good-quality solution more quickly at the beginning, then gradually switches to a multivariate mode. The results presented in Table 6 show that this variant frequently obtain the best results in comparison with all the baselines for a wide variety of distributions of instances of different sizes with a low budget of 1,000 evaluations.
>
>    Regarding wall-clock time, many thanks for the suggestion. **Appendix L provides direct computational-cost measurements, including wall-clock training time**, CPU/GPU runtimes, and memory/compute complexity, and compares them to traditional  multivariate EDAs.
>
> 4. Thanks for the comment about hyper-parameter tuning. First, we would like to remark that **our method does not rely on problem-specific tuning**: all results in Figs.2--3, 6--7 and Table 4 were obtained with a single hyperparameter setting, and (σ,σ)-RL-EDA performs consistently across QUBO, NK (K=1,8), NK3, and NASBench. Appendices N.4 and N.7 show that performance is stable with respect to the group size λ and number of epochs E, and an additional study confirms that standard Adam learning rates $10^{-3}$--$10^{-4}$ are robust.
>
>    Since Appendix N.5 indicates higher sensitivity to β, **we explored a simple adaptive mechanism**. Every 10 iterations, we estimate ruggedness using the Pearson correlation $r$ between Hamming distances and absolute fitness differences over the last 100 solutions, and update β $ \leftarrow \frac{0.12}{r + 10^{-7}}$. This decreases β on smooth landscapes and increases it on rugged ones. Preliminary results show consistent gains over fixed  β, e.g., an average of +2.65% on QUBO-64, +0.48% on QUBO-128, and small but systematic improvements on NK-64/128 (+0.17%).
>
>    We will include this adaptive-β analysis in the final version. Finally, please note that the scale of utility weights directly depends on the learning rate and β.

---

> > ### Author Rebuttal · Reviewer_nWUX · 2026-04-02
> >
> > Thank you for the rebuttal. I appreciate the additional clarifications. However, I am keeping my score unchanged for two main reasons. First, I am still not fully convinced by the degree of methodological novelty, since the core ideas remain closely related to existing RL-based combinatorial optimization approaches. Second, the practical advantage of the method is still not fully established to me, especially under tighter evaluation budgets, where the main method appears less competitive. In the rebuttal, the authors mainly refer to modified faster variants, but these results do not fully resolve my concern regarding the core method studied in the main submission.

---

> > > ### Author Response · Authors · 2026-04-03
> > >
> > > Thank you for this rebuttal acknowledgement.
> > >
> > > - Regarding novelty, we respectfully disagree with the assessment. Our work introduces a new framework for Estimation‑of‑Distribution Algorithms, which, to our knowledge, has not been explored before. While it leverages certain known RL components, their combination, purpose, and theoretical grounding within our framework are fundamentally different from prior approaches.
> > >
> > >    In particular, Appendix C demonstrates that PPO can be extended to variable‑order autoregressive models while remaining consistent with the theoretical foundations reviewed in Appendix B. This extension is non‑trivial, novel, and potentially applicable beyond our use case.
> > >
> > >    We also show the significant effect of variable orders and dropout on performance. As illustrated in Fig. 1, purely constructive autoregressive models fail without varying the orders, which likely explains why this direction had not been explored previously. We believe this represents an important and original contribution.
> > >
> > > - Regarding the practical advantage of the core method, we respectfully disagree with the reviewer’s concern. The strong empirical performance of our core method under tight evaluation budgets is already demonstrated in the paper.
> > > Specifically, on the real dataset Nasbench‑101 (see Fig. 3), as well as on QUBO and NK problems of size 64 (see Table 5 in Appendix P), the core method achieves state‑of‑the‑art performance in the small‑budget regime, outperforming more than 500 black‑box optimizers (listed in Appendix T).
> > >
> > >    Moreover, for larger instances, we show in Appendix P that a simple and controlled adjustment of a few hyperparameters, while keeping the core method unchanged, yields the best performance among these same 500 competing methods in the low‑budget setting.

---

### Official Review · Reviewer_FmE6 · 2026-03-13

**Soundness:** 3
**Presentation:** 3
**Significance:** 3
**Originality:** 3
**Overall Recommendation:** 4
**Confidence:** 1

**Summary:**

This paper proposes an order-invariant reinforcement-learning-based estimation-of-distribution algorithm for discrete black-box combinatorial optimization. The core idea is to train an autoregressive neural generator while randomizing the order of variable generation during both training and sampling, with the goal of improving diversity, regularization, and sample efficiency. The method is built on a PPO/GRPO-style objective with rank-based, scale-invariant advantages and is evaluated on synthetic QUBO, NK, and NK3 benchmarks, as well as NAS-Bench-101. The main empirical message is that the fully randomized variant, denoted ((\sigma,\sigma))-RL-EDA, performs best among the proposed variants and is competitive with a large set of Nevergrad baselines.

**Compliance With Llm Reviewing Policy:**

Affirmed.

**Final Justification:**

Thank you for the rebuttal. I have no more questions.

**Key Questions For Authors:**

1. The paper emphasizes that random orderings act as “information-preserving dropout,” yet standard dropout discards information. Can you clarify why your method avoids the instability issues noted in Hausknecht & Wagener (2022) when using different masks at generation vs. training? Is the KL regularization alone sufficient, or does the autoregressive structure play a key role?

2. The method assumes full observability of all variables during generation, but many combinatorial problems (e.g., graph coloring, scheduling) have hard constraints that make certain partial assignments invalid. How would your framework handle infeasible intermediate states, and does the current formulation implicitly assume unconstrained search spaces, as in QUBO?

**Limitations:**

yes

**Strengths And Weaknesses:**

**Strength:**
1. The paper has a clear and interesting high-level idea.
2. The method is technically nontrivial.
3. The empirical comparison among the proposed variants is useful. The benchmark sweep is broad in terms of the number of competing optimizers.

**Weakness:**
1. The paper emphasizes that random orderings act as “information-preserving dropout,” yet standard dropout discards information. Can you clarify why your method avoids the instability issues noted in Hausknecht & Wagener (2022) when using different masks at generation vs. training? Is the KL regularization alone sufficient, or does the autoregressive structure play a key role?

2. The method assumes full observability of all variables during generation, but many combinatorial problems (e.g., graph coloring, scheduling) have hard constraints that make certain partial assignments invalid. How would your framework handle infeasible intermediate states, and does the current formulation implicitly assume unconstrained search spaces like QUBO?

---

> ### Author Rebuttal · Authors · 2026-03-30
>
> We would like to thank the reviewer for their insightful comments.
>
> Answer to questions :
>
> 1. Thank you for pointing this out. Your comment is fully valid, and the instability caused by using different dropout masks at rollout and training time is indeed a well-documented issue in Hausknecht \& Wagener (2022). Interestingly, **we do not observe this behavior in our setting**. As shown in Appendices N.1–N.2, even random input dropout remains entirely stable, and is in fact beneficial: in Figures 8a/9a, varying input dropout already improves performance, and in Figures 8d/9d, our random-ordering scheme (random causal masks at rollout and update) achieves the best results, outperforming both simple dropout and consistent permutations.
>
>    Several factors may explain this discrepancy:
>
>    (1) as detailed in Appendix F (which specifically discusses this point), **we use KL-regularized PPO, which smoothly handles distribution shifts and avoids the gradient-blocking effect that causes clipped PPO to collapse under inconsistent masks**;
>
>    (2) **our masking acts on the input structure rather than on hidden activations**, thereby avoiding the stochastic subnetwork  behavior that can destabilize standard dropout;
>
>    (3) our autoregressive combinatorial tasks are deterministic and incremental, **so mask-induced mismatches do not propagate through the environment dynamics**.
>
>    Taken together, these elements shed light on why using different masks at training time remains stable, and even beneficial, in our experiments. For completeness, note that the $(\sigma,\delta)$-RL-EDA variant of our approach does use consistent dropout as recommended in Hausknecht \& Wagener (2022), although the $(\sigma,\sigma)$-RL-EDA variant performs even better.
>
> 2. We thank the reviewer for this insightful observation regarding the applicability of our framework to constrained combinatorial optimization. The reviewer is correct that our primary experimental focus was on unconstrained search spaces, such as QUBO and NK landscapes. In these settings, every point in the search space is a valid candidate, and the "black-box" nature is defined by the unknown fitness landscape rather than a restricted feasible region. This allowed us to isolate and validate the benefits of order-invariant autoregressive modeling without the interference of invalid solutions in the search space. The transition to constrained problems (e.g., Graph Coloring or Scheduling) does not require a fundamental change to our methodology. Because our model is autoregressive, it generates solutions variable-by-variable. This structure is naturally compatible with valid-action masking, a standard technique in Reinforcement Learning.
>
>    In a strictly black‑box setting, where constraints are also unknown and feasibility can only be determined after evaluating a candidate, **our framework naturally manages invalid solutions through the reward signal** by assigning infeasible configurations a negative reward or a zero reward. **This situation already arises in the NAS‑Bench‑101 real‑world dataset used in our experiments (Appendix O; results shown in Figure 3)**. In NAS‑Bench‑101, each neural architecture is mapped to its test accuracy (Ying et al., 2019), with architectures encoded by 21 binary and 5 categorical variables, yielding a discrete search space with approximately 510 million configurations. As documented in the original dataset, a large proportion of these architectures are invalid and receive an assigned accuracy of 0, leaving only 423,000 valid models with strictly positive accuracy. Despite this high proportion of invalid points, **our method learns to avoid infeasible regions through its multivariate generative model** and achieves the best performance on NAS‑Bench‑101 after both 1,000 and 10,000 objective‑function evaluations (counting evaluations of invalid models with an accuracy of 0) compared to all competing methods.

---

> > ### Author Rebuttal · Reviewer_FmE6 · 2026-04-04
> >
> > Thank you for the rebuttal. I have no more questions. I will maintain my score. Additionally, based on other reviewers' responses, the authors rely on existing methods, such as PPO, and the paper lacks a theoretical analysis, which may diminish its significance. Therefore, I maintain low confidence.

---

> > > ### Author Response · Authors · 2026-04-06
> > >
> > > Thank you for this rebuttal acknowledgement.
> > >
> > > We also thank the reviewer for raising this point and would like to clarify **the novelty and the theoretical contributions of this paper**. Our work introduces a new framework for Estimation‑of‑Distribution Algorithms, which, to the best of our knowledge, has not been explored before. While it leverages certain known PPO components, their combination, purpose, and theoretical grounding within our framework are different from prior approaches.
> > >
> > > In particular, **Appendix C demonstrates that PPO can be extended to autoregressive models with variable generation and learning orders, while remaining consistent with the theoretical foundations reviewed in Appendix B**. This extension is non‑trivial, novel, and potentially applicable beyond our use case.
> > >
> > > In addition to proposing this framework, **we provide theoretical analyses that explain why its key mechanisms enable effective and robust optimization**. Due to space constraints, these analyses have been included in the appendices. In particular:
> > > - Appendix D analyzes convergence behavior in the infinite‑data limit.
> > > - Appendix E discusses the role of structural dropout in promoting stability and diversity.
> > > - Appendix G highlights the connection of the GRPO-based approach to natural gradients and the Information Geometric Optimization framework (Ollivier et al., 2017), including the use of rank‑based rewards similar to those employed in the well-known CMA‑ES algorithm (Hansen and Ostermeier 2001).
> > >
> > > We hope this clarifies both the novelty and theoretical contributions of our work, and we appreciate the reviewer for prompting us to make these aspects more explicit.

---

### Official Review · Reviewer_h3K3 · 2026-03-23

**Soundness:** 3
**Presentation:** 3
**Significance:** 2
**Originality:** 3
**Overall Recommendation:** 4
**Confidence:** 4

**Summary:**

This paper studies discrete black-box combinatorial optimization problems. The authors reformulate the sampling process of multivariate EDAs as a sequential decision problem, and use reinforcement learning to train an autoregressive generative model. Instead of using a fixed variable order, they introduce random orders during both generation and training, so that the model learns an order-invariant policy, which improves exploration.

For training, the authors use a PPO-style objective and adopt a GRPO-like relative advantage design, making the updates more robust to the scale of rewards. The experiments are conducted on standard benchmarks including QUBO, NK, and NK3, and compared with a large number of methods from Nevergrad as well as classical EDAs. The results show that the proposed method performs well in most cases and can effectively avoid catastrophic failures.

**Compliance With Llm Reviewing Policy:**

Affirmed.

**Final Justification:**

The rebuttal addressed my main concerns and I maintain my positive rating of 4.

**Key Questions For Authors:**

1. In the main experiments, the method is compared with many Nevergrad algorithms. Can the authors further clarify whether the conclusions still hold when comparing only with the most relevant and strongest discrete optimization or EDA baselines?

2. How sensitive is the method to RL-related hyperparameters, especially those related to GRPO? Does it require significant tuning for different optimization problems?

3. Are there existing methods that apply RL to black-box combinatorial optimization? Can the authors provide a clearer comparison in terms of methodology and performance?

4. Besides the number of function evaluations, can the authors provide more direct measures of computational cost, such as training time and memory or compute requirements, and compare them with traditional EDA methods?

**Limitations:**

Yes.

**Strengths And Weaknesses:**

Strengths

First, the problem formulation and motivation are reasonable. Modeling the solution construction process as an MDP makes it natural to apply RL methods.

Second, the discussion of order invariance is interesting. The authors use random generation orders in both sampling and training, and provide empirical evidence to support this design. This helps improve diversity and exploration.

Third, the experiments are extensive. The authors compare multiple variants of their method and conduct large-scale comparisons with many existing black-box optimization methods. They also include ablation studies to analyze the impact of different components.

Fourth, from an RL perspective, the training objective is well designed. The PPO-style objective and GRPO-inspired advantage are suitable for black-box optimization, where rewards can be unstable, samples are expensive, and training a critic is difficult.


Weaknesses

First, the paper explains the use of random generation orders as a form of “information-preserving dropout”, which is an interesting idea. However, this point could be supported by more detailed theoretical analysis.

Second, the discussion on computational cost and scalability is limited. The paper does not clearly show how the method compares with traditional optimization methods in terms of training cost, stability, and sensitivity to hyperparameters. This is especially important since the method introduces RL training.

Third, the applicability of the method is not very clear. While the method performs well on discrete black-box problems, it is not clear where the main advantage comes from, and whether it generalizes well across different types of problems.

---

> ### Author Rebuttal · Authors · 2026-03-30
>
> We would like to thank the reviewer for their insightful comments.
>
> Addressing weaknesses:
>
> 1. We appreciate the question about how random generation orders relate to an information-preserving form of structural dropout. We would like to first refer to **Appendix E (‘Order Permutations vs Input Dropout’)**, where we formally analyze this point. Unlike classical input dropout, permutations never discard information and maintain **stable inference under a preserved full-joint distribution**. Across permutations, every variable appears in many informative contexts, and once the model is order-invariant, each trajectory encodes the same dependencies regardless of order. This analysis supports our claim that random generation orders constitute a theoretically grounded regularization mechanism rather than an ad-hoc heuristic. Finally, they increase population diversity and provide a principled structural regularization, which helps extraction of main dependencies between problem variables.
>
> 2. **Appendices L, N, Q, R provide detailed complexity and runtime analysis, scalability experiments up to $n=1024$, and extensive stability/hyperparameter studies**, all demonstrating that our approach remains computationally practical and stable while outperforming classical EDAs and Nevergrad algorithms on different black box problems.
>
> 3. The advantages of our method come directly from its architecture: **the order-invariant autoregressive model and the permutation-based regularization**, which together enable robust search in discrete spaces where gradients and smooth structure are absent. **The approach generalizes well across very different discrete families** (binary QUBO and NK, categorical NK3, and real‑world discrete NAS benchmark), without hyperparameter tuning, as shown across hundreds of baselines. Although our experiments focus on discrete problems, extending the framework to continuous variables only requires changing the output distributions, while the overall architecture and update rules remain the same.
>
> Answer to questions :
>
> 1. We perfectly understand possible misinterpretation of the proposed experimental conslusions. In Figures 2 and 3, only the best performing competitors are plotted. Nevertheless, in **Appendix M, we report results against all 500+ competing algorithms from the Nevergrad library as well as the three most relevant EDA baselines (PBIL, MIMIC, BOA)**. The complete list of competitors is given in Appendix T.
>
> 2. **Our method does not require problem‑specific tuning**. As shown in Figures 2, 3, 6, and 7, as well as in the comprehensive results reported in Table 4 (Appendix M), (σ,σ)-RL-EDA performs consistently well across a diverse set of discrete black‑box optimization problems, including pseudo‑Boolean QUBO and NK landscapes of different sizes with smooth (NK with K=1) and  rugged fitness landscapes (NK with K=8), categorical problems NK3, and the real‑world NASBench benchmark, without any modification of its hyperparameters.  The sensitivity of our method to the main GRPO‑related hyperparameters—namely the group size λ, the KL‑regularization coefficient β, and the number of epochs per generation E, is already documented in Appendices N.4, N.5, and N.7, respectively. To complete this answer, we additionally conducted a sensitivity study on the Adam learning rate. We found that standard values in the range $10^{-3}$ to $10^{-4}$ provide the most stable and performant behavior. We also experimented with replacing our utility weighting scheme $U(x)=1-2x$, which assigns weights in [-1,1] with mean 0 (as a true advantage function), by a CMA‑ES-like truncation scheme (Hansen et al.,2001), where the top μ individuals receive weight 1/μ and others 0. Across all tested settings, this alternative  consistently underperformed the original formulation.
>
>
>
> 3. Existing RL methods for combinatorial optimization (e.g., Bello et al., 2017; Kool et al., 2019; Nazari et al., 2018; Kwon et al., 2020; Kim et al., 2023; Chung et al., 2025; Khalil et al., 2017; Chen \& Tian, 2019) mostly exploit problem-specific structure, and learn general heuristics, making them unsuitable for fully black-box settings.
>
>    **In contrast, our method is data-scarce, problem-agnostic, and black-box, and to our knowledge, the first to leverage an autoregressive construction model for this purpose**. In this context, we show that accurate performance requires order permutations and structured regularization.
>
>    **Our evaluation includes a broad set of 500+ baselines, covering adaptive, learning-enhanced, and portfolio-based strategies**; the full list is in Appendix T.
>
> 4. **Appendix L provides direct computational-cost measurements**, including wall-clock training time, CPU/GPU runtimes, and memory/compute complexity, and reports real runtimes for multiple instance sizes and shows that our method remains competitive or faster than multivariate EDAs, such as BOA, despite using RL updates.

---

> > ### Author Rebuttal · Reviewer_h3K3 · 2026-04-02
> >
> > Thank you for the detailed rebuttal.
> >
> > The additional clarifications on computational cost, scalability, and hyperparameter sensitivity are helpful and improve my understanding of the method. In particular, the discussion on sensitivity analysis and the additional experimental details strengthen my confidence in the empirical results.
> >
> > I appreciate the authors’ efforts in the rebuttal. My overall positive assessment remains unchanged.

---

### Official Review · Reviewer_yNpH · 2026-03-23

**Soundness:** 2
**Presentation:** 2
**Significance:** 2
**Originality:** 2
**Overall Recommendation:** 4
**Confidence:** 5

**Summary:**

This paper proposes a reinforcement learning (RL)-based estimation-of-distribution algorithms (EDAs) for black-box combinatorial optimization. Motivated by the fact that most EDAs fail to capture complex variable interactions, this work designs an order-invariant parameterization with casual masks and adopts randomly sampled orders when generating solutions, yielding a group of diverse outputs, which increases the probability of finding the best variable permutation (interaction) and enhances the exploration ability. Experiments on several benchmark problems showcase the superior performance of the proposed RL-EDA method.

**Compliance With Llm Reviewing Policy:**

Affirmed.

**Ethical Review Flag:**

Flag this paper for an ethics review.

**Final Justification:**

My concerns have been addressed.

**Key Questions For Authors:**

To my knowledge, the proposed causal mask can also work for popular sequence models, e.g., transformer. Why the authors choose to use an MLP-based neural networks for sequential generation?

**Limitations:**

Yes.

**Strengths And Weaknesses:**

Strengths:
1. The proposed order-invariant sampling and training is reasonable to increase the solution diversity and thus improve the exploration of EDAs, which is important for solving complex combinatorial optimization problems. The authors also designed a corresponding MLP-based neural network and group-based training methods to facilitate this core mechanism. These contributions are non-trivial.

2. Experiments are comprehensive to cover comparison, ablation studies, sensitivity analysis and further improvement of the current method. The results demonstrate the effectiveness of the proposed method.

Weaknesses:
1. The idea of leveraging diverse samples for exploration and computing group-relative advantages for stable training has been a widely adopted paradigm in neural methods for combinatorial optimization, especially vehicle routing problems, which is first proposed in POMO [1]. I understand that the mechanism for achieving diversity is different between this work and POMO (POMO used different start nodes for different solutions, i.e., symmetry), but the core mechanism of diversity and group-relative advantage is indeed similar.

2. The performance of the proposed neural method is consistently poor when the evaluation budget is tight, as shown in Figure 2.

Refs:
[1] POMO: policy optimization with multiple optima for reinforcement learning. In NeurIPS 2020.

---

> ### Author Rebuttal · Authors · 2026-03-30
>
> We thank the reviewers for their insightful comments.
>
> Addressing weaknesses:
>
> 1. Thanks for this question. First, please note that our introduction (line 82) already mentions POMO and explicitly position our work w.r.t RL-based constructive methods such as (Kim et al., 2022; Kwon et al., 2020). **These approaches exploit solution-space symmetries** (e.g., rotations or permutations of tours), and POMO in particular enforces diversity by sampling solutions from multiple starting nodes. This is a natural strategy for permutation-structured problems like TSP, where dimensions correspond to interchangeable entities (cities). Similar forms of symmetry can arise in graphs or text.
>
>    Our setting, however, is fundamentally different. In black-box optimization each variable corresponds to distinct information, and we have no prior knowledge of potential symmetries. **The invariance we leverage concerns the generation order itself, implemented via random causal masks**. This invariance does not alter the solution space, unlike constructive approaches, but instead encourages a more flexible internal representation, helping the model uncover variable dependencies of increasing complexity. By varying the generation order, we expect simpler modes (e.g., univariate, bivariate) to be more easily detected, as discussed in Appendix E.
>
>    We also respectfully disagree with the claim that POMO and our method rely on the same group-relative advantage mechanism. POMO is built upon a REINFORCE estimator with a specific baseline design. **In contrast, our training procedure is order-invariant, uses rank-based utilities, and includes explicit KL control, with theoretical justification provided in Appendices C and D**. These distinctions lead to a training process that is conceptually and practically different from POMO’s.
>
> 2. We acknowledge that our algorithm (σ,σ)-RL-EDA may require more evaluation budget to surpass certain baselines in some cases as seen in Figure 2. However, this behavior is expected and **does not undermine the practicality of our approach for several reasons**, which we detail below based on results that we report in Appendices O and P:
>
> - First, performance depends on instance characteristics: on QUBO or NK landscapes with small size ($n=64$)  our method is the best with a low budget  (See Table 5 in Appendix P for detailed results with low budget of 1,000 evaluations). A similar trend appears in our real‑world NAS experiment (Appendix O), where **our approach is the top performer for small budgets (Figure 3)**.
> - Second, EDAs are designed to start with maximum entropy in the sampling distribution and gradually reduce entropy to avoid premature convergence to local optima. Our RL-based approach follows this principle, which explains slower early progress but ensures robustness in high-dimensional, multimodal landscapes.
> - Third, comparing against 500 algorithms means some metaheuristics will always appear better for a specific evaluation budget; it may be easy to design a heuristic that converges very quickly but gets trapped in poor local optima. Our method prioritizes exploration-exploitation balance and long-term performance.
> - Fourth, learning a multivariate model is inherently more complex than univariate models like PBIL, which explains the initial gap. To overcome this problem, we introduce in Appendix P a  variant of our algorithm called Fast-(σ,σ)-RL-EDA which use a curriculum approach, so that the model is univariate at the start of the search in order to find a good-quality solution more quickly at the beginning, then gradually switches to a multivariate mode. The results presented in Table 6 in Appendix P show that **this variant frequently obtain the best results in comparison with all the baseline for a wide variety of distributions of instances of different sizes with a low budget of 1,000 evaluations**.
>
> Answer to the question :
>
> 1. We thank the reviewer for the question regarding our architectural choice. While the proposed causal masking mechanism can indeed be applied to sequence models such as Transformers, our setting differs from typical unstructured sequence domains where token semantics depend on context. **In our BBO setting, each dimension corresponds to a fixed and semantically distinct variable, and the input vector always preserves the same canonical ordering**.
>
>    Given this tabular and structured nature of the domain, an MLP is a natural fit: **it directly models fixed-dimensional inputs without the overhead associated with positional encodings, attention layers, or the inductive biases required for variable-length sequences**. Although Transformers could be applied in principle, they introduce unnecessary architectural complexity for this setting without offering clear advantages. Our goal is to isolate the effect of order-invariant training through random causal masks, and an MLP provides a simpler and more controlled environment for demonstrating this effect.

---

> > ### Author Rebuttal · Reviewer_yNpH · 2026-04-04
> >
> > I appreciate the response of the authors.

---

### Decision · Program_Chairs · 2026-04-30

**Decision:**

Accept (regular)

**Comment:**

This paper proposes a reinforcement learning-based estimation-of-distribution algorithms (EDAs) for black-box combinatorial optimization. The authors reformulate the sampling process of multivariate EDAs as a sequential decision problem, and design an order-invariant parameterization with casual masks and adopt randomly sampled orders when generating solutions, enhancing the exploration ability.

Though some reviewers have concerns on the novelty and the practical performance under tighter evaluation budgets, they generally appreciate the value of this work: reasonable problem formulation and method, as well as comprehensive experiments.